# T cell cholesterol efflux suppresses apoptosis and senescence and increases atherosclerosis in middle aged mice

Venetia Bazioti [1,2], Anouk M. La Rose [1], Sjors Maassen[3], Frans Bianchi [3], Rinse de Boer[3], Benedek Halmos[1], Deepti Dabral [3], Emma Guilbaud[4], Arthur Flohr-Svendsen [5], Anouk G. Groenen [1], Alejandro Marmolejo-Garza[1], Mirjam H. Koster[1], Niels J. Kloosterhuis[1], Rick Havinga[1], Alle T. Pranger[6], Miriam Langelaar-Makkinje[1], Alain de Bruin[1,7], Bart van de Sluis [1], Alison B. Kohan[8], Laurent Yvan-Charvet [4], Geert van den Bogaart [3] & Marit Westerterp [1✉]

Atherosclerosis is a chronic inflammatory disease driven by hypercholesterolemia. During aging, T cells accumulate cholesterol, potentially affecting inflammation. However, the effect of cholesterol efflux pathways mediated by ATP-binding cassette A1 and G1 (ABCA1/ABCG1) on T cell-dependent age-related inflammation and atherosclerosis remains poorly understood. In this study, we generate mice with T cell-specific *Abca1/Abcg1*-deficiency on the low-density-lipoprotein-receptor deficient (*Ldlr*$^{-/-}$) background. T cell *Abca1/Abcg1*-deficiency decreases blood, lymph node, and splenic T cells, and increases T cell activation and apoptosis. T cell *Abca1/Abcg1*-deficiency induces a premature T cell aging phenotype in middle-aged (12–13 months) *Ldlr*$^{-/-}$ mice, reflected by upregulation of senescence markers. Despite T cell senescence and enhanced T cell activation, T cell *Abca1/Abcg1*-deficiency decreases atherosclerosis and aortic inflammation in middle-aged *Ldlr*$^{-/-}$ mice, accompanied by decreased T cells in atherosclerotic plaques. We attribute these effects to T cell apoptosis downstream of T cell activation, compromising T cell functionality. Collectively, we show that T cell cholesterol efflux pathways suppress T cell apoptosis and senescence, and induce atherosclerosis in middle-aged *Ldlr*$^{-/-}$ mice.

[1] Department of Pediatrics, University Medical Center Groningen, University of Groningen, 9713 AV Groningen, the Netherlands. [2] Institute for Cardiovascular Prevention (IPEK), Ludwig-Maximilians-Universität, 80336 Munich, Germany. [3] Department of Molecular Immunology and Microbiology, Groningen Biomolecular Sciences and Biotechnology Institute, University of Groningen, 9747 AG Groningen, the Netherlands. [4] Institut National de la Santé et de la Recherche Médicale (INSERM) U1065, Université Côte d'Azur, Centre Méditerranéen de Médecine Moléculaire (C3M), Atip-Avenir, Fédération Hospitalo-Universitaire (FHU) Oncoage, 06204 Nice, France. [5] European Research Institute for the Biology of Ageing, University Medical Center Groningen, University of Groningen, 9713 AV Groningen, the Netherlands. [6] Laboratory of Medicine, University Medical Center Groningen, University of Groningen, 9713 AV Groningen, the Netherlands. [7] Department of Biomolecular Health Sciences, Dutch Molecular Pathology Center, Faculty of Veterinary Medicine, Utrecht University, 3584 CL Utrecht, the Netherlands. [8] Division of Endocrinology and Metabolism, Department of Medicine, University of Pittsburgh, Pittsburgh, PA 15260, USA. ✉email: m.westerterp@umcg.nl

A therosclerosis is a lipid-driven chronic inflammatory disease of the large- and mid-sized arteries that can lead to myocardial infarction or stroke[1]. T cells are key contributors to atherosclerosis, by secreting pro- and anti-inflammatory cytokines that affect plaque formation[2,3]. While initially most T cells in atherosclerotic plaques were reported to be of the T helper 1 ($T_h1$) phenotype that express the transcription factor Tbet and are pro-atherogenic due to secretion of the cytokines interferon γ (IFN-γ) and tumor necrosis factor (TNF), later studies have shown that regulatory T cells ($T_{reg}$) can be anti-atherogenic by secreting transforming growth factor beta (TGF-β) and interleukin 10 (IL-10)[2,4]. More recent studies revealed that during atherosclerosis and cardiovascular disease (CVD), $T_{reg}$ cells acquire pro-inflammatory characteristics of $T_h1$ and T follicular helper ($T_{FH}$) cells[5–7]. Feeding a cholesterol-rich Western-type diet (WTD) to mice deficient in apolipoprotein E ($Apoe^{-/-}$) for 15 weeks induces a phenotypic switch from $T_{reg}$ cells to $T_h1$ and $T_{FH}$ cells[5]. This phenotypic switch was prevented by injections of apolipoprotein A-I (apoA-I), which stimulates cholesterol efflux. Hence, these data suggest that cholesterol accumulation renders $T_{reg}$ cells pro-inflammatory[5]. Similarly, increased plasma membrane cholesterol accumulation in CD8+ T cells deficient in the enzyme Acetyl-CoA cholesterol Acyltransferase 1 (ACAT1) that esterifies cholesterol, enhances differentiation of naïve T cells into IFN-γ producing T cells[8], presumably with pro-atherogenic effects. However, a recent study showed that in advanced atherosclerosis induced by 20 weeks of WTD feeding in $Apoe^{-/-}$ mice, IFN-γ was decreased in CD4+Tbet+($T_h1$) cells, suggesting that CD4+ T cell functionality is compromised in advanced atherosclerosis[9], when these T cells accumulate cholesterol[5]. While exacerbated cholesterol accumulation in T cells thus has mixed effects on T cell inflammation, its consequences for atherogenesis have not been studied directly.

T cells from aged (~73–76 years old) compared to young (~22–23 years old) humans show increased membrane cholesterol accumulation[10,11]. During aging, the number of naïve T cells decreases, while activated T cells that show features of cellular senescence and that secrete pro-inflammatory cytokines increase[12,13]. This process may contribute to inflammaging[14], the pro-inflammatory state that develops during aging. Aging is a major risk factor for atherosclerosis, presumably due to inflammaging[15]. Although thymic atrophy may contribute to the decrease in naïve T cells and increase in activated T cells with features of cellular senescence that promote inflammaging[16], data in mice with T cell Acat1 deficiency or Apoe deficiency[5,8,9] suggest that cholesterol accumulation in T cells during aging may also directly affect secretion of pro-inflammatory cytokines.

The cholesterol transporters ATP Binding Cassette A1 and G1 (ABCA1 and ABCG1) mediate cholesterol efflux to apoA-I and high-density-lipoprotein (HDL), respectively[17–19]. Previous studies have shown that activation of the T cell receptor (TCR) by αCD3, which increases T cell proliferation, decreases expression of Abca1 and Abcg1 by >90%[20], leading to T cell plasma membrane cholesterol accumulation[20,21]. It has been suggested that Abcg1 is the most highly expressed cholesterol transporter in T cells[20]. T cell Abcg1 deficiency increases differentiation of naïve T cells into $T_{reg}$ cells, which suppresses atherosclerosis[22]. Even though it has been reported that Abca1 expression in T cells is low[20], $Abcg1^{-/-}$ T cells show a 6-fold increase in Abca1 expression[23], suggesting that similar to macrophages[24], the cholesterol transporters ABCA1 and ABCG1 show mutual compensation in T cells.

Here we show that combined T cell Abca1/Abcg1 deficiency, which induces T cell cholesterol and cholesteryl ester (CE) accumulation by ~2.5-fold, decreases peripheral T cell numbers, and induces T cell activation and T cell apoptosis in wild-type

and $Ldlr^{-/-}$ mice. T cell Abca1/Abcg1 deficiency decreases T cell functionality in terms of fighting pathogens and mounting an efficient immune response, presumably due to increased T cell apoptosis. Further, T cell Abca1/Abcg1 deficiency induces a premature T cell aging phenotype, reflected by upregulation of senescence markers. While not affecting atherogenesis in young $Ldlr^{-/-}$ mice, T cell Abca1/Abcg1 deficiency decreases atherogenesis in middle-aged (~12–13 months old) $Ldlr^{-/-}$ mice fed a chow diet, accompanied by decreased T cells in atherosclerotic plaques and a decrease in aortic inflammation, which we attribute to increased T cell apoptosis. These data reveal that cholesterol efflux pathways suppress T cell apoptosis, which promotes T cell functionality, but increases atherogenesis during aging.

## Results

**T cell Abca1/Abcg1 Deficiency Increases Cholesterol Accumulation.** We generated a T cell specific Abca1/Abcg1-deficient mouse model by crossbreeding $Abca1^{fl/fl}Abcg1^{fl/fl}$ mice with mice expressing the T cell specific CD4Cre promoter. The CD4Cre promoter starts to be expressed at the CD4+CD8+ double positive (DP) stage of thymic T cell maturation and results in deletion of loxP flanked genes in DP and CD4+ or CD8+ single positive (SP) thymocytes[25]. To assess the deletion of Abca1 and Abcg1 in T cells, we isolated splenic T cells, since the yield of thymic SP thymocytes was too low to properly assess the expression of these transporters. In $CD4CreAbca1^{fl/fl}Abcg1^{fl/fl}$ splenic T cells, Abca1 and Abcg1 mRNA expression were decreased by >90% compared to $Abca1^{fl/fl}Abcg1^{fl/fl}$ T cells (Supplementary Fig. 1a). To assess the effect of T cell ABCA1 and ABCG1 mediated cholesterol efflux pathways on atherosclerosis, $CD4CreAbca1^{fl/fl}Abcg1^{fl/fl}$ mice were crossbred with $Ldlr^{-/-}$ mice to generate $CD4CreAbca1^{fl/fl}Abcg1^{fl/fl}Ldlr^{-/-}$ mice and $Abca1^{fl/fl}Abcg1^{fl/fl}Ldlr^{-/-}$ controls. We refer to these mice as $T$-$Abc^{dko}Ldlr^{-/-}$ and $Ldlr^{-/-}$ mice, respectively.

To assess the functional consequences of T cell Abca1/Abcg1 deficiency, we performed filipin staining to measure free cholesterol accumulation. T cell Abca1/Abcg1 deficiency increased filipin staining in thymic DP, CD4+, and CD8+ SP cells (Supplementary Fig. 1b–d), and in CD4+ and CD8+ T cells in blood (Supplementary Fig. 1e–g). We next assessed the cellular localization of cholesterol in splenic T cells by confocal microscopy. T cell Abca1/Abcg1 deficiency increased filipin staining at the plasma membrane, reflecting cholesterol accumulation, in $Ldlr^{-/-}$ and $Ldlr^{+/+}$ T cells (Supplementary Fig. 2a, b). We also observed increased staining of choleratoxin B, suggestive of increased ganglioside GM1, a component of cholesterol-enriched lipid rafts (Supplementary Fig. 2c–e).

To validate the findings on cholesterol accumulation, we performed Gas Chromatography – Mass Spectrometry (GC-MS). T cell Abca1/Abcg1 deficiency increased total cholesterol by 2.5-fold, reflected by increases in free cholesterol and cholesteryl esters (CE) (Supplementary Fig. 2f). In line with CE accumulation, T cell Abca1/Abcg1 deficiency increased Oil Red O staining, reflecting lipid droplets in para-aortic lymph nodes (LNs) (Supplementary Fig. 2g), and BODIPY 493/503 staining in $Ldlr^{-/-}$ and $Ldlr^{+/+}$ T cells (Supplementary Fig. 3a, b). We also observed BODIPY 493/503 staining reflecting lipid droplets, in Abcg1-deficient T cells, in line with CE accumulation data as shown in[23], but not in Abca1-deficient T cells or controls (Supplementary Fig. 3c). To assess whether specific cell organelles were affected in Abca1/Abcg1-deficient T cells, we performed transmission electron microscopy. We confirmed the presence of large lipid droplets in Abca1/Abcg1-deficient CD4+ and CD8+ T cells, and otherwise observed no overt differences (Supplementary Fig. 3d, e). Previous studies have shown that Abcg1

deficiency increases sterol regulatory element binding transcription factor 1 (Srebf1) mRNA expression in CD4[+] T cells, while decreasing Ldlr, 3-hydroxy-3-methyl-glutaryl-coenzyme A reductase (Hmgcr), HMG-CoA synthase (Hmgcs), and Srebf2 mRNA expression, reflecting a decrease in cholesterol synthesis, consistent with increased cholesterol accumulation in the ER[23,26]. In line with these data, T cell Abca1/Abcg1 deficiency moderately decreased Srebf2 mRNA expression in CD4[+] T cells and Ldlr and Hmgcr mRNA expression in CD8[+] T cells (Supplementary Fig. 4a, b).

Collectively, previous studies have shown that Abcg1 deficiency increases membrane cholesterol[22] and choleratoxin B staining reflecting increased lipid rafts as well as CE in T cells[23]. We here show that T cell Abcg1 deficiency increases intracellular lipid droplets, consistent with CE accumulation, while Abca1 deficiency shows no effect, in line with Abcg1 being the predominant T cell cholesterol transporter[20]. However, Abcg1 deficiency in T cells increases Abca1 expression by 6-fold[23], suggesting that Abca1 may contribute to cholesterol accumulation in the setting of Abcg1 deficiency. Consistently, we found that combined deficiency of Abca1 and Abcg1 in T cells increased both free and esterified cholesterol, reflected by plasma membrane cholesterol accumulation and presence of lipid droplets, which was independent of Ldlr expression.

**T cell Abca1/Abcg1 deficiency leads to plasma membrane stiffening.** High concentrations of plasma membrane cholesterol increase plasma membrane stiffness in model membranes and cells[27]. To examine whether plasma membrane stiffness was affected by T cell Abca1/Abcg1 deficiency, we stained T cells with the fluorescent dye BODIPY C10 and performed Fluorescence-Lifetime Imaging Microscopy (FLIM). BODIPY C10 is a molecular rotor that has stiffness-dependent fluorescence lifetime[28]. When BODIPY C10 binds to a fluid membrane, it shows high rotational speed which decreases its fluorescence lifetime, while binding to a stiff membrane leads to increased fluorescence lifetime[28]. Abca1/Abcg1-deficient CD4[+] and CD8[+] T cells showed increased BODIPY C10 fluorescence lifetime compared to control CD4[+] and CD8[+] T cells (Fig. 1a, b and Supplementary Fig. 5a–d), suggesting that T cell Abca1/Abcg1 deficiency increased cell stiffness. In Abca1/Abcg1-deficient CD4[+] and CD8[+] T cells, structures that resemble CE-rich lipid droplets showed yellow staining, which reflects very high fluorescence lifetime (Fig. 1a, b). When we excluded these from analysis, BODIPY C10 fluorescence lifetime was still increased in Abca1/Abcg1-deficient CD4[+] and CD8[+] T cells compared to control CD4[+] and CD8[+] T cells (Fig. 1c–f). These data suggest that cholesterol accumulation in Abca1/Abcg1-deficient T cells leads to plasma membrane stiffening.

**Effects of T cell Abca1/Abcg1 deficiency on peripheral T cells and T cell activation.** LckCreAbca1[fl/fl]Abcg1[fl/fl] mice with T cell specific Abca1/Abcg1 deficiency have decreased thymic CD4[+] and CD8[+] SP cells compared to littermate controls and decreased splenic CD4[+] and CD8[+] T cells[29]. We thus assessed whether T cell Abca1/Abcg1 deficiency affected the number of T cells in T-Abc[dko]Ldlr[−/−] mice. While single T cell Abca1 or Abcg1 deficiency did not affect T cells as a percentage of total blood leukocytes or after correction for total blood leukocyte counts, as measures for T cell concentration in blood (Supplementary Fig. 6a–c), combined T cell Abca1/Abcg1 deficiency decreased blood CD4[+] and CD8[+] T cells by ~50% in Ldlr[−/−] (Fig. 2a–c and Supplementary Fig. 6a) and Ldlr[+/+] mice (Supplementary Fig. 6d). Because of these decreases, we evaluated thymic cell populations. T cell Abca1/Abcg1 deficiency did not affect thymic

weight (Supplementary Fig. 7a). T-Abc[dko]Ldlr[−/−] mice showed no changes in thymic CD4[−]CD8[−] double negative (DN) cells, thymic CD4[+]CD8[+] DP cells, or thymic CD4[+] or CD8[+] SP cell populations compared to littermate controls, when shown as percentage of thymocytes, or after correction for total thymic cell numbers (Supplementary Fig. 1b and Supplementary Fig. 7b–d), even though TCRβ[+]CD24[−] and TCRβ[+]CD69[−] cells, indicative of excessive negative selection[30], were decreased compared to Ldlr[−/−] controls as a percentage of thymocytes (Supplementary Fig. 7e–k). Also, unlike LckCreAbcg1[fl/fl] mice that showed an increase in thymic T[reg] cells[22], thymic T[reg] cells were not different between T-Abc[dko]Ldlr[−/−] mice and controls (Supplementary Fig. 7l–o). We next examined T cell numbers in spleen and para-aortic LNs. Similar to observations in blood, we found that T cell Abca1/Abcg1 deficiency decreased splenic and para-aortic LN CD4[+] and CD8[+] T cells by ~50–70% (Fig. 2d–g). Together these data indicate that T cell Abca1/Abcg1 deficiency decreases blood, splenic, and para-aortic LN T cells without affecting thymic DN, DP, CD4[+], or CD8[+] SP cells.

We have previously shown that T cell Abca1/Abcg1 deficiency mildly enhanced CD4[+] T cell activation in the LckCreAbca1[fl/fl]Abcg1[fl/fl] model[31]. We thus assessed the effect of single and combined T cell Abca1/Abcg1 deficiency on T cell activation. Single T cell Abca1 or Abcg1 deficiency did not affect T cell activation in blood (Supplementary Fig. 8a–c). Combined T cell Abca1/Abcg1 deficiency increased T[memory/effector] (T[mem/eff]; CD4[+]CD44[+]CD62L[−] and CD8[+]CD44[+]CD62L[−]) and T[central memory] (T[CM]; CD8[+]CD44[+]CD62L[+]) cells as a percentage of CD4[+] or CD8[+] T cells by >3-fold in Ldlr[−/−] (Fig. 2h–j and Supplementary Fig. 8a) and Ldlr[+/+] mice (Supplementary Fig. 8d, e). Further, T cell Abca1/Abcg1 deficiency decreased CD4[+] and CD8[+] T[naive] cells (CD44[−]CD62L[+]) cells by ~50% as a percentage of CD4[+] or CD8[+] T cells in Ldlr[−/−] (Fig. 2h–j) and Ldlr[+/+] mice (Supplementary Fig. 8d, e). We observed similar effects in spleen and para-aortic LNs of Ldlr[−/−] mice (Supplementary Fig. 8f–i). Collectively, as percentages of total CD4[+] or CD8[+] T cells, T cell Abca1/Abcg1 deficiency increased T[mem/eff] and T[CM] cells in blood, spleen, and para-aortic LNs. However, when corrected for total number of blood leukocytes or total number of cells per LN or spleen, the increases in T[mem/eff] and T[CM] cells were no longer significant between the genotypes (Fig. 2k, l and Supplementary Fig. 8j–l). Moreover, T cell Abca1/Abcg1 deficiency decreased CD4[+] and CD8[+] T[naive] cells by >70% (Fig. 2k, l, and Supplementary Fig. 8j, k, m). In sum, while increasing T[mem/eff] and T[CM] subsets as percentages of CD4[+] and CD8[+] T cells, T cell Abca1/Abcg1 deficiency dramatically decreased T[naive] cells in blood, spleen, and para-aortic LNs.

**Effects of T cell Abca1/Abcg1 deficiency on T cell exhaustion.** Previous studies have shown that T cell cholesterol accumulation induces T cell exhaustion, which was mainly attributed to cholesterol accumulation in the ER[32]. Exhausted T cells are prone to apoptosis and show reduced proliferative capacity[32,33]. We therefore investigated whether increased cholesterol accumulation due to loss of Abca1/Abcg1 in T cells would induce T cell exhaustion and thus explain the decrease in T cell numbers.

The percentage of para-aortic LN CD4[+] T cells expressing the exhaustion marker programmed cell death 1 (PD-1) was increased in T-Abc[dko]Ldlr[−/−] mice compared to Ldlr[−/−] controls (Supplementary Fig. 9a–c). However, T cell Abca1/Abcg1 deficiency did not affect the total number of CD4[+]PD-1[+] T cells in whole para-aortic LNs (Supplementary Fig. 9d). T cell Abca1/Abcg1 deficiency induced only a modest increase in cytotoxic T-lymphocyte-associated protein 4 (CTLA4) on CD4[+] and CD8[+] T cells, and T cell immunoglobulin and mucin-

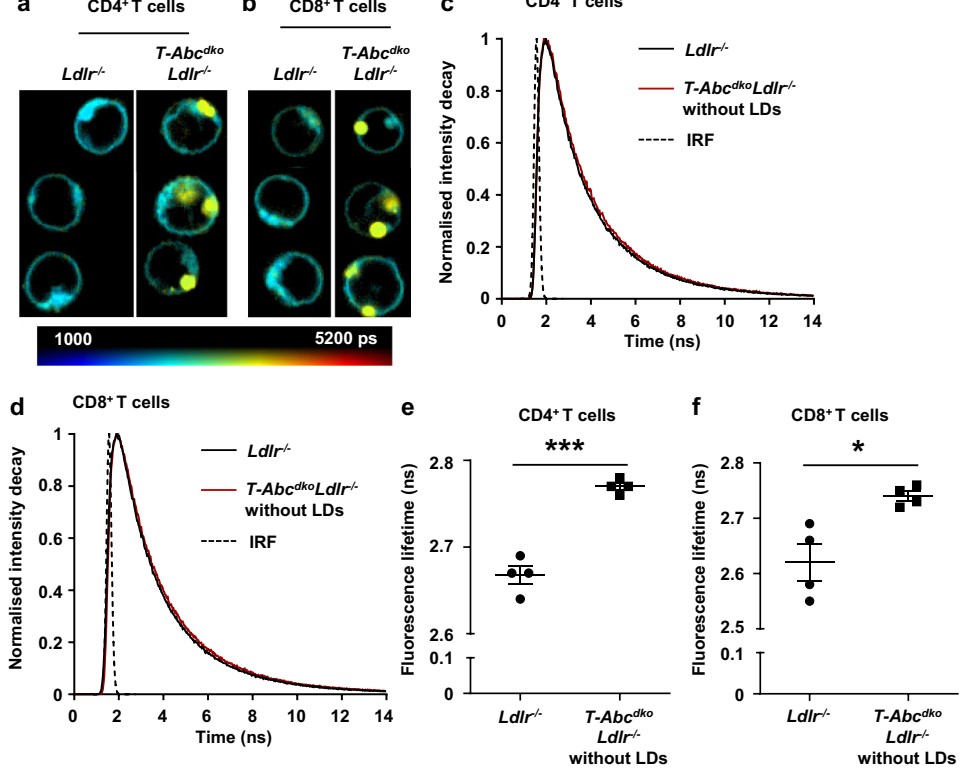

**Fig. 1 T cell *Abca1/Abcg1* deficiency leads to plasma membrane stiffening.** $Ldlr^{-/-}$ and $T\text{-}Abc^{dko}Ldlr^{-/-}$ mice were fed a chow diet. Spleens were collected, CD4+ and CD8+ T cells were isolated, stained with BODIPY C10, and analyzed by Fluorescence-Lifetime Imaging Microscopy (FLIM). Representative microscopy pictures for CD4+ (**a**) and CD8+ (**b**) T cells. Scale bar represents picosec (ps). Representative fluorescence lifetime decay curves for CD4+ (**c**) and CD8+ (**d**) T cells. LDs, lipid droplets. Dashed lines represent fits with mono-exponential decay functions convoluted with the instrument response function (IRF). BODIPY C10 fluorescence lifetime (determined by fitting the fluorescence lifetime decay curves with a mono-exponential decay function) for the plasma membrane of (**e**) CD4+ ($p = 0.00009$) and (**f**) CD8+ ($p = 0.013$) T cells (without LDs). Per mouse, 43–76 CD4+ and CD8+ T cells were analyzed. $n = 4$ $Ldlr^{-/-}$ and $n = 4$ $T\text{-}Abc^{dko}Ldlr^{-/-}$ mice. For all panels, error bars represent SEM. Biologically independent samples were included. $p$ value was determined by unpaired two-tailed Student's $t$ test. *$p < 0.05$, ***$p < 0.001$. Source data are provided as a Source Data file.

domain containing 3 (TIM3) and lymphocyte-activation gene 3 (LAG3) on CD4+ T cells (Supplementary Fig. 9e–l). T cell *Abca1/Abcg1* deficiency increased the transcription factor Eomesodermin (Eomes) in CD8+ T cells (Supplementary Fig. 9m–o). We observed similar effects of T cell *Abca1/Abcg1* deficiency on splenic CD4+PD-1+ T cell numbers and on expression of CTLA4 or Eomes in splenic T cells (Supplementary Fig. 10a–g). Eomes is an exhaustion marker[34], but has also been associated with an increase in $T_{CM}$ cells[35]. Increased PD-1 on CD4+ T cells may be due to increased T cell activation[36]. Since TIM3, LAG3, and CTLA4 were minimally increased by T cell *Abca1/Abcg1* deficiency, and the increase in the percentage of PD-1+CD4+ T cells and Eomes expression on CD8+ T cells could be attributed to the expansion of $T_{mem/eff}$ and $T_{CM}$ populations, respectively, these data are suggestive of only minor effects of T cell *Abca1/Abcg1* deficiency on T cell exhaustion.

**T cell *Abca1/Abcg1* deficiency increases T cell apoptosis upon TCR stimulation**. We then investigated other mechanisms contributing to the decrease in peripheral T cell numbers. Previous studies have shown that stimulation of the TCR by αCD3 decreases the expression of *Abca1* and *Abcg1* by >90%, while upregulating the expression of *Acat1*, *Hmgcr*, and *Ldlr*[20]. These effects were confirmed by a later study[8], and suggest that TCR stimulation induces a change in gene expression that favors cholesterol accumulation, likely to generate substrates for cellular

growth and proliferation. We confirmed that TCR stimulation decreased *Abca1* and *Abcg1* mRNA expression in T cells (Supplementary Fig. 11a, b). In genetic models of plasma membrane cholesterol accumulation in T cells, TCR signaling is enhanced[8,20,23]. Conversely, TCR signaling is suppressed in T cells deficient in SCAP (SREBP cleavage activating protein) that cannot synthesize cholesterol[37]. We thus assessed whether T cell *Abca1/Abcg1* deficiency affected T cell proliferation downstream of TCR signaling. Upon TCR stimulation, *Abca1/Abcg1* deficiency increased T cell proliferation in CD4+ and CD8+ $Ldlr^{-/-}$ and $Ldlr^{+/+}$ T cells (Supplementary Fig. 11c–i). Even though in line with observations of increased T cell proliferation in T cell *Abcg1* deficiency[20,23], these findings cannot explain the decreased peripheral T cells in $T\text{-}Abc^{dko}Ldlr^{-/-}$ mice compared to $Ldlr^{-/-}$ controls. However, these data strongly suggest that T cell *Abca1/Abcg1* deficiency enhances TCR signaling. Consistent with increased TCR signaling, we also observed increased Jun N-terminal kinase (JNK)1/2 phosphorylation upon stimulation with αCD3 in CD4+ and CD8+ T cells (Supplementary Fig. 12a, b). We thus investigated whether T cell *Abca1/Abcg1* deficiency affects other pathways that regulate peripheral T cell numbers, downstream of TCR signaling.

TCR stimulation enhances the differentiation of $T_{naive}$ cells into $T_{mem/eff}$ and $T_{CM}$ cells[38]. The increase in $T_{mem/eff}$ and $T_{CM}$ cells in mice with T cell *Abca1/Abcg1* deficiency (Fig. 2h–j and Supplementary Fig. 8f–i) is consistent with increased TCR signaling. Upon TCR stimulation, T cells differentiate into

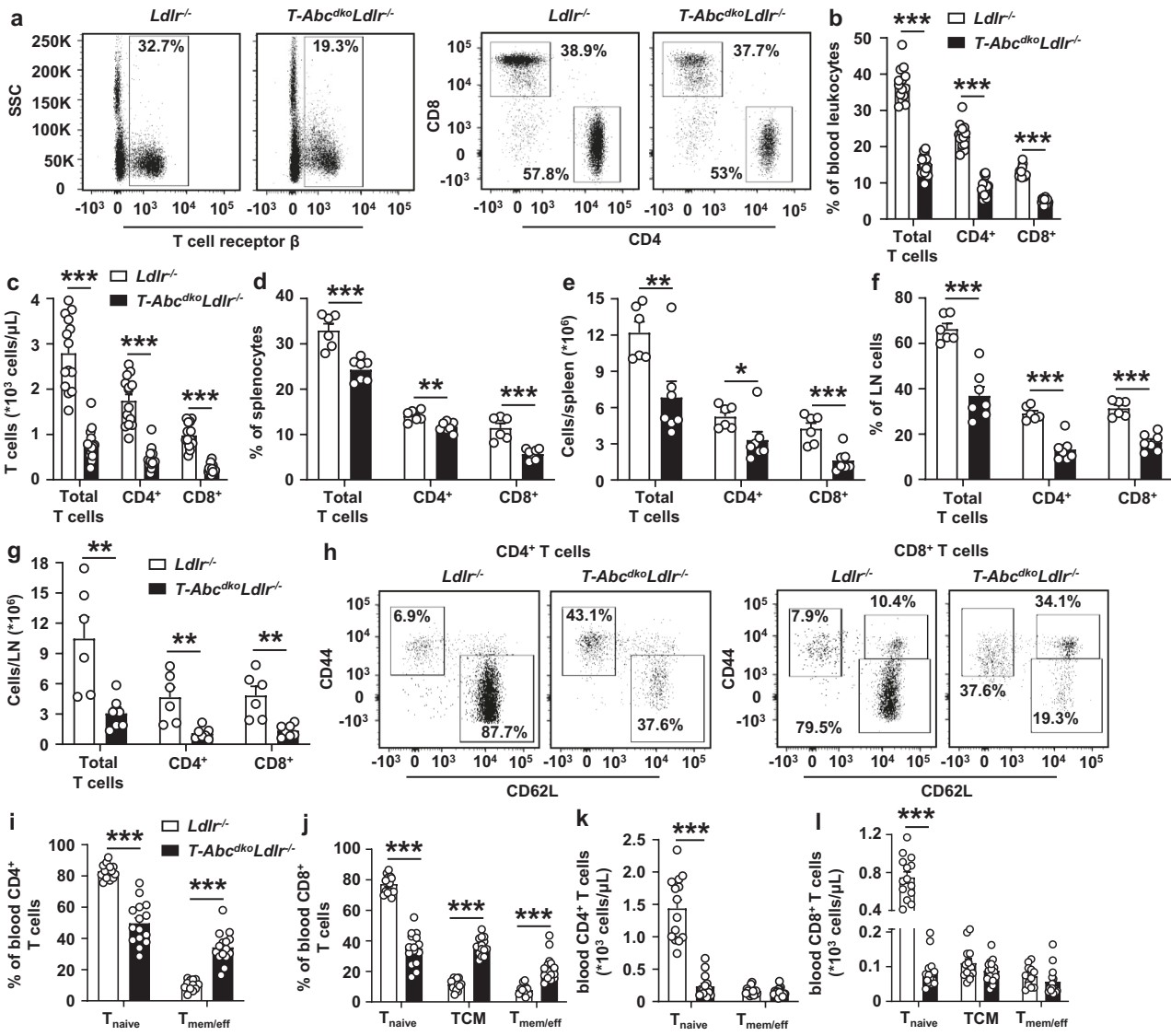

**Fig. 2 T cell *Abca1/Abcg1* deficiency decreases blood, splenic, and para-aortic lymph node T cells and increases T cell activation.** $Ldlr^{-/-}$ and $T\text{-}Abc^{dko}Ldlr^{-/-}$ mice were fed a chow diet. Blood, spleens, and para-aortic lymph nodes (LNs) were collected. Cells were stained with the indicated antibodies and analyzed by flow cytometry. **a** Representative flow cytometry plots of T cell receptor β (TCRβ)$^+$ (total T cells), CD4$^+$, and CD8$^+$ T cells gated as in Supplementary Fig. 6a. **b** Quantification of total ($p < 0.000001$), CD4$^+$ ($p < 0.000001$), and CD8$^+$ ($p < 0.000001$) T cells as percentage of total white blood cells (leukocytes). **c** Total ($p < 0.000001$), CD4$^+$ ($p < 0.000001$), and CD8$^+$ ($p < 0.000001$) T cell concentration in blood after correction for total blood leukocyte counts. $n = 14$ $Ldlr^{-/-}$ and $n = 15$ $T\text{-}Abc^{dko}Ldlr^{-/-}$ mice. Total ($p = 0.00039$), CD4$^+$ ($p = 0.0030$), and CD8$^+$ ($p = 0.00024$) T cells as a percentage of total splenic cells (**d**) and total ($p = 0.0087$), CD4$^+$ ($p = 0.0420$), and CD8$^+$ ($p = 0.00085$) T cells as cells/ spleen after correction for total splenic cell number (**e**). Total ($p = 0.000084$), CD4$^+$ ($p = 0.000037$), and CD8$^+$ ($p = 0.000016$) T cells as a percentage of total para-aortic LN cells (**f**) and total ($p = 0.0044$), CD4$^+$ ($p = 0.00297$), and CD8$^+$ ($p = 0.0021$) T cells as cells/LN after correction for total para-aortic LN cell number (**g**). **d**–**g** $n = 6$ $Ldlr^{-/-}$ and $n = 7$ $T\text{-}Abc^{dko}Ldlr^{-/-}$ mice. Representative flow cytometry plots of (**h**) CD4$^+$CD44$^-$CD62L$^+$ (T$_{naive}$), CD44$^+$CD62L$^-$ (T$_{memory/effector}$ (T$_{mem/eff}$)), CD8$^+$ T$_{naive}$, CD8$^+$CD44$^+$CD62L$^+$ (T$_{central\ memory}$ (T$_{CM}$)), and T$_{mem/eff}$ cells gated as in Supplementary Fig. 7a, (**i**, **j**) quantifications of CD4$^+$ T$_{naive}$ ($p < 0.000001$) and T$_{mem/eff}$ ($p < 0.000001$) cells (**i**) or CD8$^+$ T$_{naive}$ ($p < 0.000001$), T$_{CM}$ ($p < 0.000001$), and T$_{mem/eff}$ ($p = 0.0000027$) cells (**j**) as a percentage of CD4$^+$ (**i**) or CD8$^+$ (**j**) T cells, and (**k**, **l**) concentrations in blood for CD4$^+$ T$_{naive}$ ($p < 0.000001$) and T$_{mem/eff}$ cells (**k**) or CD8$^+$ T$_{naive}$ ($p < 0.000001$), T$_{CM}$, and T$_{mem/eff}$ cells (**l**) after correction for total blood leukocyte counts. $n = 14$ $Ldlr^{-/-}$ and $n = 15$ $T\text{-}Abc^{dko}Ldlr^{-/-}$ mice. **b**, **i**, **j** The experiments were performed twice with the same results. For all panels, error bars represent SEM. Biologically independent samples were included. $p$ value was determined by unpaired two-tailed Student's $t$ test. *$p < 0.05$, **$p < 0.01$, ***$p < 0.001$. Source data are provided as a Source Data file.

T$_{mem/eff}$ cells that may undergo apoptosis, in a pathway known as activation-induced cell death (AICD)[39]. Using αCD3 and interleukin-2 (IL-2) as stimuli, we examined whether T cell *Abca1/Abcg1* deficiency enhanced apoptosis downstream of TCR stimulation, by monitoring expression of cleaved caspase 3/7 in T cells over time using the Incucyte system. T cell *Abca1/Abcg1* deficiency increased cleaved caspase 3/7 by 3.7-fold in CD4$^+$

T cells and 1.8-fold in CD8$^+$ T cells, reflecting increased T cell apoptosis (Fig. 3a, b and Supplementary Fig. 12c). These experiments were carried out in $Ldlr^{-/-}$ T cells. We found that T cell *Abca1/Abcg1* deficiency also increased cleaved caspase 3/7 in $Ldlr^{+/+}$ T cells, as assessed by flow cytometry (Fig. 3c, d and Supplementary Fig. 12d), while single *Abca1* or *Abcg1* deficiency showed no effect (Supplementary Fig. 12d–f). We then evaluated

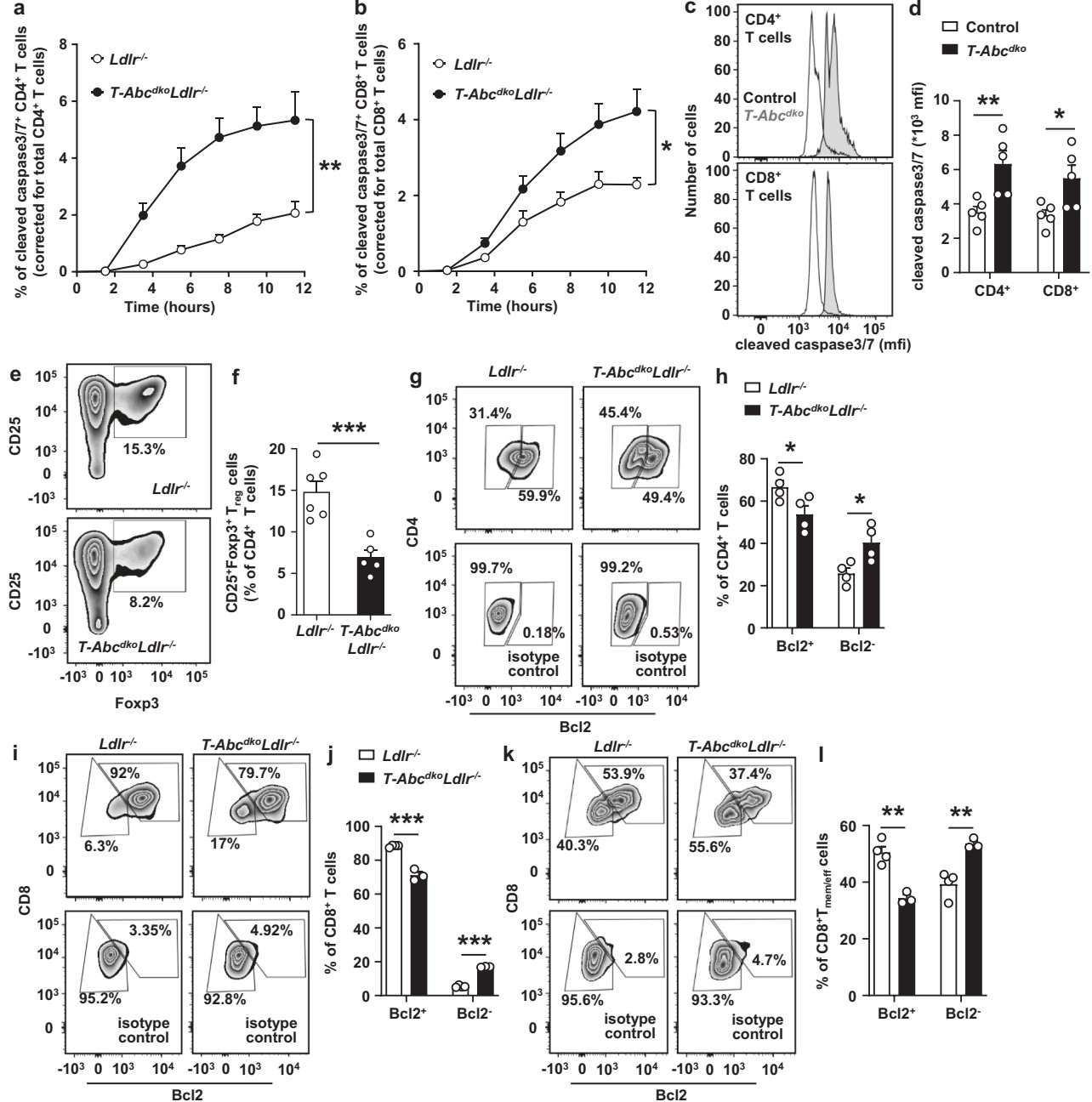

**Fig. 3 T cell *Abca1/Abcg1* deficiency increases T cell apoptosis.** Splenic CD4+ and CD8+ T cells were isolated from *Ldlr*−/−, *T-Abc*dko*Ldlr*−/−, control (*Ldlr*+/+), and *T-Abc*dko mice fed chow diet. **a–d** T cells were stimulated with αCD3 and interleukin 2 (IL-2) for 12 h and stained for cleaved caspase 3/7. **a** CD4+ ($p = 0.0045$) and (**b**) CD8+ ($p = 0.0475$) T cells acquiring cleaved caspase 3/7 staining over time. $n = 4$ *Ldlr*−/− and $n = 4$ *T-Abc*dko*Ldlr*−/− mice. **c** Representative flow cytometry plots of cleaved caspase 3/7 in CD4+ ($p = 0.0101$) and CD8+ ($p = 0.0315$) T cells gated as in Supplementary Fig. 12d, and (**d**) quantification. $n = 5$ control and $n = 5$ *T-Abc*dko mice. **e, f** CD4+ T cells were stimulated with αCD3/αCD28 beads, IL-2, and transforming growth factor beta (TGF-β) for 72 h. **e** Representative flow cytometry plots of CD25+Foxp3+T$_{regulatory}$ cells (T$_{reg}$) gated as in Supplementary Fig. 12h, and (**f**) quantification ($p = 0.00096$) as a percentage of CD4+ T cells. $n = 6$ *Ldlr*−/− and $n = 5$ *T-Abc*dko*Ldlr*−/− mice. **g–l** T cells were stimulated with αCD3/IL-2 for 12 h, fixed, permeabilized, and stained for B-cell lymphoma 2 (Bcl2). **g** Representative flow cytometry plots of CD4+Bcl2+ and CD4+Bcl2− T cells gated as in Supplementary Fig. 13c. **h** Quantification of CD4+Bcl2+ ($p = 0.045$) and CD4+Bcl2− ($p = 0.025$) T cells as a percentage of CD4+ T cells. $n = 4$ *Ldlr*−/− and $n = 4$ *T-Abc*dko*Ldlr*−/−. Same staining for CD8+ T cells as in (**g**) in combination with CD44 and CD62L. Representative flow cytometry plots of CD8+Bcl2+, CD8+Bcl2− (**i**), CD8+T$_{mem/eff}$Bcl2+, and CD8+T$_{mem/eff}$Bcl2− cells (**k**) gated as in Supplementary Fig. 8a. Quantification of CD8+Bcl2+ ($p = 0.00015$) and CD8+Bcl2− ($p < 0.000001$) cells as a percentage of CD8+ T cells (**j**) and CD8+T$_{mem/eff}$Bcl2+ ($p = 0.0017$) and CD8+T$_{mem/eff}$Bcl2− ($p = 0.0041$) cells as a percentage of CD8+T$_{mem/eff}$ cells (**l**). $n = 4$ *Ldlr*−/− and n = 3 *T-Abc*dko*Ldlr*−/−. **f, h, j, l** Data were corrected for their respective isotype controls. For all panels, error bars represent SEM. Biologically independent samples were included. $p$ value was determined by unpaired two-tailed Student's $t$ test. For (**a, b**), $p$ value is based on area under the curve. *$p < 0.05$, **$p < 0.01$, ***$p < 0.001$. Source data are provided as a Source Data file.

whether other modes of cell death may have contributed to the decrease in T cells in mice with T cell *Abca1/Abcg1* deficiency. T cell pyroptosis, a highly pro-inflammatory form of lytic programmed cell death is executed by gasdermin D cleavage[40], and decreases T cell numbers, at least in the setting of chronic HIV-1 infection[41]. Upon αCD3/IL-2 stimulation, we detected a very low level of cleaved gasdermin D p30, which was not different between the genotypes (Supplementary Fig. 12g), indicating that T cell pyroptosis was not affected.

Given that *Abcg1* deficiency alone enhances differentiation of $T_{naive}$ cells in $T_{reg}$ cells[22], and $T_{reg}$ cells are athero-protective, we then also studied $T_{reg}$ differentiation in T cells with combined *Abca1/Abcg1* deficiency. We used TGF-β, αCD3, and IL-2 to induce CD25$^+$Foxp3$^+$T$_{reg}$ differentiation, similar to previous studies[22,42], and found that, in contrast to T cell *Abcg1* deficiency[22], T cell *Abca1/Abcg1* deficiency suppressed $T_{reg}$ differentiation by ~50% (Fig. 3e, f and Supplementary Fig. 12h). Given that, with the exception of TGF-β, stimuli for the apoptosis assay were similar to those used for $T_{reg}$ differentiation, it is likely that the decrease in $T_{reg}$ cells is simply the consequence of increased T cell apoptosis. We thus studied mechanisms that cause T cell apoptosis.

We found that the expression of the death receptor FAS that is highly expressed in lipid rafts and involved in the extrinsic apoptosis pathway[43] was not affected by T cell *Abca1/Abcg1* deficiency upon stimulation with αCD3 and IL-2 (Supplementary Fig. 12d and and Supplementary Fig. 13a, b). We next measured the expression of the anti-apoptotic protein B-cell lymphoma 2 (Bcl2) that regulates the intrinsic apoptosis pathway[44]. Upon αCD3/IL-2 stimulation, T cell *Abca1/Abcg1* deficiency decreased anti-apoptotic Bcl2$^+$CD4$^+$ and Bcl2$^+$CD8$^+$ T cells by ~20%, while increasing pro-apoptotic Bcl2$^-$CD4$^+$ and Bcl2$^-$CD8$^+$ T cells by ~1.6 and ~3.1-fold, respectively (Fig. 3g–j and Supplementary Fig. 13c). However, less than 20% of *Abca1/Abcg1*-deficient CD8$^+$ T cells were Bcl2$^-$, while this was ~40% for CD4$^+$ T cells (Fig. 3g–j), which may be due to the expanded CD8$^+$T$_{CM}$ cell population. Indeed, CD8$^+$T$_{CM}$ cells are long-lived[45]. Since AICD implies apoptosis of the T$_{mem/eff}$ cell population, we studied the effect of T cell *Abca1/Abcg1* deficiency on Bcl2 expression in CD8$^+$T$_{CM}$ and T$_{mem/eff}$ cells separately. Upon anti-CD3/IL-2 stimulation, ~90% of CD8$^+$T$_{CM}$ cells were Bcl2$^+$ in mice with T cell *Abca1/Abcg1* deficiency and controls (Supplementary Fig. 13d, e). T cell *Abca1/Abcg1* deficiency decreased anti-apoptotic Bcl2$^+$CD8$^+$T$_{mem/eff}$ cells by ~30%, while increasing pro-apoptotic Bcl2$^-$CD8$^+$T$_{mem/eff}$ cells by ~40% (Fig. 3k, l). Once the Bcl2 pathway is activated, mitochondrial ROS (mitoROS) stimulates cytochrome c release from mitochondria, which ultimately leads to caspase activation[46]. Indeed, upon αCD3/IL-2 stimulation, T cell *Abca1/Abcg1* deficiency increased mitoROS levels, mainly in CD8$^+$T$_{naive}$ and T$_{mem/eff}$ cells (Supplementary Fig. 13f–i). There was no effect on CD4$^+$ T cells (Supplementary Fig. 13f–i), perhaps because T cells with high mitoROS had already undergone apoptosis. Collectively, these data indicate that T cell *Abca1/Abcg1* deficiency enhances T cell apoptosis, independent of *Ldlr* expression, and downstream of TCR and CD25, mediated by Bcl2 in the intrinsic apoptosis pathway, especially in T$_{mem/eff}$ cells. This likely contributed to the ~50% decrease in peripheral T cells in T cell *Abca1/Abcg1* deficiency.

We then studied effects of T cell *Abca1/Abcg1* deficiency on T cell functionality. Upon stimulation by αCD3/IL-2, T cell *Abca1/Abcg1* deficiency increased CD8$^+$granzyme B$^+$ T cells as a percentage of total CD8$^+$ T cells (Fig. 4a, b and Supplementary Fig. 14a), and CD4$^+$IFN-γ$^+$ and CD8$^+$IFN-γ$^+$ T cells as percentages of CD4$^+$ and CD8$^+$ T cells, respectively (Fig. 4c, d and Supplementary Fig. 14a). T cell *Abca1/Abcg1* deficiency also

increased lysosomal-associated membrane protein 1 (LAMP-1) surface expression on CD8$^+$ T cells (Fig. 4e, f and Supplementary Fig. 14b), indicating T cell degranulation. These findings are suggestive of T cell *Abca1/Abcg1* deficiency increasing the TCR response, and increasing T cell functionality. However, in the same assay, T cell *Abca1/Abcg1* deficiency decreased IFN-γ secretion into the media (Fig. 4g), presumably due to increased T cell apoptosis. Indeed, *Abca1/Abcg1* deficiency decreased T cell counts by ~80% after stimulation by αCD3/IL-2 (Supplementary Fig. 14c), while the number of T cells at the start of the assay was similar between genotypes. Therefore, even though T cell *Abca1/Abcg1* deficiency increased the percentage of IFN-γ$^+$ T cells, it decreased IFN-γ secretion presumably due to increased T cell apoptosis. We subsequently investigated T cell mediated effects on macrophage function.

T cell IFN-γ secretion enhances the capacity of macrophages to kill bacteria[47]. To examine whether this was affected by T cell *Abca1/Abcg1* deficiency, we co-incubated wild-type bone marrow derived macrophages (BMDMs) with conditioned medium from αCD3/IL-2 stimulated *Ldlr*$^{-/-}$ or *T-Abc*$^{dko}$*Ldlr*$^{-/-}$ CD8$^+$ T cells prior to infection with *E.coli* bacteria. Conditioned medium from *T-Abc*$^{dko}$*Ldlr*$^{-/-}$ CD8$^+$ T cells increased the number of *E.coli* bacteria colony forming units (CFU) compared to medium from *Ldlr*$^{-/-}$ CD8$^+$ T cells (Fig. 4h), reflecting reduced bacterial killing capacity by macrophages.

We then examined effects of *Abca1/Abcg1* deficiency on T cell mediated cytotoxicity by co-incubating αCD3/IL-2 stimulated *Ldlr*$^{-/-}$ or *T-Abc*$^{dko}$*Ldlr*$^{-/-}$ T cells with wild-type BMDMs. Lactate dehydrogenase (LDH), a cytosolic enzyme released into the medium upon damage of the plasma membrane, reflects cell death[40]. *T-Abc*$^{dko}$*Ldlr*$^{-/-}$ T cells induced less BMDM LDH release compared to *Ldlr*$^{-/-}$ T cells (Fig. 4i), indicating less macrophage killing and decreased T cell mediated cytotoxicity.

Therefore, despite increasing IFN-γ$^+$ T cells and CD8$^+$granzyme B$^+$ T cells, T cell *Abca1/Abcg1* deficiency decreased the capacity of macrophages to kill bacteria as well as T cell mediated macrophage killing, presumably due to increased T cell apoptosis. These findings indicate that T cell *Abca1/Abcg1* deficiency decreases T cell functionality.

**Aging increases cholesterol accumulation and apoptosis in T cells.** We then asked whether there would be a broader physiological relevance of the phenotype we observed in T cell *Abca1/Abcg1* deficiency. Aged humans (~73–76 years old) show increased T cell plasma membrane cholesterol accumulation compared to young humans (~22–23 years old)[10,11] and a decrease in peripheral T cells[13]. The phenotype in T cells from aged individuals resembles the phenotype of mice with T cell *Abca1/Abcg1* deficiency and may suggest a broader physiological relevance for T cell plasma membrane cholesterol accumulation in regulating peripheral T cell numbers. Using T cells from aged (~24 months old) and young (~3 months old) mice, we investigated this further.

Similar to findings in aged humans[10,11], blood CD4$^+$ and CD8$^+$ T cells from aged mice show an increase in filipin staining compared to T cells from young mice, reflecting an increase in plasma membrane cholesterol accumulation (Fig. 5a, b and Supplementary Fig. 1e). Aging also increased choleratoxin B staining on CD8$^+$ T cells, suggestive of more lipid rafts (Fig. 5c, d and Supplementary Fig. 2c). Further, aging increased total T cell cholesterol content as assessed by GC-MS by ~1.5-fold (Fig. 5e), while there were no signs of lipid droplets based on BODIPY 493/503 staining (Supplementary Fig. 15a), indicating that the increase in cellular cholesterol reflects an increase in free cholesterol. Aging decreases *Abca1* and *Abcg1* mRNA expression

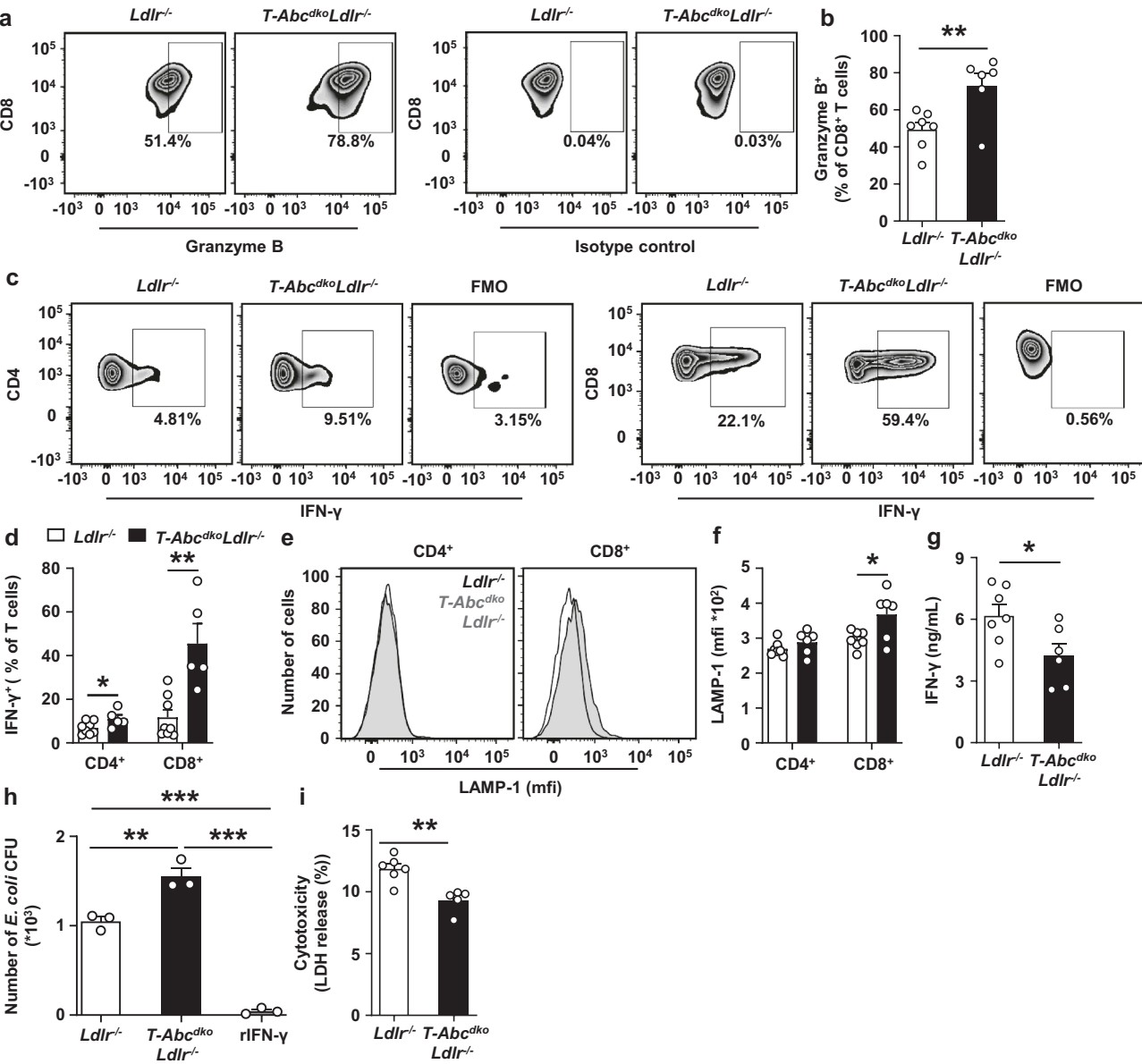

**Fig. 4 Effects of T cell *Abca1/Abcg1* deficiency on T cell functionality.** Splenic CD4[+] **a–f**, CD8[+] (**a–f**, **j**), or total T cells (**g–i**) were isolated from *Ldlr*[−/−] and *T-Abc*[dko]*Ldlr*[−/−] mice. **a–f** CD4[+] and CD8[+] T cells were stimulated with αCD3/IL-2 for 12 h, fixed, permeabilized, stained for granzyme B (**a**, **b**), or interferon gamma (IFN-γ) (**c**, **d**), or alternatively, stained for lysosomal-associated membrane protein-1 (LAMP-1) (**e**, **f**). Gating strategies are shown in Supplementary Fig. 14. Representative flow cytometry plots of (**a**) CD8[+]granzyme B[+] (*p* = 0.0096), (**c**) CD4[+]IFN-γ[+] (*p* = 0.048) and CD8[+]IFN-γ[+] (*p* = 0.002) T cells and (**b**, **d**) quantification as a percentage of CD8[+] T cells after correction for isotype control (**b**) and as a percentage of CD4[+] or CD8[+] T cells after correction for fluorescence minus one (FMO) control (**d**). **e** Representative flow cytometry plots and (**f**) quantification of LAMP-1 expression on CD4[+] and CD8[+] (*p* = 0.027) T cells. **b**, **f** *n* = 7 *Ldlr*[−/−] and *n* = 6 *T-Abc*[dko]*Ldlr*[−/−] mice. **d** *n* = 8 *Ldlr*[−/−] and *n* = 5 *T-Abc*[dko]*Ldlr*[−/−] mice. **g** T cells were stimulated with αCD3/IL-2 for 12 h. Medium was collected and IFN-γ levels (*p* = 0.034) were measured by ELISA. *n* = 7 *Ldlr*[−/−] and *n* = 6 *T-Abc*[dko]*Ldlr*[−/−] mice. **h** Wild-type bone marrow derived macrophages (BMDMs) were incubated with conditioned medium from CD8[+] T cells stimulated with αCD3/IL-2 for 12 h, or recombinant IFN-γ (rIFN-γ; positive control) for 3 days. After medium removal, BMDMs were co-incubated with *E. coli* for 1 h, washed, incubated with gentamicin for 3 h, washed, and lysed. BMDM lysates were plated on agar overnight and *E. coli* colony forming units (CFU) were counted (*Ldlr*[−/−] vs *T-Abc*[dko]*Ldlr*[−/−], *p* = 0.0032; *Ldlr*[−/−] or *T-Abc*[dko]*Ldlr*[−/−] vs rIFN-γ, *p* < 0.0001). *n* = 3 *Ldlr*[−/−] and *n* = 3 *T-Abc*[dko]*Ldlr*[−/−] mice. **i** BMDMs were co-incubated with T cells in the presence of αCD3/IL-2 (to stimulate T cells) for 24 h. Medium was collected and endogenous lactate dehydrogenase (LDH) was assessed (*p* = 0.0023). *n* = 6 *Ldlr*[−/−] and *n* = 5 *T-Abc*[dko]*Ldlr*[−/−] mice. For all panels, error bars represent SEM. Biologically independent samples were included. *p* value was determined by unpaired two-tailed Student's *t* test (**a–i**) or one-way ANOVA with Bonferroni post-test (**j**). **p* < 0.05, ***p* < 0.01, ****p* < 0.001. Source data are provided as a Source Data file.

in mouse splenic and peritoneal macrophages[48]; however, we found no effect of aging on mRNA expression of *Abca1* and *Abcg1* in T cells, while mRNA expression of other genes affecting cholesterol metabolism such as *Hmcgr* and *Hmgcs* was minimally decreased and *Ldlr* mRNA expression showed a moderate

decrease, especially in CD8[+] T cells (Supplementary Fig. 15b, c). *Ldlr* expression does not affect T cell cholesterol accumulation (Supplementary Fig. 2a, b and Supplementary Fig. 3a, b). Therefore, effects of aging on T cell cholesterol accumulation cannot be explained by changes in expression of these genes.

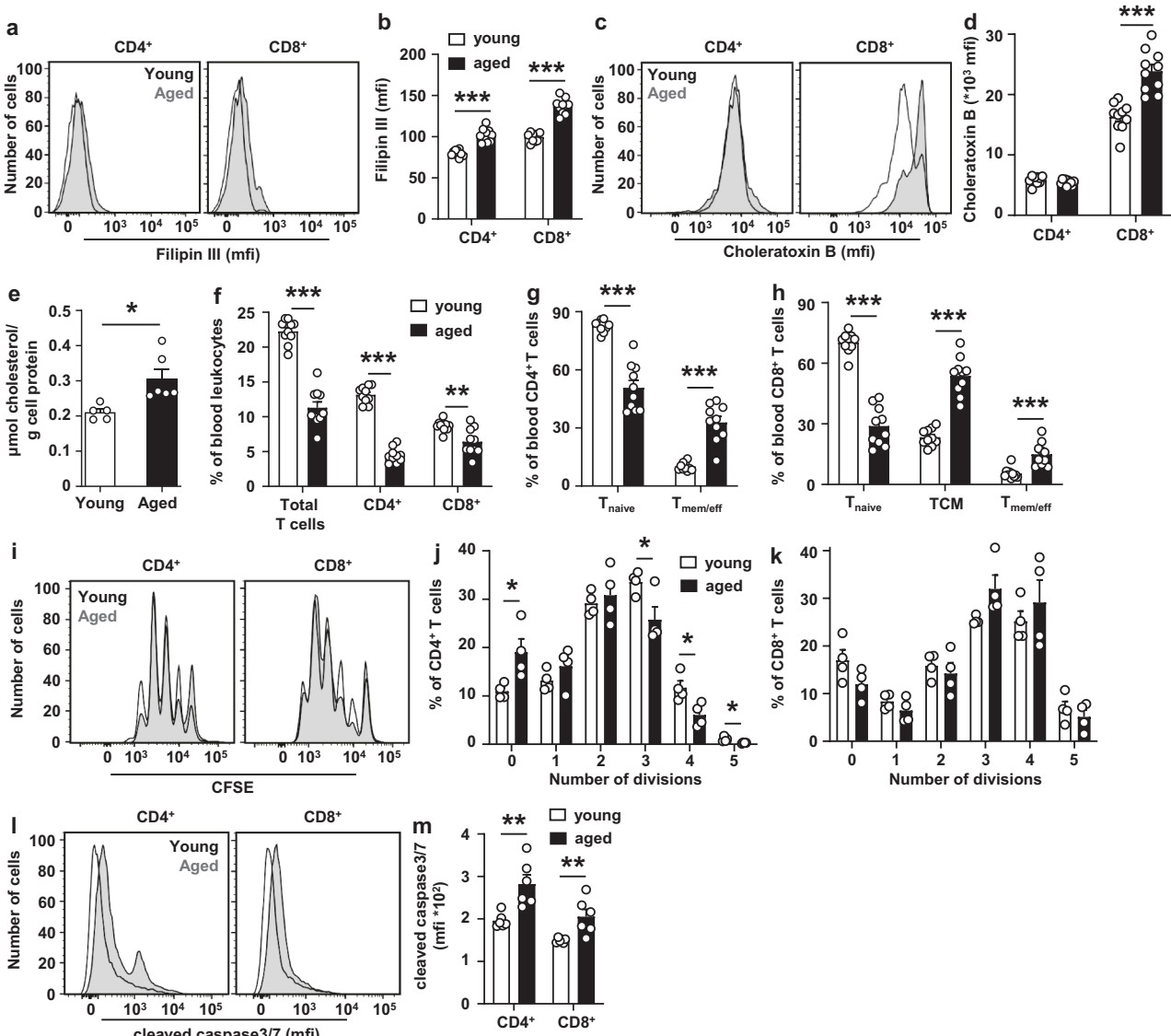

**Fig. 5 Aging increases cholesterol accumulation and apoptosis in T cells.** Blood and spleens from young (3 months) and aged (24 months) wild-type mice fed a chow diet were collected. Filipin (**a**, **b**) and choleratoxin B (**c**, **d**) stainings on blood T cells. **a** Representative flow cytometry plots of filipin staining on CD4$^+$ ($p < 0.000001$) and CD8$^+$ ($p < 0.000001$) T cells gated as in Supplementary Fig. 1e and (**b**) quantification. **c** Representative flow cytometry plots of choleratoxin B staining on CD4$^+$ and CD8$^+$ ($p = 0.00002$) T cells gated as in Supplementary Fig. 2c and (**d**) quantification. **b**, **d** $n = 10$ young and $n = 10$ aged wild-type mice. **e** Splenic T cells were isolated and total cholesterol ($p = 0.013$) was measured by Gas Chromatography – Mass Spectrometry. $n = 5$ young and $n = 6$ aged wild-type mice. **f** Total ($p < 0.000001$), CD4$^+$ ($p < 0.000001$), and CD8$^+$ ($p = 0.0026$) T cells as a percentage of total blood leukocytes. T$_{naive}$ ($p < 0.000001$) and T$_{mem/eff}$ ($p < 0.000001$) cells as a percentage of CD4$^+$ (**g**) and T$_{naive}$ ($p < 0.000001$), T$_{mem/eff}$ ($p < 0.000001$), and T$_{CM}$ ($p = 0·00016$) cells as a percentage of CD8$^+$ T cells (**h**) in blood. **f**–**h** $n = 10$ young and $n = 10$ aged wild-type mice. **i**–**m** Splenic total, CD4$^+$, and CD8$^+$ T cells were isolated. **i**–**k** T cells were labeled with CFSE and stimulated with αCD3/αCD28 beads. CFSE dilution was measured by flow cytometry at 72 h after stimulation. **i** Representative CFSE dilutions gated as in Supplementary Fig. 11c and quantification of the number of divisions of (**j**) CD4$^+$ (division 0, $p = 0.030$; 3, $p = 0.035$; 4, $p = 0.023$; 5, $p = 0.026$) and (**k**) CD8$^+$ T cells. $n = 4$ young and $n = 4$ aged wild-type mice. **l**–**m** CD4$^+$ and CD8$^+$ T cells were stimulated with αCD3/IL-2 for 12 h with concomitant staining for cleaved caspase 3/7. **l** Representative flow cytometry plots of cleaved caspase 3/7 in CD4$^+$ ($p = 0.004$) and CD8$^+$ ($p = 0.009$) T cells gated as in Supplementary Fig. 12d and (**m**) quantification. $n = 6$ young and $n = 6$ aged wild-type mice. For all panels, error bars represent SEM. Biologically independent samples were included. $p$ value was determined by unpaired two-tailed Student's $t$ test. *$p < 0.05$, **$p < 0.01$, ***$p < 0.001$. Source data are provided as a Source Data file.

Perhaps repeated TCR stimulation increases cholesterol accumulation during aging.

Aging did not affect plasma cholesterol, or its distribution over lipoproteins, but, in line with previous observations[49], decreased plasma triglycerides (TG) by ~42% ($P < 0.001$), reflected by a decrease in the very low-density lipoprotein (VLDL)-TG fraction (Supplementary Table 1 and Supplementary Fig. 15d, e). Similar to previous observations from aged humans and mice[12,50,51], aged

mice showed a decrease in peripheral T cells compared to young mice (Fig. 5f), mainly due to decreased T$_{naive}$ cells, while T$_{mem/eff}$ and T$_{CM}$ cells were increased (Fig. 5g, h). CD4$^+$ T cells from aged mice showed a slight decrease in T cell proliferation in response to TCR stimulation, while CD8$^+$ T cell proliferation was not affected by aging (Fig. 5i–k and Supplementary Fig. 11c), in line with previous findings showing that T$_{naive}$ cells from aged mice still proliferate efficiently[12]. We then studied apoptosis in

response to αCD3/IL-2 stimulation in young and aged T cells. αCD3/IL-2 stimulation increased cleaved caspase3/7 in aged compared to young T cells (Fig. 5l, m and Supplementary Fig. 12d), reflecting increased apoptosis. The data in aged mice (Fig. 5a–h, l, m) together with our findings in $T\text{-}Abc^{dko}Ldlr^{-/-}$ mice suggest that plasma membrane cholesterol accumulation in T cells from aged mice increases T cell activation and apoptosis, which likely contributes to the decrease in T cells during aging.

Further, upon stimulation by αCD3/IL-2, aging increased CD8[+]granzyme B[+]cells as percentage of CD8[+] T cells (Supplementary Fig. 14a and Supplementary Fig. 16a, b) and CD4[+]IFN-γ[+] and CD8[+]IFN-γ[+] cells as percentages of CD4[+] and CD8[+] T cells, respectively (Supplementary Fig. 14a and Supplementary Fig. 16c, d). Aging also increased LAMP-1 surface expression on CD4[+] and CD8[+] T cells (Supplementary Fig. 14b and Supplementary Fig. 16e, f), indicating increased T cell degranulation. These findings suggest that aging increased the TCR response and enhanced T cell functionality. Consistently, aging increased IFN-γ secretion from T cells (Supplementary Fig. 16g), in line with previous studies[52,53], but different from mice with T cell $Abca1/Abcg1$ deficiency. The latter was presumably due to the more pronounced increase of T cell apoptosis upon T cell $Abca1/Abcg1$ deficiency than upon T cell aging. Still, the immune system loses its ability to mount an effective immune response to pathogens during aging[54], and our findings suggest that increased T cell apoptosis may contribute.

**T cell $Abca1/Abcg1$ deficiency induces a premature T cell aging phenotype**. We then asked whether aging would affect TCR responses in $T\text{-}Abc^{dko}Ldlr^{-/-}$ mice. We aged $T\text{-}Abc^{dko}Ldlr^{-/-}$ mice and $Ldlr^{-/-}$ littermate controls until 12–13 months (middle-aged). The phenotype in terms of T cell numbers and activation did not differ between middle-aged and young $T\text{-}Abc^{dko}Ldlr^{-/-}$ mice (Supplementary Fig. 17a–c). T cell $Abca1/Abcg1$ deficiency increased cleaved caspase3/7 upon TCR stimulation in middle-aged mice, reflecting increased apoptosis (Fig. 6a, b and Supplementary Fig. 12d). Strikingly, upon TCR stimulation, T cell $Abca1/Abcg1$ deficiency almost completely abolished CD4[+] and CD8[+] T cell proliferation, while T cells from control mice still proliferated, in the absence (Fig. 6c–e and Supplementary Fig. 11c) and presence of $Ldlr$ expression (Fig. 6f–h and Supplementary Fig. 11c). Moreover, treatment with reconstituted HDL (rHDL), which promotes cholesterol efflux in the absence of $Abca1/Abcg1$ via passive efflux mechanisms, at least in macrophages[55], for the whole duration of the proliferation assay (72 h), still further decreased T cell proliferation in control and $Abca1/Abcg1$-deficient T cells (Supplementary Fig. 17d–h). These data indicate that even when proliferation is almost impaired in middle-aged T cells with $Abca1/Abcg1$ deficiency, cholesterol depletion still suppresses T cell proliferation. The remaining T cell proliferation is thus cholesterol-dependent. T cells with $Abca1/Abcg1$ deficiency may not enter the cell cycle due to upregulation of cyclin dependent kinase inhibitors, including Cdkn1a[56]. $Abca1/Abcg1$ deficiency increased $Cdkn1a$ mRNA expression by 6-fold in CD4[+] T cells and 2.5-fold in CD8[+] T cells compared to middle-aged control T cells (Fig. 6i, j), while $Cdkn2a$ mRNA expression was not affected (Supplementary Fig. 17i, j). Of note, unlike in other cell types, in T cells, $Cdkn2a$ rather reflects increased T cell activation than aberrant cell cycling[13]. When comparing $Cdkn1a$ mRNA expression in T cells from middle-aged $T\text{-}Abc^{dko}Ldlr^{-/-}$ mice to T cells from aged wild-type mice, we observed that $Abca1/Abcg1$ deficiency increased $Cdkn1a$ mRNA expression by ~6-fold in CD4[+] and CD8[+] T cells (Fig. 6i, j). Together, these data indicate that middle-aged $Abca1/Abcg1$-deficient T cells upregulate $Cdkn1a$ and do not enter the cell cycle, which is suggestive of T cell senescence and premature T cell aging.

**Effects of T cell $Abca1/Abcg1$ deficiency on T cell subsets and atherosclerosis**. Previous studies have shown that T cell $Abcg1$ deficiency decreases atherosclerosis by enhancing the formation of $T_{reg}$ cells in thymus and LNs of $Ldlr^{-/-}$ mice fed WTD[22]. In addition, WTD feeding enhances the conversion of $T_{reg}$ cells into $T_{FH}$ and $T_h1$ cells, which was dependent on cholesterol accumulation[5]. T cell $Acat1$ deficiency induces plasma membrane cholesterol accumulation in CD8[+] T cells, leading to expansion of CD8[+]IFN-γ[+] T cells[8] that are pro-atherogenic. We thus investigated whether exacerbated cholesterol accumulation due to loss of $Abca1/Abcg1$ in T cells would affect T cell subsets.

We assessed these T cell subsets in para-aortic LNs of $T\text{-}Abc^{dko}Ldlr^{-/-}$ and $Ldlr^{-/-}$ mice fed a chow diet. T cell $Abca1/Abcg1$ deficiency did not affect para-aortic LN CD25[+]Foxp3[+]$T_{reg}$ cells, but increased CD25[−]Foxp3[+]$T_{reg}$ cells as a percentage of CD4[+] T cells (Fig. 7a, b and Supplementary Fig. 18a), which may suggest that upon cholesterol accumulation, $T_{reg}$ cells lose their CD25 expression, and start to express markers of $T_{FH}$ and $T_h1$ cells as has been reported in $Apoe^{-/-}$ mice fed WTD[5]. Both in $T\text{-}Abc^{dko}Ldlr^{-/-}$ and $Ldlr^{-/-}$ mice, CD25[−]Foxp3[+]T cells were mainly of the activated $T_{mem/eff}$ (CD44[+]CD62L[−]) phenotype, while CD25[+]Foxp3[+]cells were mostly of the $T_{naive}$ (CD44[−]CD62L[+]) phenotype (Supplementary Fig. 18b, c). After correction for total para-aortic LN cell number, T cell $Abca1/Abcg1$ deficiency decreased CD25[+]Foxp3[+] $T_{reg}$ cells by ~75% in whole para-aortic LNs, while not affecting CD25[−]Foxp3[+] $T_{reg}$ cells (Fig. 7c), consistent with the decrease in CD25[+]Foxp3[+] $T_{reg}$ cells that we observed in vitro (Fig. 3e, f), presumably because there were less total para-aortic LN T cells (Fig. 2g) due to increased T cell apoptosis. T cell $Abca1/Abcg1$ deficiency increased $T_{FH}$ cells as a percentage of CD4[+] $T_{mem/eff}$ cells in para-aortic LNs (Fig. 7d, e and Supplementary Fig. 18d), but did not affect $T_{FH}$ cells after correction for total para-aortic LN cell number (Fig. 7f), presumably also due to the decrease in total para-aortic LN T cells (Fig. 2g).

Plasma membrane cholesterol accumulation increases formation of $T_h1$ cells that express the transcription factor Tbet[5,8]. The percentage of CD4[+]Tbet[+] and CD8[+]Tbet[+] cells was increased in para-aortic LNs of $T\text{-}Abc^{dko}Ldlr^{-/-}$ mice compared to $Ldlr^{-/-}$ controls (Fig. 7g, h and Supplementary Fig. 18e), but not affected after correction for total para-aortic LN cell numbers for CD4[+]Tbet[+] cells, while CD8[+]Tbet[+] cells were still ~2-fold increased (Fig. 7i). While CD4[+]Tbet[+] cells were mainly of the $T_{mem/eff}$ (CD44[+]CD62L[−]) phenotype (Supplementary Fig. 18f), CD8[+]Tbet[+] cells were mostly of the long-lived $T_{CM}$ (CD44[+]CD62L[+]) phenotype (Supplementary Fig. 18g). In sum, based on total para-aortic LN cell numbers, T cell $Abca1/Abcg1$ deficiency decreased CD25[+]Foxp3[+]$T_{reg}$ cells and increased CD8[+]Tbet[+] cells in para-aortic LNs, while para-aortic LN CD25[−]Foxp3[+]$T_{reg}$ cells, $T_{FH}$, and $T_h1$ cells were not affected (Supplementary Table 2). We also investigated whether these changes in T cells affected myeloid cells in blood. When expressed as a percentage of total leukocytes, T cell $Abca1/Abcg1$ deficiency increased blood monocytes, Ly6C[lo] and Ly6C[hi] monocyte subsets, as well as neutrophils by ~50% (Supplementary Fig. 19a, b). We explain these increases by T cell $Abca1/Abcg1$ deficiency decreasing T cells by 50% (Fig. 2a, b) and therefore other leukocyte populations, including myeloid cells showing an increase as percentage of total leukocytes. Indeed, the absolute number of blood leukocytes was decreased by T cell $Abca1/Abcg1$ deficiency (Supplementary Fig. 19c), presumably due to the decrease in blood T cells (Fig. 2a–c), and therefore, when corrected for total blood leukocyte number, T cell $Abca1/Abcg1$ deficiency no longer affected levels of myeloid cell counts in blood (Supplementary Fig. 19d).

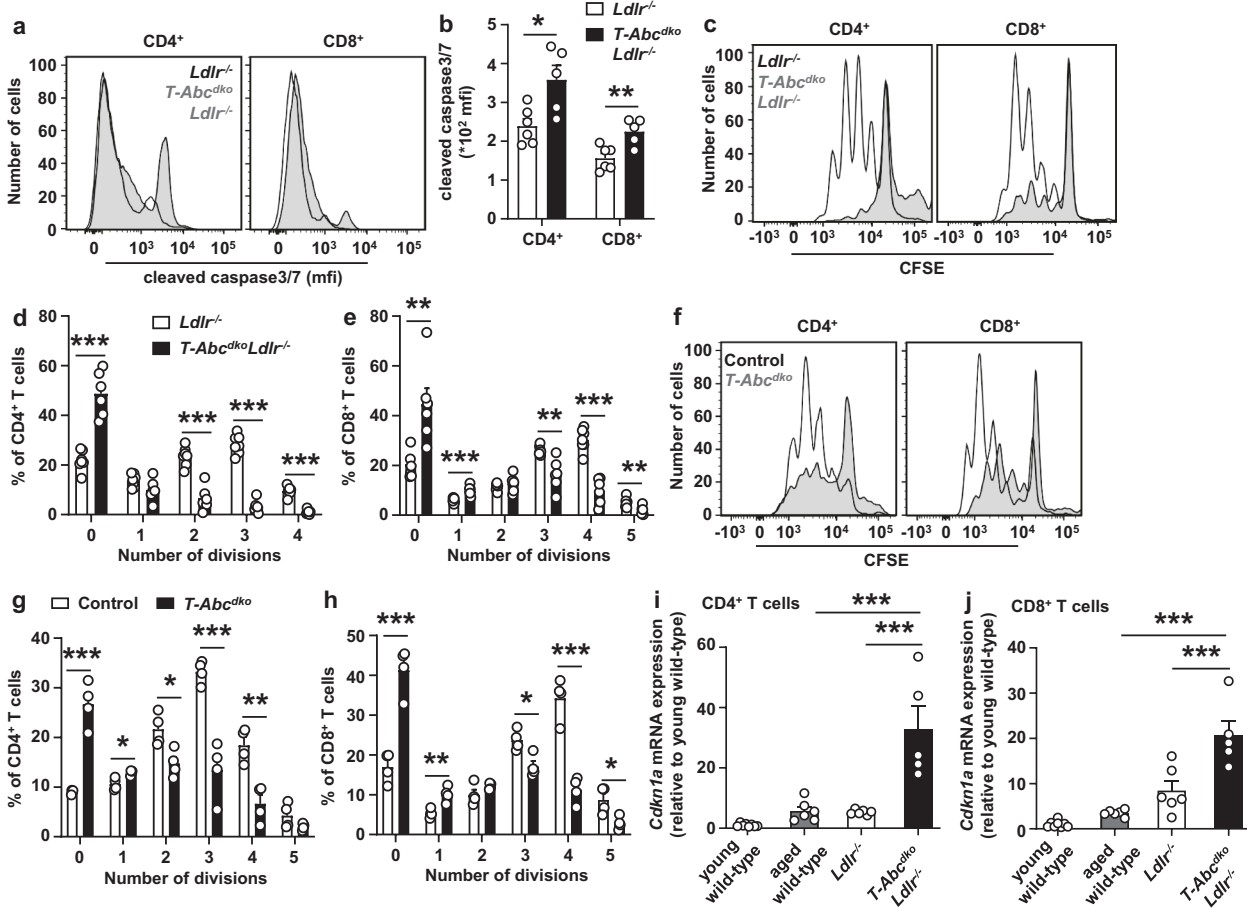

**Fig. 6 T cell *Abca1/Abcg1* deficiency induces a premature T cell aging phenotype.** *Ldlr*[−/−] and *T-Abc*[dko]*Ldlr*[−/−] mice (**a–e**, **i–j**) and control (*Ldlr*[+/+]) and *T-Abc*[dko] mice (**f–h**) were fed a chow diet for 12–13 months, and wild-type mice for 3 months (young) or 24 months (aged) (**i–j**). **a–j** Spleens were collected and CD4+ and CD8+ T cells were isolated. **a**, **b** CD4+ and CD8+ T cells were stimulated with αCD3/IL-2 for 12 h with concomitant staining for cleaved caspase3/7. **a** Representative flow cytometry plots of cleaved caspase 3/7 in CD4+ (p = 0.015) and CD8+ (p = 0.007) T cells gated as in Supplementary Fig. 12d and (**b**) quantification. n = 6 *Ldlr*[−/−] and n = 5 *T-Abc*[dko]*Ldlr*[−/−] mice. **c–h** T cells were labeled with CFSE and stimulated with αCD3/αCD28 beads. CFSE dilution was measured by flow cytometry at 72 h after stimulation. **c**, **f** Representative CFSE dilutions gated as in Supplementary Fig. 11c. The number of divisions were quantified for (**d**) CD4+ (division 0, p = 0.000017; 2, p = 0.00002; 3, p < 0.000001; 4, p < 0.000001) and (**e**) CD8+ (division 0, p = 0.0029; 1, p = 0.00095; 3, p = 0.0023; 4, p = 0.0000097; 5, p = 0.008) T cells. n = 7 *Ldlr*[−/−] and n = 6 *T-Abc*[dko]*Ldlr*[−/−] mice. The number of divisions were quantified for (**g**) CD4+ (division 0), p = 0.0002; 1, p = 0.013; 2, p = 0.017; 3, p = 0.00067; 4, p = 0.0031 and (**h**) CD8+ T cells (division 0, p = 0.0004; 1, p = 0.0056; 3, p = 0.012; 4, p = 0.0003; 5, p = 0.02). n = 4 control and n = 4 *T-Abc*[dko] mice. **i–j** T cells were isolated and RNA was extracted. Cyclin dependent kinase inhibitor 1a (*Cdkn1a*) mRNA expression was measured in (**i**) CD4+ (*Ldlr*[−/−] or aged wild-type vs *T-Abc*[dko]*Ldlr*[−/−], p < 0.0001) and (**j**) CD8+ (*Ldlr*[−/−] vs *T-Abc*[dko]*Ldlr*[−/−], p=0.0003; aged wild-type vs *T-Abc*[dko]*Ldlr*[−/−], p < 0.0001) T cells by qPCR and shown as fold change compared to young wild-type mice. n = 8 young wild-type, n = 6 aged wild-type, n = 6 *Ldlr*[−/−], and n = 5 *T-Abc*[dko]*Ldlr*[−/−] mice. For all panels, error bars represent SEM. Biologically independent samples were included. p value was determined by unpaired two-tailed Student's t test (**b**, **d–e**, **g**, **h**) or one-way ANOVA with Bonferroni post-test (**i**, **j**). *p < 0.05, **p < 0.01, ***p < 0.001. Source data are provided as a Source Data file.

We then investigated the effects of these changes on atherosclerosis. To induce atherogenesis, female *T-Abc*[dko]*Ldlr*[−/−] mice and *Ldlr*[−/−] littermate controls were fed a WTD. Findings on T cell activation and T cell subsets as well as on percentage of blood myeloid cells were similar to mice fed chow diet (Supplementary Fig. 20a–k). However, the increase in total monocytes, even though only ~15%, remained significant after correction for total blood leukocyte numbers in *Ldlr*[−/−] mice with T cell *Abca1/Abcg1* deficiency compared to controls (Supplementary Fig. 20l). After 10 weeks of WTD, we assessed atherosclerotic lesion size at the level of the aortic root. T cell *Abca1/Abcg1* deficiency did not affect atherosclerotic lesion size (Fig. 7j). This was accompanied by a decrease in plasma total cholesterol levels of ~15%, which reflects decreased VLDL and LDL cholesterol, while plasma TG or VLDL-TG was not affected (Supplementary Table 1 and Supplementary Fig. 21a, b). The decrease in plasma VLDL/LDL-cholesterol was

likely a consequence of increased total cholesterol accumulation in T cells with *Abca1/Abcg1* deficiency. Similarly, previous studies have shown that hematopoietic *Abca1/Abcg1* deficiency decreased plasma VLDL/LDL-cholesterol levels in *Ldlr*[−/−] mice fed WTD[57]. However, on a chow diet, hematopoietic *Abca1/Abcg1* deficiency, while still inducing cholesterol accumulation in hematopoietic cells, did not affect plasma VLDL/LDL-cholesterol levels[55,58], presumably because the clearance of VLDL/LDL particles by hepatic Ldlr is not as much of a limiting factor as in *Ldlr*[−/−] mice fed WTD. In an attempt to exclude the confounding factor of decreased VLDL/LDL cholesterol levels to atherogenesis, we fed mice a chow diet for 28 weeks, similar to a study we carried out previously[55]. Chow diet-fed *T-Abc*[dko]*Ldlr*[−/−] and *Ldlr*[−/−] mice had similar plasma total cholesterol and TG levels, as well as distribution of these lipids over VLDL, LDL, or HDL (Supplementary Table 1 and Supplementary Fig. 21c, d) and developed

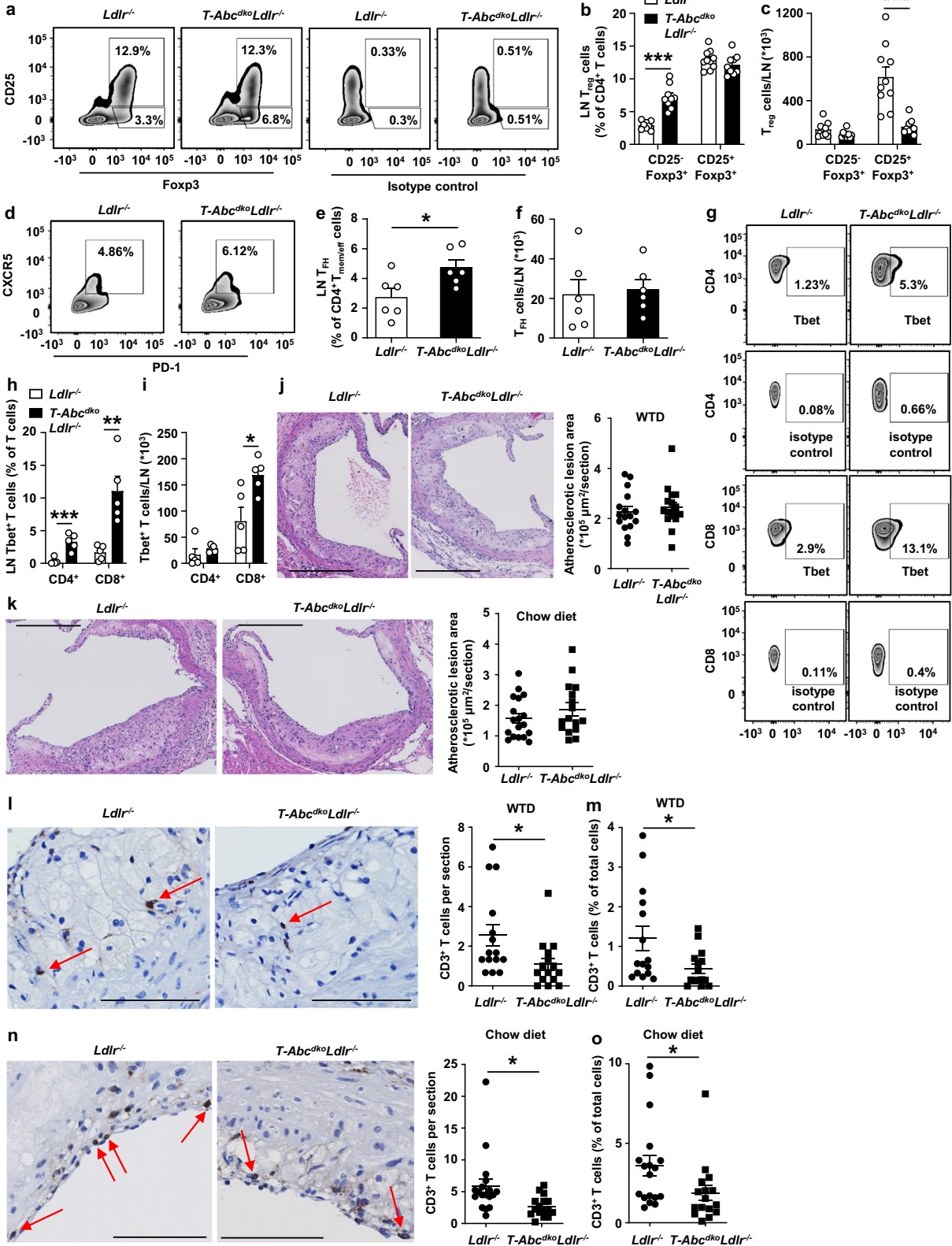

atherosclerotic lesions similar in size compared to WTD-fed $Ldlr^{-/-}$ mice, but there was no difference in atherosclerotic lesion size between the genotypes (Fig. 7k). In line with observations in blood and secondary lymphoid organs, $T\text{-}Abc^{dko}Ldlr^{-/-}$ mice showed a ~50% decrease in plaque CD3$^+$ T cells on both diets compared to $Ldlr^{-/-}$ controls, both when shown as total CD3$^+$

T cells per section or as percentage CD3$^+$ T cells of total cells (nuclei) in lesions (Fig. 7l–o). Further, characterization of atherosclerotic lesions revealed no changes in fibrous cap thickness, collagen content, smooth muscle actin (α-SMA) or galectin-3 (Lgals3 or Mac-2) staining, reflecting predominantly macrophages, after either WTD or chow diet (Supplementary

**Fig. 7 Effects of T cell *Abca1/Abcg1* deficiency on para-aortic LN T cell subsets and atherosclerosis in *Ldlr*$^{-/-}$ mice.** Female *Ldlr*$^{-/-}$ and *T-Abc*$^{dko}$*Ldlr*$^{-/-}$ mice were fed chow diet for 28 weeks (**a–i, k, n–o**) or WTD for 10 weeks (**j, l–m**). **a–i** Para-aortic LNs were isolated, stained with the indicated antibodies, and analyzed by flow cytometry. Gating strategies are shown in Supplementary Fig. 18. **a** Representative flow cytometry plots and (**b**) quantification of CD25$^-$Foxp3$^+$ ($p = 0.000001$) and CD25$^+$Foxp3$^+$ T$_{reg}$ cells as a percentage of CD4$^+$ T cells after correction for isotype control. **c** CD25$^-$Foxp3$^+$ and CD25$^+$Foxp3$^+$ T$_{reg}$ cells ($p = 0.00026$) as cells/LN after correction for total LN cells. $n = 10$ *Ldlr*$^{-/-}$ and $n = 9$ *T-Abc*$^{dko}$*Ldlr*$^{-/-}$ mice. **d** Representative flow cytometry plots and (**e**) quantification of CD4$^+$CD44$^+$CD62L$^-$ C-X-C chemokine receptor type 5 (CXCR5)$^+$ programmed cell death 1 (PD-1)$^+$ T$_{follicular\ helper}$ (T$_{FH}$) cells ($p = 0.023$) as a percentage of CD4$^+$ T$_{mem/eff}$ cells, and (**f**) as cells/LN. $n = 6$ *Ldlr*$^{-/-}$ and $n = 6$ *T-Abc*$^{dko}$*Ldlr*$^{-/-}$ mice. **g** Representative flow cytometry plots and (**h**) quantification of CD4$^+$Tbet$^+$ ($p = 0.00099$) and CD8$^+$Tbet$^+$ ($p = 0.0036$) T cells as percentages of CD4$^+$ and CD8$^+$ T cells after correction for isotype control. **i** CD4$^+$Tbet$^+$ and CD8$^+$Tbet$^+$ ($p = 0.0188$) T cells as cells/LN. $n = 5$ *Ldlr*$^{-/-}$ and $n = 5$ *T-Abc*$^{dko}$*Ldlr*$^{-/-}$ mice. Hearts were isolated and paraffin sections of the aortic root were stained for (**j, k**) haematoxylin-eosin (H&E) or (**l–o**) CD3. Representative pictures of H&E staining and quantification of atherosclerotic lesion area for WTD-fed (**j**) or chow diet-fed (**k**) mice. Scale bar represents 300 μm. Representative pictures and quantification of CD3$^+$ cells per section for WTD-fed ($p = 0.027$) (**l**) or chow diet-fed ($p = 0.0153$) (**n**) mice. CD3$^+$ cells are depicted by arrows. Scale bar represents 80 μm. CD3$^+$ T cells as percentage of total cells per section for WTD-fed ($p = 0.027$) (**m**) or chow diet-fed ($p = 0.043$) mice (**o**). **j** $n = 16$ *Ldlr*$^{-/-}$ and $n = 16$ *T-Abc*$^{dko}$*Ldlr*$^{-/-}$ mice. **k, n, o** $n = 18$ *Ldlr*$^{-/-}$ and $n = 16$ *T-Abc*$^{dko}$*Ldlr*$^{-/-}$ mice. **l–m** $n = 15$ *Ldlr*$^{-/-}$ and $n = 15$ *T-Abc*$^{dko}$*Ldlr*$^{-/-}$ mice. For all panels, error bars represent SEM. Biologically independent samples were included. $p$ value was determined by unpaired two-tailed Student's $t$ test. *$p < 0.05$, **$p < 0.01$, ***$p < 0.001$. Source data are provided as a Source Data file.

Fig. 22a–g and Supplementary Fig. 23a–g), suggesting no effects on plaque stability. In a separate cohort of mice, we evaluated lipid accumulation in atherosclerotic lesions in mice fed chow diet. After 38 weeks of chow diet (~9 months), T cell *Abca1/Abcg1* deficiency did not affect atherosclerotic lesion area, similar to observations at 28 weeks of chow diet (Fig. 7k), but increased Oil Red O area as a percentage of lesion area, reflecting increased lipid accumulation (Supplementary Fig. 22h–j), presumably in both T cells and macrophages.

Hence, despite decreasing T cells in atherosclerotic plaques and decreasing para-aortic LN CD25$^+$Foxp3$^+$ T$_{reg}$ cells, and increasing para-aortic LN CD8$^+$Tbet$^+$ cells (Supplementary Table 2), T cell *Abca1/Abcg1* deficiency did not affect atherosclerotic lesion size in *Ldlr*$^{-/-}$ mice fed chow diet or WTD. Since these subsets have both pro- and anti-inflammatory effects, and the loss of *Abca1/Abcg1* in T cells does not affect a single predominant T cell subset, this effect on atherosclerosis could be due to counter-regulatory inflammatory effects in the aorta during sterile inflammation.

**T cell *Abca1/Abcg1* deficiency decreases atherosclerosis in middle-aged *Ldlr*$^{-/-}$ mice.** We then investigated whether the premature T cell aging phenotype in middle-aged *T-Abc*$^{dko}$*Ldlr*$^{-/-}$ mice affected atherogenesis. T cell *Abca1/Abcg1* deficiency in middle-aged *Ldlr*$^{-/-}$ mice did not affect blood monocytes or neutrophils (Supplementary Fig. 24a, b). Similar to observations in young *T-Abc*$^{dko}$*Ldlr*$^{-/-}$ mice (Fig. 7a–c), T cell *Abca1/Abcg1* deficiency increased CD25$^-$Foxp3$^+$ T$_{reg}$ cells but not CD25$^+$Foxp3$^+$ T$_{reg}$ cells as a percentage of CD4$^+$ T cells (Supplementary Fig. 24c, d). T cell *Abca1/Abcg1* deficiency did not affect the number of total CD25$^-$Foxp3$^+$ T$_{reg}$ cells in whole para-aortic LNs, but decreased CD25$^+$Foxp3$^+$ T$_{reg}$ cells by ~70% (Supplementary Fig. 24e), similar to findings in young mice (Fig. 7c), and presumably due to the increased T cell apoptosis. Unlike in young mice (Supplementary Fig. 8h), T cell *Abca1/Abcg1* deficiency did not affect CD4$^+$ T$_{mem/eff}$ cells as a percentage of CD4$^+$ T cells (Supplementary Fig. 24f). Moreover, after correction for the total para-aortic LN cell number, T cell *Abca1/Abcg1* deficiency decreased this population by ~50% (Supplementary Fig. 24g). CD8$^+$Tbet$^+$ cells were increased both as a percentage and after correction for total para-aortic LN cell number in para-aortic LNs from middle-aged *T-Abc*$^{dko}$*Ldlr*$^{-/-}$ mice compared to controls, while CD4$^+$Tbet$^+$ cells were not affected (Supplementary Fig. 24h–j). Similar to CD25$^-$Foxp3$^+$ T$_{reg}$ cells and the observations in young mice (Fig. 7d–f), T cell *Abca1/Abcg1* deficiency increased T$_{FH}$

cells as a percentage of LN CD4$^+$ T$_{mem/eff}$ cells, but not after correction for total para-aortic LN cell number (Supplementary Fig. 24k–m). Under conditions of similar plasma total cholesterol and TG levels, and similar distribution of these lipids on VLDL, LDL, and HDL (Supplementary Table 1 and Supplementary Fig. 25a, b), T cell *Abca1/Abcg1* deficiency decreased atherosclerotic lesion size by ~35% (Fig. 8a), which was accompanied by a ~50% decrease in plaque CD3$^+$ T cells (Fig. 8b, c). While the extent of the decrease in plaque CD3$^+$ T cells was similar to our observations in young mice (Fig. 7l–o), the decrease in atherosclerotic lesion area was not, and also the atherosclerotic lesions of middle-aged mice showed a 2-fold higher T cell content than those of young mice fed chow diet (Fig. 7o), perhaps suggesting a more prominent effect of T cells on atherosclerosis in middle-aged *Ldlr*$^{-/-}$ mice. Also, the decrease in CD4$^+$ T$_{mem/eff}$ cells that we observed in middle-aged, but not young mice, may have contributed to the decrease in atherosclerotic lesion size.

To obtain more insights into the nature of the T cells that were affected by T cell *Abca1/Abcg1* deficiency in the aorta, we then further studied total aortic T cells based on mRNA expression. T cell *Abca1/Abcg1* deficiency did not affect mRNA expression of the T$_{reg}$ transcription factor *Foxp3* (Fig. 8d). Also, mRNA expression of anti-inflammatory *Il10* and *Tgfb1*, and pro-inflammatory *Tnf* in aortic T cells did not differ between the genotypes (Fig. 8d). In line with findings in splenic T cells (Fig. 6i, j), T cell *Abca1/Abcg1* deficiency moderately increased mRNA expression of the senescence marker *Cdkn1a* in aortic T cells, and decreased *Bcl2* mRNA by ~25% (Fig. 8d). However, T cell *Abca1/Abcg1* deficiency did not affect αCD3/αCD28 induced proliferation of CD4$^+$ T cells isolated from aortas (Fig. 8e), perhaps due to these cells not showing a high level of proliferation compared to our findings in splenic T cells (Fig. 6c–e). Nevertheless, similar to findings on splenic T cells (Fig. 6a, b), T cell *Abca1/Abcg1* deficiency also increased cleaved caspase 3/7 in T cells isolated from the aorta of middle-aged *Ldlr*$^{-/-}$ mice upon stimulation with αCD3/αCD28, which we only assessed in the CD4$^+$ T cell population (Fig. 8f, g), as we were facing technical challenges to isolate CD8$^+$ T cells from aortas. The increase in aortic CD4$^+$ T cell apoptosis was accompanied by a decrease in the percentage of aortic CD4$^+$ T$_{mem/eff}$ cells (Fig. 8h) that also showed an increase in cleaved caspase 3/7 (Fig. 8i).

To obtain more insights into the consequences of these processes for atherosclerotic plaques of middle-aged *T-Abc*$^{dko}$*Ldlr*$^{-/-}$ mice, we characterized their atherosclerotic lesions further. Similar to data in young *Ldlr*$^{-/-}$ mice (Supplementary Fig. 23a, d, e), T cell *Abca1/Abcg1* deficiency did not affect α-SMA staining or fibrous cap thickness in middle-

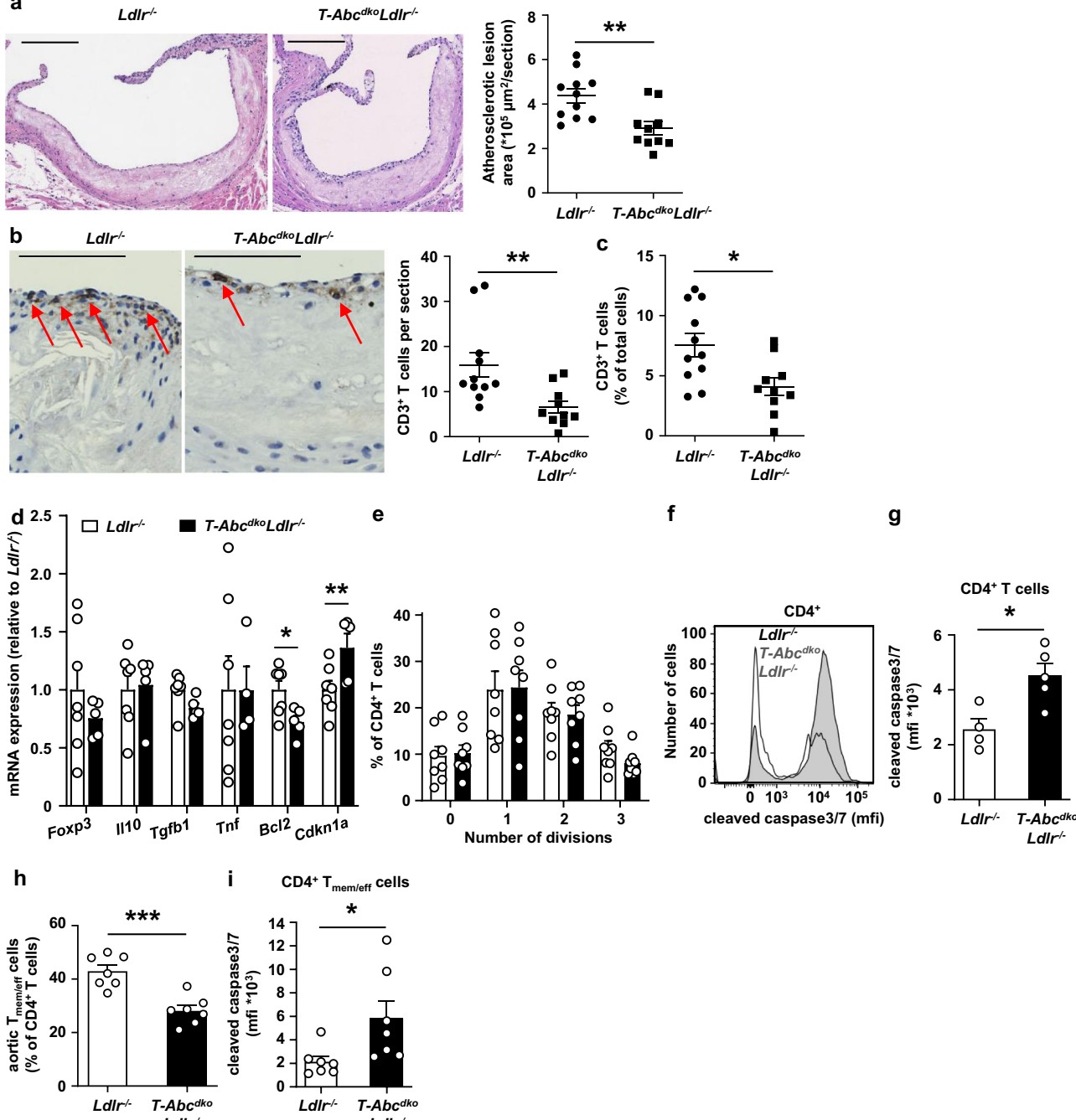

**Fig. 8 T cell *Abca1/Abcg1* deficiency decreases atherosclerosis in middle-aged *Ldlr*$^{-/-}$ mice.** *Ldlr*$^{-/-}$ and *T-Abc*$^{dko}$*Ldlr*$^{-/-}$ mice were fed a chow diet for 12–13 months. Hearts were isolated and paraffin sections of the aortic root were stained for (**a**) H&E or (**b**, **c**) CD3. **a** Representative pictures of H&E staining (left) and quantification (right) of atherosclerotic lesion area ($p = 0.003$). Scale bar represents 200 μm. $n = 11$ *Ldlr*$^{-/-}$ and $n = 10$ *T-Abc*$^{dko}$*Ldlr*$^{-/-}$ mice. **b** Representative pictures of CD3 staining on atherosclerotic lesions (left) and quantification (right) of total CD3$^+$ cells per section ($p = 0.008$). T cells were identified as cells with brown plasma membrane CD3 staining and the hematoxylin staining still visible. CD3$^+$ cells are depicted by arrows. Scale bar represents 80 μm. **c** CD3$^+$ T cells as percentage of total cells per section ($p = 0.0107$). **b**, **c** $n = 11$ *Ldlr*$^{-/-}$ and $n = 10$ *T-Abc*$^{dko}$*Ldlr*$^{-/-}$ mice. **d–i** Aortas were collected. **d** Aortic CD3$^+$ T cells were isolated, RNA was extracted, and *Foxp3*, *Il10*, *Tgfb1*, tumor necrosis factor (*Tnf*), *Bcl2* ($p = 0.023$), and *Cdkn1a* ($p = 0.025$) mRNA expression were measured by qPCR. $n = 7$ *Ldlr*$^{-/-}$ and $n = 5$ *T-Abc*$^{dko}$*Ldlr*$^{-/-}$ mice. **e** Total aortic cells were labeled with CFSE, stimulated with αCD3/αCD28 beads, and stained for CD4 after 96 h of stimulation. CFSE dilution was measured by flow cytometry and the number of divisions were quantified for aortic CD4$^+$ T cells. $n = 8$ *Ldlr*$^{-/-}$ and $n = 8$ *T-Abc*$^{dko}$*Ldlr*$^{-/-}$ mice. **f–i** Total aortic cells were stained for CD4, CD44, CD62L, and cleaved caspase 3/7. **f** Representative flow cytometry plots of cleaved caspase 3/7 in CD4$^+$ T cells gated as in Supplementary Fig. 12d, and (**g**) quantification ($p = 0.015$). $n = 4$ *Ldlr*$^{-/-}$ and $n = 5$ *T-Abc*$^{dko}$*Ldlr*$^{-/-}$ mice. **h** Aortic CD4$^+$ T$_{mem/eff}$ cells ($p = 0.0003$) as percentage of aortic CD4$^+$ T cells and (**i**) cleaved caspase3/7 in aortic CD4$^+$ T$_{mem/eff}$ cells ($p = 0.033$). $n = 7$ *Ldlr*$^{-/-}$ and $n = 7$ *T-Abc*$^{dko}$*Ldlr*$^{-/-}$ mice. For all panels, error bars represent SEM. Biologically independent samples were included. $p$ value was determined by unpaired two-tailed Student's $t$ test. *$p < 0.05$, **$p < 0.01$, ***$p < 0.001$. Source data are provided as a Source Data file.

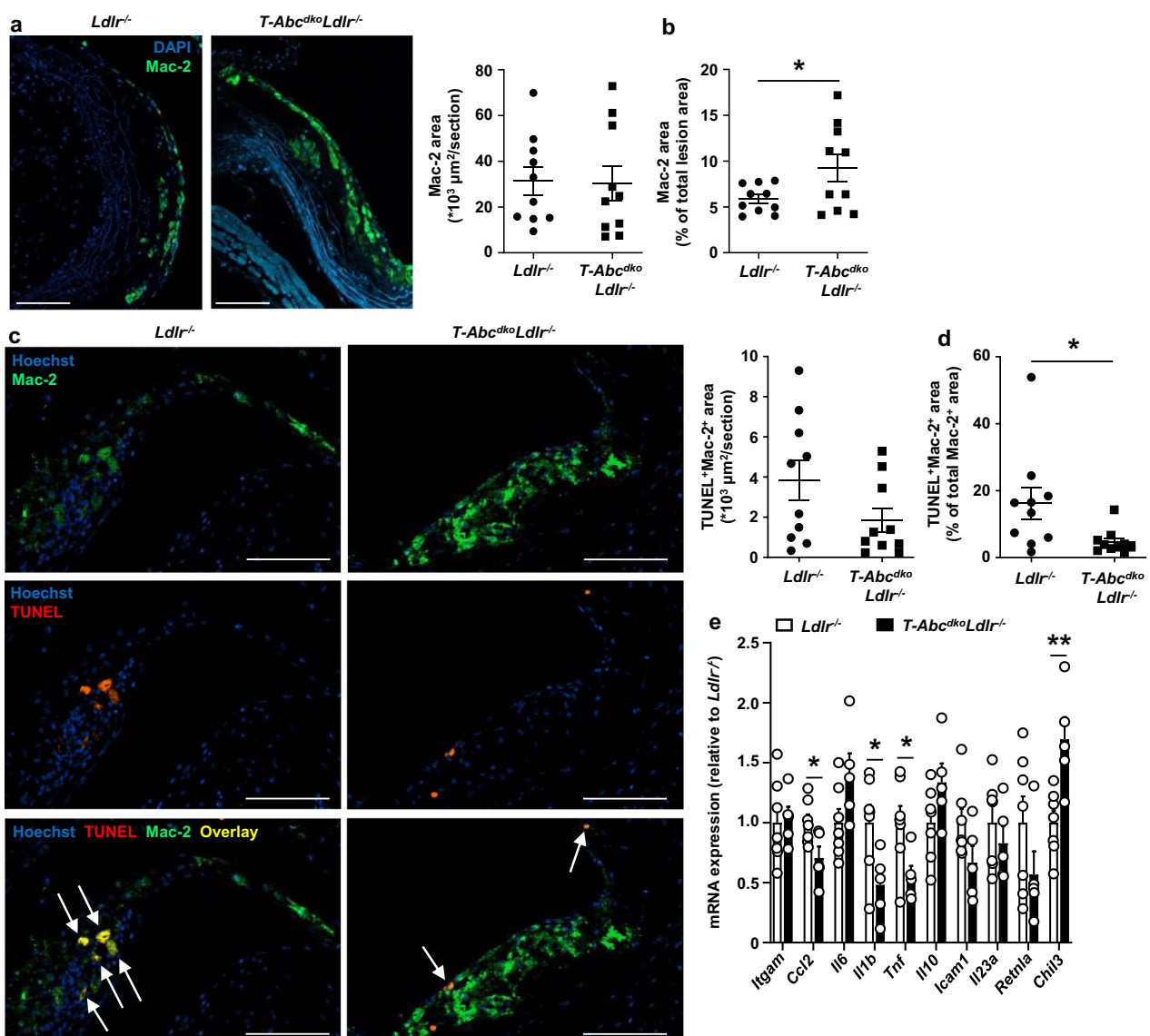

**Fig. 9 T cell *Abca1/Abcg1* deficiency increases macrophage content in atherosclerotic lesions from middle-aged *Ldlr*−/− mice.** *Ldlr*−/− and *T-Abc*dko*Ldlr*−/− mice were fed a chow diet for 12–13 months. **a–d** Hearts were isolated and paraffin sections of the aortic root were stained for (**a–c**) Mac-2 and DAPI, or (**d–f**) Mac-2, Terminal deoxynucleotidyl transferase dUTP nick end labeling (TUNEL), and Hoechst. **a** Representative pictures of Mac-2 staining (left) and quantification (right) of total Mac-2+ area. **b** Mac-2+ area as a percentage of total atherosclerotic lesion area (*p* = 0.046). **a, b** *n* = 10 *Ldlr*−/− and *n* = 10 *T-Abc*dko*Ldlr*−/− mice. **c** Representative pictures of Mac-2 and TUNEL stainings (left) and quantification (right) of total TUNEL+Mac-2+ area. Apoptotic macrophages were identified as Mac-2+TUNEL+ and are depicted by arrows. **d** TUNEL+Mac-2+ area as a percentage of total Mac-2+ area (*p* = 0.030). **c, d** *n* = 10 *Ldlr*−/− and *n* = 10 *T-Abc*dko*Ldlr*−/− mice. **a, c** Scale bar represents 100 μm. **e** Aortas from *Ldlr*−/− and *T-Abc*dko*Ldlr*−/− mice were collected, CD3− cells were isolated, RNA was extracted, and integrin subunit alpha M (*Itgam*), CC-chemokine ligand 2 (*Ccl2*) (*p* = 0.028), *Il6*, *Il1b* (*p* = 0.030), *Tnf* (*p* = 0.033), *Il10*, intracellular adhesion molecule-1 (*Icam1*), *Il23a*, resistin-like alpha (*Retnla*), and chitinase-like 3 (*Chil3*) (*p* = 0.0056) mRNA expression were measured by qPCR. *n* = 7 *Ldlr*−/− and *n* = 5 *T-Abc*dko*Ldlr*−/− mice. For all panels, error bars represent SEM. Biologically independent samples were included. *p* value was determined by unpaired two-tailed Student's *t* test. **p* < 0.05. Source data are provided as a Source Data file.

aged *Ldlr*−/− mice (Supplementary Fig. 25c–e), but decreased total collagen area in middle-aged *Ldlr*−/− mice (Supplementary Fig. 25f). However, after correction for total lesion area, collagen content was not different between the genotypes (Supplementary Fig. 25g). Only few necrotic cores were present in atherosclerotic plaques and the necrotic core area was not different between the genotypes (Supplementary Fig. 25h, i). Although the density of macrophages in atherosclerotic lesions was relatively low (~6–10% of the atherosclerotic lesions), T cell *Abca1/Abcg1* deficiency did increase macrophage content by ~60% (Fig. 9a, b). Previous studies have shown that in advanced atherosclerotic plaques, T cells induce macrophage apoptosis, primarily mediated by granzyme B or perforin[59]. While T cell *Abca1/Abcg1* deficiency showed a tendency to decrease the total Terminal deoxynucleotidyl transferase dUTP nick end labeling (TUNEL)+Mac-2+ area in middle-aged mice, this decrease became statistically significant when corrected for total Mac-2+ area (Fig. 9c, d), reflecting a decrease in macrophage apoptosis. These data are consistent with T cell *Abca1/Abcg1* deficiency inducing less macrophage killing upon TCR stimulation in the T cell macrophage co-incubation experiment (Fig. 4i). We then evaluated mRNA expression of inflammatory cytokines in the aortic T cell negative fraction,

consisting of endothelial cells, myeloid cells, and smooth muscle cells (SMCs), in middle-aged $Ldlr^{-/-}$ or $T\text{-}Abc^{dko}Ldlr^{-/-}$ mice. T cell $Abca1/Abcg1$ deficiency decreased mRNA expression of the M1 macrophage markers CC-chemokine ligand 2 ($Ccl2$), $Il1b$, and $Tnf$, and increased mRNA expression of the M2 macrophage marker chitinase-like 3 ($Chil3$), while not affecting mRNA expression of the myeloid cell marker integrin subunit alpha M ($Itgam$), or the cytokines $Il6$, $Il10$, or $Il23a$, or intracellular adhesion molecule-1 ($Icam1$), or the M2 macrophage marker resistin-like alpha ($Retnla$) (Fig. 9e). These changes in mRNA expression are consistent with a decrease in aortic inflammation, most likely due to increased T cell apoptosis and decreased IFN-γ secretion. Collectively, T cell $Abca1/Abcg1$ deficiency decreased atherosclerotic lesion size in middle-aged $Ldlr^{-/-}$ mice, presumably due to increased T cell apoptosis and decreased inflammatory gene expression in the aorta. Moreover, T cell $Abca1/Abcg1$ deficiency increased macrophage lesion content in middle-aged $Ldlr^{-/-}$ mice, likely due to decreased macrophage apoptosis.

## Discussion

These studies provide a link between defective cholesterol efflux and plasma membrane cholesterol accumulation and membrane stiffening in promoting T cell activation and T cell apoptosis with athero-protective effects in middle-aged mice (Fig. 10). Previous studies have shown that T cells accumulate cholesterol during aging[10], and that T cell aging induces T cell activation and decreases peripheral T cell numbers. Mechanistically, we now

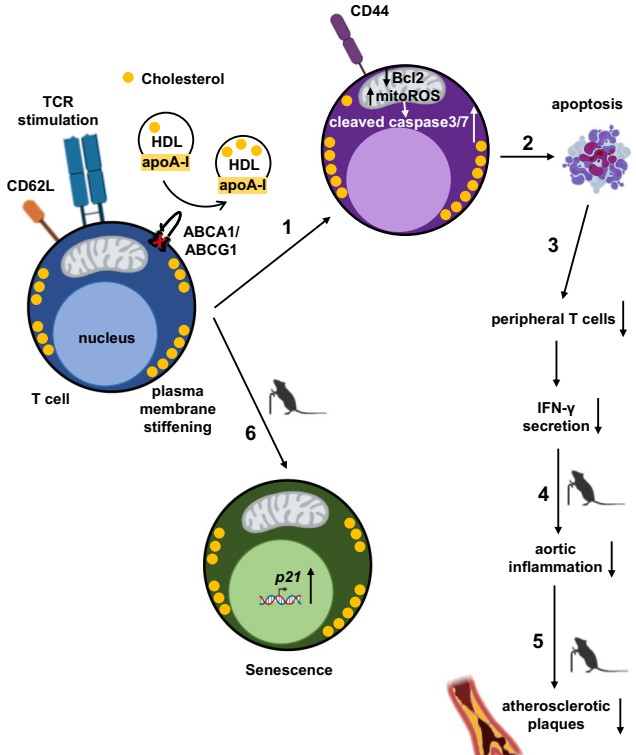

**Fig. 10 T cell *Abca1/Abcg1* deficiency induces T cell apoptosis and T cell senescence in middle-aged mice.** (1) T cell *Abca1/Abcg1* deficiency increases T cell activation, leading to (2) T cell apoptosis and (3) decreases in peripheral T cells and interferon gamma (IFN-γ) secretion. Consequently, in middle-aged mice, T cell *Abca1/Abcg1* deficiency decreases (4) aortic inflammation leading to (5) a decrease in atherosclerotic lesion area. (6) T cell *Abca1/Abcg1* deficiency induces cellular senescence in middle-aged mice. TCR, T cell receptor.

found that T cell $Abca1/Abcg1$ deficiency induced premature T cell aging, reflected by an almost complete suppression of T cell proliferation and expression of senescence markers, as well as a pronounced increase in T cell apoptosis (Fig. 10).

Association studies in the multi-ethnic study of atherosclerosis (MESA) have shown that a low level of CD4+ naïve T cells in blood was associated with increased carotid intima media thickness (cIMT), suggesting that T cell activation enhances atherogenesis in humans[60]. T cells are numerous in advanced atherosclerotic plaques of human carotids (~50–65% of all plaque cells) and most T cells in atherosclerotic plaques are of the activated CD44+ phenotype[2,61,62]. Sharing similarities with the MESA study, another study found a positive correlation between blood $T_{mem/eff}$ cells and cIMT, and elevated blood $T_{mem/eff}$ cells in patients with chronic stable angina (CSA) or acute myocardial infarction (AMI)[63]. However, these studies do not take into account the balance between T cell activation, apoptosis, or senescence. Indeed, T cell $Abca1/Abcg1$ deficiency decreased the expression of pro-inflammatory cytokines in the aorta of middle-aged $Ldlr^{-/-}$ mice despite blood T cells being more activated. However, unlike in blood, T cell $Abca1/Abcg1$ deficiency decreased the percentage of CD4+ $T_{mem/eff}$ cells in the aorta, accompanied by increased CD4+ T cell apoptosis. The increase in T cell apoptosis likely accounts for decreased IFN-γ secretion and decreased inflammation in the aorta. A recent study in $Apoe^{-/-}$ mice has also shown that 20 weeks of WTD feeding, which increases T cell cholesterol content[5], increases CD4+ T cell apoptosis in T cells from para-aortic LNs and decreases CD4+IFN-γ+ T cells[9], consistent with our data. However, this study was done in a global $Apoe$ knockout and therefore effects of T cell cholesterol accumulation on atherosclerotic plaques could not be addressed directly.

T cells in atherosclerotic plaques were increased by 2-fold in middle-aged compared to young $Ldlr^{-/-}$ mice, suggesting a more prominent effect of T cells on atherosclerosis in middle-aged mice. Consistent with this hypothesis, previous studies have found different effects of T cells at different stages of lesion development. Complete CD4+ or CD8+ T cell ablation[59,64–66] reduces atherosclerosis, suggesting that T cells in plaques are mainly pro-atherogenic. In early atherogenesis, CD4+ and CD8+ T cells induce macrophage inflammation and monopoiesis in BM mediated by IFN-γ[65,66]. In advanced lesions, CD8+ T cells induce macrophage apoptosis and necrotic core formation mediated by granzyme B and perforin[59]. T cell $Abca1/Abcg1$ deficiency did not affect blood myeloid cells or necrotic core formation, but increased macrophage content in middle-aged $Ldlr^{-/-}$ mice, accompanied by a decrease in macrophage apoptosis. The latter would perhaps not have been expected since T cell $Abca1/Abcg1$ deficiency increased CD8+granzyme B+ T cells upon αCD3/IL-2 stimulation. However, upon the same stimulus, $Abca1/Abcg1$ deficiency increased T cell apoptosis, and decreased macrophage death in a T cell mediated cytotoxicity assay. Presumably, the effect of $Abca1/Abcg1$ deficiency on T cell apoptosis is predominant, and consequently, less granzyme B may have been secreted to mediate macrophage death. The decreased macrophage death in the T cell mediated cytotoxicity assay likely explains the decreased macrophage apoptosis and increased macrophage lesion content in plaques.

T cell $Abca1/Abcg1$ deficiency also decreased CD25+Foxp3+ $T_{reg}$ cells in para-aortic LNs, which would be suggestive of T cell $Abca1/Abcg1$ deficiency being pro-atherogenic. However, $Foxp3$ mRNA expression was not affected in aortic T cells, suggesting no differences in $T_{reg}$ cells in atherosclerotic plaques. Our in vitro studies suggested that the decrease in CD25+Foxp3+$T_{reg}$ cells was entirely due to T cell apoptosis. T cell apoptosis is an anti-inflammatory process[67].

One striking observation was that unlike in young mice, middle-aged *Abca1/Abcg1*-deficient T cells almost completely lost their ability to proliferate upon TCR stimulation. While this could be the consequence of increased T cell apoptosis, the senescence marker *Cdkn1a* was also increased, suggestive of premature T cell aging. Similar to T cell *Abca1/Abcg1* deficiency, aging increases T cell plasma membrane cholesterol accumulation[10,11] and increases T cell activation, while decreasing peripheral T cell numbers[12,50,51]. During aging, thymic involution decreases output of naïve T cells[16]. However, T cells undergo homeostatic proliferation in the periphery giving rise to new T cells independent of thymic output[13,68]. We found that T cells from aged wild-type mice showed increased cholesterol accumulation compared to young wild-type mice, and increased apoptosis upon TCR stimulation, similar to our findings in mice with T cell *Abca1/Abcg1* deficiency. Together, these data indicate broader physiological relevance of our findings on apoptosis in T cell *Abca1/Abcg1* deficiency, and suggest that T cell plasma membrane cholesterol accumulation may account for increased T cell apoptosis during aging, resulting in a decrease in peripheral T cell numbers.

Our finding that T cell *Abca1/Abcg1* deficiency only affected atherosclerotic lesion size in middle-aged *Ldlr*[−/−] mice is suggestive of a specific effect of *Abca1/Abcg1* on T cells during aging that affects atherogenesis. T cell *Abca1/Abcg1* deficiency had similar effects on most T cell subsets in young and middle-aged mice, but induced T cell senescence only in middle-aged mice. In humans, T cell aging due to repeated TCR stimulation is characterized by loss of CD28[69]. Human T cells deficient in CD28 are senescent but still may have an effector function[70], and produce high levels of TNF and IFN-γ[71]. While initial studies reported that CD4[+]CD28[null] T cells correlate with unstable angina and acute coronary syndromes in humans[71–74], this was recently challenged by a multi-center study with a larger number of patients[75]. Opposite to the previous findings[71–74], in this study[75], levels of CD4[+]CD28[null] T cells in blood were associated with a lower risk for first time coronary events in a population-based cohort, and did not correlate with cIMT. Based on these findings[75], it is unlikely that senescent T cells contributed to lesion size in our study.

In addition to atherosclerotic plaques, peripheral T cells in blood, LNs, and spleen were decreased in *T-Abc*[dko]*Ldlr*[−/−] mice compared to their littermate controls. Although Ldlr mediated LDL uptake increases TCR stimulation in vitro[76], illustrating the importance of membrane cholesterol content for TCR signaling, *Ldlr* deficiency has not been reported to decrease peripheral T cell numbers in vivo. Consistently, we found that the increased T cell apoptosis and decreased peripheral T cells upon T cell *Abca1/Abcg1* deficiency were independent of *Ldlr* expression. Single deficiency of *Abca1* or *Abcg1* did not affect peripheral T cells in blood or T cell apoptosis, likely because deficiency of one transporter results in upregulation of the other. Indeed, it has been shown previously that *Abcg1* deficiency upregulates *Abca1* expression in T cells[23].

A previous study has attributed the decrease in peripheral T cells in mice with T cell *Abca1/Abcg1* deficiency or T cell deficiency of the transcription factor the *liver X receptor* (*LXR*)α and β that induces *Abca1* and *Abcg1* expression, to increased thymic CD4[+] and CD8[+] T cell apoptosis, while not excluding that extrathymic effects on T cells may also have contributed[29]. However, in these studies, the *Abca1* and *Abcg1* floxed genes and also the *Lxrα/Lxrβ* floxed genes were expressed under control of the *LckCre* promoter[29]. In T cells, *Lxrβ* is more highly expressed than *Lxrα*[20]. *CD4CreLxrβ*[fl/fl] mice showed decreased peripheral T cells with only thymic CD4[+] T cells and CD4[+] T_reg cells being decreased, while thymic CD8[+] T cells were unchanged[21]. Perhaps

the *LckCre* promoter resulted in a more complete deficiency of *Abca1* and *Abcg1* in single positive thymic CD4[+] and CD8[+] T cells than the *CD4Cre* promoter that we and others used[21], explaining the more pronounced effects on thymic T cells. The decrease in thymic CD4[+] and CD8[+] T cells in *LckCreLxrα*[fl/fl]*Lxrβ*[fl/fl] and *LckCreAbca1*[fl/fl]*Abcg1*[fl/fl] mice was attributed to increased surface expression of the cell death receptor FAS[29], which is localized in lipid rafts[43]. Also, upon TCR stimulation, peripheral CD4[+] T cells from *CD4CreLxrβ*[fl/fl] mice showed increased FAS expression and apoptosis; however it was suggested that additional mechanisms may contribute to apoptosis in *CD4CreLxrβ*[fl/fl] T cells[21]. Our data in *CD4CreAbca1*[fl/fl]*Abcg1*[fl/fl]*Ldlr*[−/−] mice show that FAS expression was not affected upon TCR stimulation, but Bcl2 expression was decreased. We thus attributed the increased apoptosis in CD4[+] and CD8[+] T cells (the latter mainly in the CD8[+]T_mem/eff T cells) from *CD4CreAbca1*[fl/fl]*Abcg1*[fl/fl]*Ldlr*[−/−] mice to a decrease in the anti-apoptotic protein Bcl2. This mechanism may also have contributed to the increased apoptosis in CD4[+]T cells from *CD4CreLxrβ*[fl/fl] mice.

Similar to *CD4CreAbca1*[fl/fl]*Abcg1*[fl/fl]*Ldlr*[−/−] mice, *CD4CreLxrβ*[fl/fl] mice showed increased spontaneous T cell activation[21]. This was attributed to a functional impairment of *CD4CreLxrβ*[fl/fl] T_reg cells, and recapitulated in *Foxp3CreLxrβ*[fl/fl] mice, indicating that it was a T_reg cell-intrinsic effect[21]. In these specific *Foxp3CreLxrβ*[fl/fl] T cells, unlike in other CD4[+] T cells, *Abca1* and *Abcg1* expression were not affected by *Lxrβ* deficiency, suggesting that other LXR target genes contributed to the dysfunctional T_reg differentiation[21]. Although peripheral T_reg cells were also decreased in *CD4CreAbca1*[fl/fl]*Abcg1*[fl/fl]*Ldlr*[−/−] mice, there was no specific decrease in the percentage of CD25[+]Foxp3[+] T cells, as in *CD4CreLxrβ*[fl/fl] mice[21], and the decreases in peripheral T_reg cells were mainly the consequence of an overall increase in T cell apoptosis. Given that TCR signaling was increased in *CD4CreAbca1*[fl/fl]*Abcg1*[fl/fl]*Ldlr*[−/−] mice, and enhanced TCR signaling promotes T cell activation[38], we postulate that this mechanism accounted for the increase in T cell activation in *CD4CreAbca1*[fl/fl]*Abcg1*[fl/fl]*Ldlr*[−/−] mice compared to controls.

Several studies have shown that increased plasma membrane cholesterol accumulation enhances TCR signaling[8,20,23,77]. Conversely, impaired cholesterol synthesis in the context of SCAP deficiency suppresses TCR signaling[37]. We found that even when TCR-induced T cell proliferation was almost completely abolished in T cells with *Abca1/Abcg1* deficiency, presumably due to T cell senescence, cholesterol depletion by rHDL still further decreased it, substantiating the importance of T cell membrane cholesterol for TCR signaling. *Acat1* deficiency increases plasma membrane cholesterol accumulation and TCR nanoclustering, which enhances TCR signaling[8]. Cholesterol accumulation also increased TCR nanoclustering in a study employing artificial membranes[78]; however a later study from the same group showed that binding of cholesterol to TCRβ inhibited TCR signaling[79], suggesting rather an inhibitory than a stimulating effect. Upon TCR signaling, the plasma membrane cholesterol content increases[20,21], presumably due to suppression of *Abca1* and *Abcg1* expression and increased cholesterol synthesis[20]. Indeed, our studies have shown that T cell *Abca1/Abcg1* deficiency increased plasma membrane cholesterol accumulation leading to plasma membrane stiffening and enhanced TCR signaling. Moreover, a recent study has shown that T cell *histone deacetylase 3* (*Hdac3*) deficiency decreases TCR signaling and CD4[+] T cell proliferation, which was attributed to decreased membrane cholesterol content and increased *Abca1* and *Abcg1* mRNA expression[77]. This substantiates our data on T cell *Abca1/Abcg1* deficiency increasing TCR signaling.

In conclusion, our studies indicate a link between T cell plasma membrane cholesterol accumulation, T cell plasma membrane stiffening, TCR signaling, T cell activation, and T cell apoptosis in

T cell *Abca1/Abcg1* deficiency, which suppresses aortic inflammation and atherosclerosis in middle-aged, but not young mice. Our studies indicate that plasma membrane cholesterol accumulation not only induces apoptosis, but also a premature T cell aging phenotype characterized by an increase in T cell senescence. This suggests that upregulation of *Abca1* and *Abcg1* in T cells, for example by an LXR agonist, may suppress T cell apoptosis and senescence, which could be crucial in maintaining T cell numbers and peripheral T cell tolerance, especially during aging. Further, statins, although reported to inhibit TCR induced T cell proliferation[8], could also suppress T cell apoptosis and senescence by suppressing cholesterol synthesis in T cells. It would be of interest to investigate this further in a large population cohort.

## Methods

**Animals**. *Abca1*$^{fl/fl}$*Abcg1*$^{fl/fl}$ (strain #: 021067), *CD4Cre* (strain #: 017336), *Ldlr*$^{-/-}$ (strain #: 002207), and wild-type (strain #: 000664) mice were purchased from Jackson Laboratories (Bar Harbor, ME). All mice were in the C57Bl6/J background. *Abca1*$^{fl/fl}$*Abcg1*$^{fl/fl}$ (control; *Ldlr*$^{+/+}$) and *CD4Cre* mice were intercrossed to obtain *Abca1*$^{fl/fl}$*Abcg1*$^{fl/fl}$, *Abca1*$^{fl/fl}$, *Abcg1*$^{fl/fl}$, *CD4CreAbca1*$^{fl/fl}$, *CD4CreAbcg1*$^{fl/fl}$, and *CD4CreAbca1*$^{fl/fl}$*Abcg1*$^{fl/fl}$ (*T-Abc*$^{dko}$) littermates. *CD4CreAbca1*$^{fl/fl}$, *CD4CreAbcg1*$^{fl/fl}$, and *CD4CreAbca1*$^{fl/fl}$*Abcg1*$^{fl/fl}$ mice were intercrossed with *Ldlr*$^{-/-}$ mice to generate *Abca1*$^{fl/fl}$*Ldlr*$^{-/-}$, *Abcg1*$^{fl/fl}$*Ldlr*$^{-/-}$ (all *Ldlr*$^{-/-}$), *CD4CreAbca1*$^{fl/fl}$*Ldlr*$^{-/-}$ (*T-Abca1*$^{ko}$*Ldlr*$^{-/-}$), *CD4CreAbcg1*$^{fl/fl}$*Ldlr*$^{-/-}$ (*T-Abcg1*$^{ko}$*Ldlr*$^{-/-}$), and *CD4CreAbca1*$^{fl/fl}$*Abcg1*$^{fl/fl}$*Ldlr*$^{-/-}$ (*T-Abc*$^{dko}$*Ldlr*$^{-/-}$) littermates. Mice were fed a chow diet (10% fat, 23% protein, and 67% carbohydrates; V1554-703, Ssniff Spezialdiäten GmbH) or, starting at 8–12 weeks of age, a WTD (40% fat, 0.15% cholesterol; D12079B, Research Diets).

For all studies, littermates were used, and mice were housed under standard laboratory conditions at 21.5 °C (range: 20.5–22.5 °C) ambient temperature and 55% (range: 30–70%) humidity with a light cycle of 12 h and ad libitum water and food (chow diet or WTD). Water and cages were autoclaved. Cages were changed once weekly, and the health status of the mice was monitored based on body weight, coat, and behavior. The mouse genotype did not cause changes in body weight (mouse body weight between 20–40 g, depending on gender and age) or health. Female littermates (atherosclerosis studies) or littermates from both sexes were randomly assigned to experimental groups, unless stated otherwise. The exact number of the mice used for the experiments are indicated for each experiment in the figure legends. In general, $n = 3$–18 mice were used per group and experiments were repeated at least once if $n = 3$ mice were used per experiment to confirm the reproducibility of the results. No inclusion or exclusion criteria were used. All protocols were approved by the Institutional Animal Care and Use Committee from the University of Groningen (Groningen, the Netherlands) under permit number AVD105002015244 and adhered to guidelines set out in the 2010/63/EU directive.

**Reagents and resources**. Please see Supplementary Table 3.

**T cell and thymocyte isolations**. Spleens were isolated from control (*Ldlr*$^{+/+}$), *T-Abc*$^{dko}$, *Ldlr*$^{-/-}$, *T-Abc*$^{dko}$*Ldlr*$^{-/-}$, *T-Abca1*$^{ko}$*Ldlr*$^{-/-}$, *T-Abcg1*$^{ko}$*Ldlr*$^{-/-}$, and young (3 months) and aged (24 months) wild-type mice, mashed on a 40 µm strainer, and red blood cells (RBCs) were lysed (BD Pharm Lyse, BD Biosciences). Para-aortic LNs were isolated from *Ldlr*$^{-/-}$ and *T-Abc*$^{dko}$*Ldlr*$^{-/-}$ mice and mashed on a 40 µm strainer in PBS. Splenic and LN homogenates were then centrifuged, washed, and resuspended in MACS buffer (PBS containing 0.5% BSA and 2 mM EDTA). Dead cells were removed using the Dead Cell Removal kit (Miltenyi Biotec), and total T cells were isolated using the Pan T cell kit (Miltenyi Biotec). CD4$^+$ and CD8$^+$ T cells were isolated using CD4 and CD8 coated magnetic beads (Miltenyi Biotec), respectively. For thymocyte isolations, *Ldlr*$^{-/-}$ and *T-Abc*$^{dko}$*Ldlr*$^{-/-}$ thymi were isolated and mashed on a 40 µm strainer in PBS.

**Aortic cell isolations**. Aortas were isolated from middle-aged (12–13 months) female *Ldlr*$^{-/-}$ and *T-Abc*$^{dko}$*Ldlr*$^{-/-}$ mice. For mRNA expression analyses, aortas were digested for 1 h at 37 °C using an enzyme mixture that contained liberase TH (4 U/mL; Roche), DNAse I (40 U/mL; Sigma-Aldrich), and hyaluronidase (60 U/mL; Sigma-Aldrich). Aortic homogenates were then centrifuged, washed, and resuspended in MACS buffer. Total aortic CD3$^+$ T cells and CD3$^-$ cells were isolated using the Pan T cell kit. For in vitro stimulation, aortas were digested for 45 min at 37 °C using an enzyme mixture that contained liberase TM (4 U/mL; Roche), DNAse I (120 U/mL), and hyaluronidase (60 U/mL). Aortic digestion was stopped by adding 10% FBS in RPMI 1640 medium (Gibco). Aortic homogenates were then passed through a 70 µm strainer, centrifuged, washed, and resuspended in RPMI 1640 medium supplemented with 10% FCS, 1% pen-strep, and 50 µM β-mercaptoethanol.

**RNA extraction from splenic and aortic cells**. Splenic total T cells, CD4$^+$, and CD8$^+$ T cells from 4–5 months old male and female control, *T-Abc*$^{dko}$, and *Ldlr*$^{-/-}$ mice, middle-aged (12–13 months) male and female *Ldlr*$^{-/-}$ and *T-Abc*$^{dko}$*Ldlr*$^{-/-}$ mice, and young (3 months) and aged (24 months) wild-type male and female mice, and aortic total CD3$^+$ T cells and CD3$^-$ cells from middle-aged female *Ldlr*$^{-/-}$ and *T-Abc*$^{dko}$*Ldlr*$^{-/-}$ mice were isolated as described above and resuspended directly into RLT buffer. RNA was then extracted using the RNeasy Mini Kit (Qiagen) and cDNA was synthesized using the Transcriptor Universal cDNA Master kit (Roche). *Abca1*, *Abcg1* (in splenic total T cells from control and *T-Abc*$^{dko}$ mice and CD4$^+$ and CD8$^+$ T cells from young wild-type and aged wild-type mice), *Ldlr*, *Hmgcr*, *Hmgcs*, *Srebf2*, *Srebf1*, *Acat1* (in CD4$^+$ and CD8$^+$ T cells from control, *T-Abc*$^{dko}$, young wild-type, and aged wild-type mice), *Cdkn2a* (in splenic CD4$^+$ and CD8$^+$ T cells from middle-aged *Ldlr*$^{-/-}$ and *T-Abc*$^{dko}$*Ldlr*$^{-/-}$ mice), *Cdkn1a* (in splenic CD4$^+$ and CD8$^+$ T cells and aortic total CD3$^+$ T cells from middle-aged *Ldlr*$^{-/-}$ and *T-Abc*$^{dko}$*Ldlr*$^{-/-}$ mice), *Foxp3*, *Tgfb1*, *Bcl2*, *Il10*, *Tnf* (in aortic total CD3$^+$ T cells from middle-aged *Ldlr*$^{-/-}$ and *T-Abc*$^{dko}$*Ldlr*$^{-/-}$ mice), *Itgam*, *Il6*, *Il1b*, *Ccl2*, *Icam1*, *Il23a*, *Retnla*, and *Chil3* (in aortic CD3$^-$ cells from middle-aged *Ldlr*$^{-/-}$ and *T-Abc*$^{dko}$*Ldlr*$^{-/-}$ mice) mRNA levels were assessed by qPCR using QuantStudio 7 Flex Real-Time PCR System (Applied Biosystems), and initial differences in RNA quantity were corrected for using the housekeeping genes *36b4* and *Cyclophilin B*. All primer sequences are available in Supplementary Table 3.

**Filipin staining—flow cytometry**. To analyze T cell membrane cholesterol, blood leukocytes (4–5 months old male and female *Ldlr*$^{-/-}$ and *T-Abc*$^{dko}$*Ldlr*$^{-/-}$ mice, and young (3 months) and aged (24 months) wild-type male and female mice) and thymocytes (4–5 months old male and female *Ldlr*$^{-/-}$ and *T-Abc*$^{dko}$*Ldlr*$^{-/}$ mice) were stained with antibodies against TCRβ-APC, CD8-PE, and CD4-FITC, and thymocytes with antibodies against CD4-APC and CD8-PE. Cells were then centrifuged, washed, and stained with 50 µg/mL Filipin III (Sigma-Aldrich) in 10% FCS/PBS for 45 min at room temperature (RT) and then washed twice with PBS. Samples were analyzed by flow cytometry immediately after staining. Filipin III mfi on blood TCRβ$^+$CD4$^+$ and TCRβ$^+$CD8$^+$ cells, and on thymic CD4$^+$CD8$^+$, CD4$^+$, and CD8$^+$ T cells was assessed on LSRII (BD Biosciences), running FACSDiVa software (BD Biosciences). The data were analyzed using the FlowJo software (FlowJo).

**Confocal microscopy, BODIPY, and filipin staining**. Splenic CD4$^+$ and CD8$^+$ T cells from 4–5 months old male and female control, *T-Abc*$^{dko}$, *Ldlr*$^{-/-}$, and *T-Abc*$^{dko}$*Ldlr*$^{-/-}$ mice, and total T cells from 4–5 months old male and female *T-Abca1*$^{ko}$*Ldlr*$^{-/-}$ and *T-Abcg1*$^{ko}$*Ldlr*$^{-/-}$ mice, and young (3 months) and aged (24 months) wild-type male and female mice were isolated as described above. Cells were seeded on Poly-L-Lysine (PLL)-coated glass coverslips and incubated at 37 °C for 1 h. For lipid droplet imaging, cells were fixed (4% PFA; 15 min; 4 °C), permeabilized for 30 min at RT using CLSM buffer (PBS containing 3% BSA, 10 mM glycine, 0.1% saponin), and stained for 30 min at RT with BODIPY 493/503 (Invitrogen; 7.5 µM in CLSM buffer). After two PBS washes, cells were mounted using VECTASHIELD Antifade Mounting Medium with DAPI (Vector Laboratories). To assess free cholesterol, cells were fixed (3% PFA; 1 h; RT) and stained with filipin (50 µg/mL in 10% FCS in PBS; 2 h; RT). After three PBS washes, cells were mounted using VECTASHIELD Antifade Mounting Medium (Vector Laboratories).

All samples were imaged on the Zeiss LSM800 microscope with immersion oil lens (63X,1.4 N.A) running Zen software (Zeiss). Background signal was removed using Zen software and fluorescence intensity of filipin or BODIPY was analyzed using ImageJ software (NIH).

**Choleratoxin B staining**. For analysis of T cell lipid rafts, blood leukocytes from 4–5 months old male and female *Ldlr*$^{-/-}$ and *T-Abc*$^{dko}$*Ldlr*$^{-/-}$ mice, and young (3 months) and aged (24 months) wild-type male and female mice were stained with a cocktail of antibodies against TCRβ-PB, CD8-PE, and CD4-APC. Cells were then centrifuged, washed, and stained with 200 ng/mL choleratoxin B-FITC (Sigma-Aldrich) for 1 h at RT. Samples were analyzed by flow cytometry immediately after staining. Choleratoxin B mfi on TCRβ$^+$CD4$^+$ and TCRβ$^+$CD8$^+$ cells was assessed by flow cytometry on the LSRII as described above for the filipin staining, and is an indirect indicator of lipid rafts.

**Gas chromatography—mass spectrometry**. Splenic T cells from female *Ldlr*$^{-/-}$ and *T-Abc*$^{dko}$*Ldlr*$^{-/-}$ mice fed a chow diet for 28 weeks, and young (3 months) and aged (24 months) wild-type male and female mice were isolated as described above. Cholestanol-D5 was added to the T cells as internal standard and cholesterol was extracted using hexane. For T cells from *Ldlr*$^{-/-}$ and *T-Abc*$^{dko}$*Ldlr*$^{-/-}$ mice, samples were split for measurement of either total or free cholesterol content using Gas Chromatography—Mass Spectrometry (7890B GS system, 5973 MS system, and 7693 A automatic liquid sampler from Agilent; positive chemical ionization mode with 5% ammonia in methane as reaction gas). A polar DB-WAXetr (30 m × 0.25 mm × 0.25 µm) column was used. CE content was calculated by subtracting free cholesterol from total cholesterol. Cholesterol content was normalized to cellular protein content measured by the Lowry assay. For T cells of

young and aged wild-type mice, total cholesterol content was assessed using the same procedure.

**Oil red O staining**. Female $Ldlr^{-/-}$ and $T\text{-}Abc^{dko}Ldlr^{-/-}$ mice were fed a chow diet for 28 or 38 weeks. After the indicated period of time on diet, mice were sacrificed and para-aortic LNs (28 weeks) and hearts (38 weeks) were isolated and embedded into OCT compound on dry ice. Frozen cross-sections (4 µm thickness) were made and stained with Oil Red O (Sigma-Aldrich) to assess neutral lipid accumulation. Sections were counterstained with haematoxylin. Quantification of the number of lipid droplets on sections of para-aortic LN sections (per field) was performed in a blinded fashion, *i.e.* the observer was unaware of the genotypes. For atherosclerosis analysis, the average from 6 heart sections (40 µm distance between sections) for each animal was used to determine lesion size. Lesion size was quantified by morphometric analysis and Oil Red O$^{+}$area was measured using ImageJ software (NIH).

**Transmission electron microscopy**. Splenic CD4$^{+}$ and CD8$^{+}$ T cells from 4–5 months old male and female $Ldlr^{-/-}$and $T\text{-}Abc^{dko}Ldlr^{-/-}$ mice were isolated as described above. Cells were seeded on PLL-coated glass coverslips placed in 24-well plates and incubated at 37 °C for 1 h. Cells were then fixed with 2% glutaraldehyde (Sigma-Aldrich) in PB (0.1 M phosphate buffer, pH 7.4) for 60 min at RT. After three PB washes, cells were post-fixed with 1% osmium tetroxide in PB for 60 min at RT. Cells were incubated overnight in 0.5% uranyl acetate, dehydrated with graded steps of ethanol (50%, 70%, 96%, 100%), and embedded in Epon resin. Sections of 70 nm thickness were made and stained with 2% uranyl acetate solution and lead citrate solution. Stained sections were then examined using a CM12 transmission electron microscope (Phillips).

**Fluorescence-lifetime imaging microscopy**. Splenic CD4$^{+}$ and CD8$^{+}$ T cells from 4–5 months old male and female $Ldlr^{-/-}$and $T\text{-}Abc^{dko}Ldlr^{-/-}$ mice were isolated as described above. Cells were stained with BODIPY C10 (kind gift from Dr. Ulf Diederichsen, Georg-August-Universität Göttingen, Germany; 4 µM; 30 min; on ice). Cells were washed twice with phenol red free RPMI 1640 medium and seeded in PLL coated µ-Slide 8 Well Glass Bottom Chambers (Ibidi). Cells were incubated at 37 °C for 30 min, and then BODIPY C10 fluorescence lifetime was assessed by fluorescence-lifetime imaging microscopy (FLIM). All images were collected with a MicroTime 200 microscope (PicoQuant) equipped with an Olympus (100x/1,4) oil immersion objective. Images were acquired using the SymPhoTime 64 software (PicoQuant). Data analysis of the FLIM images was performed using the open-source FLIMfit software[80]. Further image loading options included 2 × 2 spatial binning and time bins 1563 (32 ps/bin). After loading FLIM images, the software's default settings were used. BODIPY C10 fluorescence lifetime of a whole cell (with lipid droplets (LDs)) or of the plasma membrane without including LDs was determined by fitting the fluorescence histograms with a mono-exponential decay function, using the tool "Fit Selected Decay". Per mouse, 43–76 CD4$^{+}$ and CD8$^{+}$ T cells were analyzed.

**Flow cytometry and total white blood cell counts**. Blood samples were collected from 4–5 months old male and female control, $T\text{-}Abc^{dko}$, $Ldlr^{-/-}$, $T\text{-}Abca1^{ko}Ldlr^{-/-}$ and $T\text{-}Abcg1^{ko}Ldlr^{-/-}$ mice, male and female $Ldlr^{-/-}$ and $T\text{-}Abc^{dko}Ldlr^{-/-}$ mice (4–5 or 12–13 months old, or on WTD for 10 weeks), and young (3 months) and aged (24 months) wild-type male and female mice by tail bleeding into Eppendorf tubes containing EDTA. Total white blood cell counts were assessed using the Medonic CD620 hematology analyzer (Boule Medical). For flow cytometry, tubes were kept at 4 °C for the whole procedure unless stated otherwise. RBCs were lysed, and white blood cells were centrifuged, washed, and resuspended in FACS buffer (HBSS containing 0.1% BSA and 0.5 M EDTA). For all stainings, antibodies listed below were used at a 1/200 dilution, unless stated otherwise. For analysis of blood T cells, cells were stained with a cocktail of antibodies against TCRβ-PB, CD8-FITC, and CD4-APC. CD4$^{+}$ T cells were identified as TCRβ$^{+}$CD4$^{+}$ and CD8$^{+}$ T cells as TCRβ$^{+}$CD8$^{+}$. For analysis of naïve and activated blood T cells, cells were stained with a cocktail of antibodies against TCRβ-PB, CD8-FITC, CD44-PE-Cy7, and CD62L-APC. Among these populations, T$_{naive}$ cells were identified as CD44$^{-}$CD62L$^{+}$, T$_{CM}$ cells as CD8$^{+}$CD44$^{+}$CD62L$^{+}$, and T$_{mem/eff}$ cells as CD44$^{+}$CD62L$^{-}$. The same stainings were carried out on para-aortic LN and splenic homogenates. For analysis of thymocytes (4–5 months old male and female $Ldlr^{-/-}$ and $T\text{-}Abc^{dko}Ldlr^{-/-}$ mice), thymic homogenates were stained with CD4-APC and CD8-FITC or TCRβ-PerCP-Cy5.5, CD24-PE-Cy7, and CD69-APC. For analysis of blood myeloid cells (for male and female $Ldlr^{-/-}$ and $T\text{-}Abc^{dko}Ldlr^{-/-}$ mice; 4–5 months old or on WTD for 10 weeks), cells were stained with a cocktail of antibodies against CD45-PE-Cy7, Ly6-C/G-PerCP-Cy5.5, and CD115-PE. Monocytes were identified as CD45$^{+}$CD115$^{+}$ and further separated into Ly6-C$^{hi}$ and Ly6-C$^{lo}$ subsets, and neutrophils were identified as CD45$^{+}$CD115$^{-}$Ly6-C/G$^{hi}$.

For analysis of T$_{reg}$ cells in thymus (4–5 months old male and female $Ldlr^{-/-}$ and $T\text{-}Abc^{dko}Ldlr^{-/-}$ mice) and para-aortic LN (female $Ldlr^{-/-}$ and $T\text{-}Abc^{dko}$ $Ldlr^{-/-}$ mice fed a chow diet for 28 weeks or a WTD for 10 weeks), tissue homogenates were stained with a cocktail of antibodies against CD4-FITC and CD25-APC (for LN T$_{reg}$ cells; Mouse Regulatory T Cell Staining Kit #1, eBioscience) or CD4-PB, CD8-FITC, and CD25-PECy7 (for thymic T$_{reg}$ cells).

Cells were then fixed and permeabilized using the Foxp3/Transcription Factor Staining Buffer Set (eBioscience), and subsequently stained with Foxp3-PE (for LN T$_{reg}$ cells) or Foxp3-APC (for thymic T$_{reg}$ cells) (1/100 dilution) or their isotype control (1/100 dilution). T$_{reg}$ cells were identified as CD8$^{-}$CD4$^{+}$CD25$^{+}$Foxp3$^{+}$ and CD8$^{-}$CD4$^{+}$CD25$^{-}$Foxp3$^{+}$. For analysis of naïve and activated T$_{reg}$ cells, LN homogenates were stained with a cocktail of antibodies against CD4-PE, CD25-PECy7, CD44-PB, and CD62L-FITC. Cells were then fixed and permeabilized using the Foxp3/Transcription Factor Staining Buffer Set, and subsequently stained with Foxp3-APC (1/100 dilution) or its isotype control (1/100 dilution). Naïve T$_{reg}$ cells were identified as CD4$^{+}$CD25$^{+}$Foxp3$^{+}$CD44$^{-}$CD62L$^{+}$and CD4$^{+}$CD25$^{-}$Foxp3$^{+}$CD44$^{-}$CD62L$^{+}$. Activated T$_{reg}$ cells were identified as CD4$^{+}$CD25$^{+}$Foxp3$^{+}$CD44$^{+}$CD62L$^{-}$ and CD4$^{+}$CD25$^{-}$Foxp3$^{+}$CD44$^{+}$CD62L$^{-}$ (T$_{mem/eff}$). For analysis of T$_{FH}$ cells and total PD1$^{+}$ T cells, splenic and LN homogenates were stained with a cocktail of antibodies against CD4-APC-Cy7, CD44-PE-Cy7, CD62L-APC, PD1-PB, and CXCR5-FITC (1/400 dilution). T$_{FH}$ cells were defined as CD4$^{+}$CD44$^{+}$CD62L$^{-}$CXCR5$^{+}$PD1$^{+}$and total PD1$^{+}$ T cells as CD4$^{+}$PD1$^{+}$.

Splenic and LN homogenates from 4–5 months old male and female $Ldlr^{-/-}$ and $T\text{-}Abc^{dko}Ldlr^{-/-}$ mice were analyzed for the exhaustion markers CTLA4, TIM3, LAG3, and Eomes. Tissue homogenates were stained with a cocktail of antibodies against CD4-APC, CD8-FITC, and CTLA4-PB or with a cocktail of antibodies against TCRβ-PerCPCy5.5, CD4-FITC, CD8-PE, and TIM3-APC or LAG3-APC. Alternatively, tissue homogenates were stained with a cocktail of antibodies against TCRβ-PB, CD8-APC, and CD4-FITC, then fixed and permeabilized, and subsequently stained with Eomes-PE (1/100 dilution) or its isotype control (1/100 dilution). CTLA4, TIM3, and LAG3 mean fluorescence intensity (mfi) on CD4$^{+}$and CD8$^{+}$T cells, and Eomes mfi in CD8$^{+}$T cells was assessed by flow cytometry.

For analysis of Tbet$^{+}$ cells, LN homogenates from female $Ldlr^{-/-}$ and $T\text{-}Abc^{dko}Ldlr^{-/-}$ mice fed a chow diet for 28 weeks or a WTD for 10 weeks were stained with a cocktail of antibodies against TCRβ-PB, CD4-APC, and CD8-FITC, then fixed and permeabilized, and subsequently stained with Tbet-PE (1/100 dilution) or its isotype control (1/100 dilution). Tbet$^{+}$cells were identified as TCRβ$^{+}$CD4$^{+}$Tbet$^{+}$and TCRβ$^{+}$CD8$^{+}$Tbet$^{+}$. For analysis of naïve and activated Tbet$^{+}$cells, LN homogenates were stained with a cocktail of antibodies against TCRβ-PB, CD8-PE, CD44-PECy7, and CD62L-APC, then fixed and permeabilized, and subsequently stained with Tbet-AF488 (1/100 dilution) or its isotype control (1/100 dilution). Naive Tbet$^{+}$cells were identified as TCRβ$^{+}$CD4$^{+}$Tbet$^{+}$CD44$^{-}$CD62L$^{+}$and TCRβ$^{+}$CD8$^{+}$Tbet$^{+}$CD44$^{-}$CD62L$^{+}$. Activated Tbet$^{+}$cells were identified as TCRβ$^{+}$CD4$^{+}$Tbet$^{+}$CD44$^{+}$CD62L$^{-}$, TCRβ$^{+}$CD8$^{+}$Tbet$^{+}$CD44$^{+}$CD62L$^{-}$ (T$_{mem/eff}$) and TCRβ$^{+}$CD8$^{+}$Tbet$^{+}$CD44$^{+}$CD62L$^{+}$(T$_{CM}$) cells.

All samples were analyzed on LSRII (BD Biosciences), running FACSDiVa software (BD Biosciences). The data were analyzed using the FlowJo software (FlowJo).

**In vitro T cell proliferation assay**. Splenic T cells from male and female control, $T\text{-}Abc^{dko}$, $Ldlr^{-/-}$, and $T\text{-}Abc^{dko}Ldlr^{-/-}$ mice (4–5 or 12–13 months old), young (3 months) and aged (24 months) wild-type male and female mice, and total aortic cells from middle-aged (12–13 months) male and female $Ldlr^{-/-}$ and $T\text{-}Abc^{dko}Ldlr^{-/-}$ mice were isolated and labeled with 10 µM CFSE-PB (Invitrogen) and stimulated with Mouse T-Activator CD3/CD28 Dynabeads (ratio beads to cells 1:2 for splenic cells and 10:1 for aortic cells) in RPMI 1640 medium supplemented with 10% FCS, 1% pen-strep, and 50 µM β-mercaptoethanol at 37 °C, 5% CO$_2$. Splenic T cells from $Ldlr^{-/-}$ and $T\text{-}Abc^{dko}Ldlr^{-/-}$ mice were also incubated with 50 µg/mL rHDL for the whole duration of the proliferation assay. Cells were incubated at 37 °C for 72 h (splenic) or 96 h (aortic). Subsequently, beads were removed, T cells were stained using CD4 and CD8 antibodies as described above, and CFSE-PB dilution was assessed by flow cytometry. Using the Proliferation Modeling tool in the FlowJo software, histograms of CFSE dilution in CD4$^{+}$ and CD8$^{+}$ T cells were obtained where each cell division was depicted as a separate peak. The number of CD4$^{+}$ and CD8$^{+}$ T cells in each division was automatically calculated by the software after designing gates for each peak. To calculate the percentage of CD4$^{+}$ and CD8$^{+}$ T cells in each division, the number of CD4$^{+}$ and CD8$^{+}$ T cells in one division was divided by the total number of CD4$^{+}$ and CD8$^{+}$ T cells in all divisions, respectively.

**Western blot of JNK and gasdermin D**. Splenic CD4$^{+}$ and CD8$^{+}$ T cells from 4–5 months old male and female $Ldlr^{-/-}$ and $T\text{-}Abc^{dko}Ldlr^{-/-}$ mice were isolated as described above. Cell culture conditions were the same as described above. To assess JNK1/2 phosphorylation, ~1.5 million CD4$^{+}$ and CD8$^{+}$ T cells were stimulated with 5 µg/mL αCD3 (Biolegend) for 0, 5, and 10 min. To assess gasdermin D cleavage, ~2 million CD4$^{+}$ T cells were stimulated with 5 µg/mL plate-bound αCD3 and 100 U/mL IL-2 (Peprotech) for 12 h. After stimulation, cells were harvested and lysed in radio immuno precipitation assay (RIPA) buffer (1% IGEPAL, 0.1% SDS, and 0.5% sodium deoxycholate in PBS) supplemented with complete protease inhibitor cocktail (Sigma-Aldrich), and phosphatase inhibitor cocktail 2 and cocktail 3 (Sigma-Aldrich). Cell lysates were separated by SDS-PAGE gel electrophoresis and immunoblotted with primary antibodies against total JNK1/2 (1:1000; Cell Signaling Technology), phosphorylated JNK1/2 (1:1000; Cell Signaling Technology), or gasdermin D (1:1000; a kind gift from Genentech). Heat shock

protein 90 (HSP90) (1:1000; Cell Signaling Technology) was used as loading control. Goat anti-mouse (1:2000; Invitrogen; for phosphorylated JNK1/2), goat anti-rabbit (1:2000; Invitrogen; for total JNK1/2 and HSP90), and goat anti-rat (1:1000; Cell Signaling Technology; for gasdermin D) HRP-conjugated secondary antibodies were used. Blots were developed using the Amersham Imager 600 (GE Healthcare) and bands were quantified using ImageJ software (NIH). Uncropped and unprocessed scans of blots are shown in the Source Data file.

**In vitro T cell apoptosis assays.** Splenic CD4$^+$ and CD8$^+$ T cells from 4–5 months old male and female $Ldlr^{-/-}$ and $T\text{-}Abc^{dko}Ldlr^{-/-}$ mice were isolated and labeled using Caspase3/7 Red (APC) Reagent (Sartorius). Cells were cultured as described above, and stimulated with 5 µg/mL plate-bound αCD3 and 100 U/mL IL-2 for 12 h. For Incucyte experiments, images were taken every 2 h. For image analysis, background signal was removed using the Incucyte ZOOM software (Sartorius) and the number of total cells and cleaved caspase3/7$^+$ cells per image were quantified using ImageJ software (NIH) in a blinded fashion, *i.e.* the observer was unaware of the genotypes. Alternatively, splenic T cells from 4–5 month old male and female control ($Ldlr^{+/+}$), $T\text{-}Abc^{dko}$, $T\text{-}Abca1^{ko}Ldlr^{-/-}$ and $T\text{-}Abcg1^{ko}Ldlr^{-/-}$ mice, middle-aged (12–13 months) male and female $Ldlr^{-/-}$ and $T\text{-}Abc^{dko}Ldlr^{-/-}$ mice, and young (3 months) and aged (24 months) wild-type male and female mice were isolated and labeled using caspase3/7 reagent (1/1000 dilution), stimulated as described for the Incucyte assay, harvested, stained with CD4-PB (1/200 dilution) or CD8-PB (1/200 dilution), and analyzed by flow cytometry for cleaved caspase 3/7 mfi in splenic CD4$^+$ or CD8$^+$ T cells. Alternatively, total aortic cells from middle-aged (12–13 months) female $Ldlr^{-/-}$ and $T\text{-}Abc^{dko}Ldlr^{-/-}$ mice were isolated as described above and stimulated with Mouse T-Activator CD3/CD28 Dynabeads (ratio beads to cells 10:1) in RPMI 1640 medium supplemented with 10% FCS, 1% pen-strep, and 50 µM β-mercaptoethanol for 96 h. Beads were removed and T cells were stained with caspase3/7 reagent for 1 h at 37 °C. Cells were then centrifuged, washed, resuspended in FACS buffer, and stained with a cocktail of antibodies against CD4-FITC (1/200 dilution), CD44-PE-Cy7 (1/200 dilution), and CD62L-PB (1/200 dilution), and analyzed by flow cytometry. Cleaved caspase3/7 mfi in aortic CD4$^+$ or CD4$^+$T$_{mem/eff}$ T cells, and the percentages of aortic CD4$^+$T$_{mem/eff}$ cells were assessed by flow cytometry.

**In vitro T$_{reg}$ differentiation assay.** Splenic T cells from 4–5 months old male and female $Ldlr^{-/-}$ and $T\text{-}Abc^{dko}Ldlr^{-/-}$ mice were isolated and stimulated with Mouse T-Activator CD3/CD28 Dynabeads (ratio beads to cells 1:2) in the presence of 100 U/mL IL-2 and 5 ng/mL TGF-β1 (Peprotech) for 72 h. Then, beads were removed, T cells were stained using CD4, CD25, and Foxp3 antibodies as described above, and samples were analyzed by flow cytometry.

**FAS and Bcl2 assays.** For all stainings, antibodies listed below were used at a 1/200 dilution, unless stated otherwise. Splenic T cells from 4–5 months old male and female $Ldlr^{-/-}$ and $T\text{-}Abc^{dko}Ldlr^{-/-}$ mice were isolated as described above and stimulated as described for the Incucyte caspase 3/7 assay. Cells were harvested and stained with a cocktail of antibodies against CD4-PB, CD8-PE, and FAS-AF488 (for FAS analysis), or CD4-PB (for Bcl2 analysis in CD4$^+$ T cells), or TCRβ-PB, CD8-PE, CD44-PE-Cy7, and CD62L-APC (for Bcl2 analysis in CD8$^+$ T cells). For Bcl2 analysis, cells were then fixed and permeabilized, and subsequently stained with Bcl2-FITC (1/40 dilution) or its isotype control (1/40 dilution). Samples were analyzed by flow cytometry. FAS mfi on CD4$^+$ and CD8$^+$ T cells was assessed. Percentages of CD4$^+$Bcl2$^+$, CD4$^+$Bcl2$^-$, CD8$^+$Bcl2$^+$, CD8$^+$Bcl2$^-$, CD8$^+$T$_{CM}$Bcl2$^+$, and CD8$^+$T$_{CM}$Bcl2$^-$, CD8$^+$T$_{mem/eff}$Bcl2$^+$, and CD8$^+$T$_{mem/eff}$Bcl2$^-$ T cells were quantified.

**Mitochondrial ROS assay.** Splenic T cells from 4–5 months old male and female $Ldlr^{-/-}$ and $T\text{-}Abc^{dko}Ldlr^{-/-}$ mice were isolated as described above and stimulated as described for the Incucyte caspase 3/7 assay. Cells were harvested and stained with 2 µM MitoSOX Red Mitochondrial Superoxide Indicator or 50 nM MitoTracker Green FM (Invitrogen; carbocyanine-based dye) for 30 min at 37 °C. For all stainings, antibodies were used at a 1/200 dilution. For MitoSOX (eBioscience) analysis, cells were stained with a cocktail of antibodies against CD44-PE-Cy7, CD62L-APC, CD4-PB or CD8-PB. For MitoTracker analysis, cells were stained with a cocktail of antibodies against CD44-PB, CD62L-APC, CD4-PE or CD8-PE. MitoSOX and Mitotracker mfi on CD4$^+$and CD8$^+$T$_{naive}$, CD4$^+$and CD8$^+$T$_{mem/eff}$, and CD8$^+$T$_{CM}$ cells was assessed by flow cytometry. To correct the levels of mitochondrial ROS for mitochondrial mass, the MitoSOX/Mitotracker ratio was calculated.

**Analysis of T cell granzyme B, IFN-γ, and LAMP-1.** Splenic CD4$^+$ and CD8$^+$ T cells from 4–5 months old male and female $Ldlr^{-/-}$ and $T\text{-}Abc^{dko}Ldlr^{-/-}$ mice, and young (3 months) and aged (24 months) wild-type male and female mice were isolated as described above and stimulated as described for the Incucyte caspase 3/7 assay. Cells were harvested and stained with a cocktail of antibodies against CD4-PB (1/200 dilution) and CD8-APC (1/200 dilution), and in some assays LAMP-1-FITC (1/200 dilution), or subsequently fixed and permeabilized, and stained with granzyme B-FITC (1/100 dilution) or its isotype control (1/100 dilution), or IFN-γ-FITC (1/100 dilution). Samples were analyzed by flow cytometry. LAMP-1 surface

expression on CD4$^+$ and CD8$^+$ T cells was assessed as mfi. Percentages of CD8$^+$granzyme B$^+$, CD4$^+$IFN-γ$^+$, and CD8$^+$IFN-γ$^+$T cells were quantified. For IFN-γ analysis, IFN-γ$^+$T cells were selected based on the fluorescence minus one (FMO) control.

**IFN-γ secretion.** Splenic T cells from 4–5 months old male and female $Ldlr^{-/-}$ and $T\text{-}Abc^{dko}Ldlr^{-/-}$ mice, and young (3 months) and aged (24 months) wild-type male and female mice were isolated as described above and stimulated as described for the Incucyte caspase 3/7 assay Medium was collected after stimulation and the concentration of IFN-γ was determined by ELISA (R&D Systems).

**T cell mediated cytotoxicity.** Bone marrow (BM) cells were isolated from the femurs and the tibias of young (3 months) wild-type mice and were cultured at 37 °C, 5% CO$_2$ in DMEM (Gibco) supplemented with 10% FCS, 1% pen strep, and 20% L-cell conditioned medium to induce differentiation into BM derived macrophages (BMDMs).
    Splenic total T cells from 4–5 months old male and female $Ldlr^{-/-}$ and $T\text{-}Abc^{dko}Ldlr^{-/-}$ mice were isolated as described above. To study the effect of $Abca1/Abcg1$ deficiency on T cell mediated cytotoxicity on macrophages, T cells, while being stimulated with 5 µg/mL αCD3 and 100 U/mL IL-2, were incubated with BMDMs (T cell to BMDM ratio 6:1). After 24 h, medium was collected and macrophage death as a measure for T cell mediated cytotoxicity, was assessed by quantifying endogenous lactate dehydrogenase (LDH) release using the CyQUANT LDH Cytotoxicity Assay kit (Invitrogen) according to the manufacturer's instructions. To correct for LDH release from T cells, concomitantly T cells from the same mice were stimulated with αCD3 and IL-2 for 24 h, LDH release in medium was measured, and subtracted from levels of LDH in medium from co-cultures.

**In vitro macrophage bacteria killing assay.** BMDMs from young (3 months) wild-type mice were generated as described above. Splenic CD8$^+$ T cells from 4–5 months old male and female $Ldlr^{-/-}$ and $T\text{-}Abc^{dko}Ldlr^{-/-}$ mice were isolated as described above and stimulated as described for the Incucyte caspase 3/7 assay. Then, CD8$^+$ T cell medium was collected. Subsequently, BMDMs were cultured at 37 °C, 5% CO$_2$ in 100 µL DMEM supplemented with 10% FCS, 1% pen strep, and 500 µL CD8$^+$ T cell conditioned medium. As a positive control, BMDMs were incubated with 600 µL DMEM supplemented with 10% FCS and 1% pen strep containing recombinant IFN-γ (420 U/mL; Peprotech). After 3 days, medium was aspirated, BMDMs were washed and co-incubated with *E. coli* K-12 MG1655 bacteria (BMDM to *E.coli* ratio 50:1; kind gift from Dr. Marjon G.J. de Vos, University of Groningen, the Netherlands) in DMEM supplemented with 10% FCS, without antibiotics for 1 h. After another wash, BMDMs were incubated in DMEM supplemented with 10% FCS and 200 µg/mL gentamicin (Gibco) for 1 h, washed for removal of dead bacteria, and incubated in DMEM supplemented with 10% FCS and 20 µg/mL gentamicin for 2 h. BMDMs were then washed extensively (three times) and lysed in 0.1% Triton X-100 in PBS. To assess bacterial viability, BMDM cell lysates were plated on agar and incubated o/n at 37 °C. Bacterial colony forming units (CFU) were counted manually in a blinded fashion, *i.e.* the observer was unaware of the genotypes, and were quantified for macrophages incubated with conditioned medium from $Ldlr^{-/-}$ or $T\text{-}Abc^{dko}Ldlr^{-/-}$ CD8$^+$ T cells. The number of bacterial CFU directly reflects the macrophage killing capacity (i.e. the lower the number of CFU, the higher the macrophage killing capacity).

**Plasma lipid and lipoprotein analyses.** Blood samples were collected by tail bleeding. Plasma was separated by centrifugation and assayed for cholesterol and TG using enzymatic kits (Roche and Diasys Diagnostic Systems, respectively). The Cholesterol standard FS (DiaSys Diagnostic Systems) and Precimat Glycerol (Roche) were used for the cholesterol and TG calibration curves, respectively. To assess lipoprotein cholesterol distribution in plasma from female $Ldlr^{-/-}$ and $T\text{-}Abc^{dko}Ldlr^{-/-}$ mice fed a WTD (10 weeks) by fast performance liquid chromatography (FPLC), a system containing a PU-980 pump with a linear degasser and a UV-975 UV/VIS detector (Jasco) was used. Pooled plasma ($n = 15$ per pool) was injected onto a Superose 6 HR 10/300 GL column (GE Healthcare) and eluted at a constant flow rate of 500 µL/min in PBS (pH 7.4). Plasma fractions were assayed for cholesterol using an enzymatic kit as described above. To assess lipoprotein cholesterol and TG distribution in plasma from female $Ldlr^{-/-}$ and $T\text{-}Abc^{dko}Ldlr^{-/-}$ mice fed a chow diet (28 weeks), young (3 months) and aged (24 months) male and female wild-type mice, and lipoprotein TG distribution in female $Ldlr^{-/-}$ and $T\text{-}Abc^{dko}Ldlr^{-/-}$ mice fed a WTD (10 weeks) by FPLC, a system containing a PU-4180 pump with a linear degasser and a UV-4075 UV/VIS detector (Jasco) was used. Pooled plasma ($n = 10$–18 per pool) was injected onto a Superose 6 Increase 10/300 GL column (GE Healthcare) and eluted at a constant flow rate of 0.31 mL/min in PBS (pH 7.4). Cholesterol and TG were measured in line by the addition of cholesterol and TG reagents, respectively, at a constant flow rate of 0.1 mL/min using an additional PU-4080i infusion pump (Jasco). Data acquisition and analysis were performed using ChromNav software (Jasco).

**Atherosclerotic lesion analysis.** Female $Ldlr^{-/-}$ and $T\text{-}Abc^{dko}Ldlr^{-/-}$mice were fed a WTD for 10 weeks (starting at 8–12 weeks of age; $n = 16$ $Ldlr^{-/-}$ and $n = 16$ $T\text{-}Abc^{dko}Ldlr^{-/-}$mice) or a chow diet for 28 weeks ($n = 18$ $Ldlr^{-/-}$ and $n = 16$

$T\text{-}Abc^{dko}Ldlr^{-/-}$ mice) or 12–13 months ($n = 11$ $Ldlr^{-/-}$ and $n = 10$ $T\text{-}Abc^{dko}Ldlr^{-/-}$ mice). After the indicated period of time on diet, mice were sacrificed and the hearts were isolated and fixed in phosphate-buffered formalin. Hearts were dehydrated and embedded in paraffin, and cross-sectioned throughout the aortic root area (4 μm sections). Haematoxylin-eosin (H&E) staining was performed on the sections and the average from 6 sections (40 μm distance between sections) for each animal was used to determine lesion size. Lesion size was quantified by morphometric analysis. Necrotic core area was determined as acellular area, lacking nuclei and cytoplasm, under the fibrous cap of lesions from H&E stained sections. Necrotic core area was differentiated from regions of dense fibrous scars by the presence of cell debris, as reported previously[81]. To assess the number of T cells per aortic root section, for antigen retrieval, paraffin sections were incubated in 10 mM citrate buffer at pH 6.0 (15 min; microwave), and blocked in 10% goat serum for 30 min at RT. Subsequently, sections were incubated o/n with rabbit anti-human CD3 antibody at 4 °C (1/250 dilution; Dako), which also cross-reacts with mouse, and subsequently with biotinylated goat anti-rabbit secondary antibody for 30 min at RT (1/250 dilution; Vector Laboratories), and Vectastain Elite ABC Kit-peroxidase kit (Vector Laboratories) according to the manufacturer's instructions. Sections were then stained with 3,3′-diaminobenzidine (DAB) for 10 min at RT and counterstained with haematoxylin. T cells were identified as cells with brown plasma membrane CD3 staining and the hematoxylin staining still visible. To assess fibrous cap thickness and collagen content in atherosclerotic lesions, aortic root sections (one per mouse) taken from similar regions of the lesions (mouse to mouse) were stained with Weigert's haematoxylin for 8 min to stain the nuclei and subsequently with Sirius Red (0.1% (w/v)) in 1.3% aqueous picric acid solution for 1 h at RT. Slides were then washed twice 10 sec in acidified water (1% glacial acetic acid solution). Fibrous cap thickness was measured in the largest section at even intervals 2 μm apart. Then the average thickness of lesion was reported in length units, as described[82]. Total collagen area (whole area staining positive for Sirius Red) was also measured. To assess SMC area, sections were blocked in 10% goat serum for 30 min at RT, incubated o/n at 4 °C with rabbit anti-mouse α-SMA primary antibody (1/200 dilution; Lab Vision) and subsequently with biotinylated goat anti-rabbit secondary antibody for 30 min at RT (1/250 dilution), and Vectastain ABC-peroxidase kit (Vector Laboratories) according to the manufacturer's instructions. Sections were then stained with DAB for 10 min at RT and counterstained with haematoxylin. SMC positive area was assessed. To assess Mac-2 (Galectin-3) area, mainly reflecting macrophages, for antigen retrieval, paraffin sections were incubated with proteinase K for 9 min at RT (20 μg/mL in PBS; Merck) (immunohistochemistry (IHC)) or Tris-Base EDTA at pH 9.0 (15–20 min; microwave) (immunofluorescence (IF)), and blocked in 10% goat serum for 30 min at RT. Sections were then incubated o/n at 4 °C with rat anti-mouse/human Mac-2 (Galectin-3) primary antibody (1/10000 dilution; Cedarlane). For IHC, sections were subsequently stained with biotinylated goat anti-rat secondary antibody for 30 min at RT (1/125 dilution) and Vectastain ABC-peroxidase kit according to the manufacturer's instructions, followed by staining with DAB for 10 min at RT and counterstaining with haematoxylin. For IF, sections were stained with anti-rat IgG-AF488 secondary antibody for 30 min at RT (1/200 dilution; Invitrogen) and mounted using ProLong Gold Antifade Mountant with DAPI (Invitrogen). To assess apoptotic macrophages, sections were stained with the In Situ Cell Death Detection Kit TMR red (Roche). After perm wash (BD Biosciences), slides were incubated with proteinase K for 9 min at RT (20 μg/mL in PBS), and further staining was performed according to the manufacturer's instructions, in combination with Mac-2 staining as described above. Nuclei were stained with Hoechst 33342 (Invitrogen) for 10 min at RT and sections were mounted using VECTA-SHIELD Antifade Mounting Medium. For IF, sections were imaged using an AxioObserver Z1 compound microscope (10x objective; AxioCam MRm3 CCD camera; Carl Zeiss) running Zen software. The total area of lesions (based on H&E staining), fibrous cap thickness, total collagen and SMC positive areas, Mac-2+ area and TUNEL+Mac-2+ area were assessed using ImageJ software (NIH). Quantification of all parameters was performed in a blinded fashion, i.e. the observer was unaware of the genotypes.

**Quantification and statistical analysis**. All data are presented as means ± SEM. In each experiment, n defines the number of mice included. The statistical parameters (n, mean, SEM) can be found within the figure legends. Two-tailed Student's t test was used to define differences between two datasets. To define differences between three or four datasets, One-way Analysis of Variance (ANOVA) was used with a Bonferroni multiple comparison post-test. The criterion for significance was set at $p < 0.05$. Statistical analyses were performed using GraphPad Prism 9 (San Diego, California).

**Reporting Summary**. Further information on research design is available in the Nature Research Reporting Summary linked to this article.

## Data availability

The authors declare that the data supporting the findings of this study are available within the paper and its supplementary information files. All the raw data generated in this study are provided in the Source Data file. Source data are provided with this paper.

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

## Acknowledgements

This work is supported by VIDI (917.15.350) and Aspasia grants from the Netherlands Organization of Scientific Research (NWO) (M.W.), and a Rosalind Franklin Fellowship from the University of Groningen with EU Co-Fund attached (M.W.).

## Author contributions

V.B.: conceived, designed, and performed experiments, carried out data analyses, and wrote the original draft of the manuscript. A.M.L.R., B.H., A.G.G., A.M.-G., A.T.P.: performed experiments and carried out data analyses. S.M., F.B., R.d.B., D.D.: performed experiments, carried out data analyses, and provided reagents. E.G., A.F.-S., M.H.K., N.J.K., R.H., M.L.-M.: performed experiments. A.d.B., B.v.d.S., A.K., L.Y.-C., G.v.d.B.: provided intellectual input and reviewed the original and revised manuscript. M.W.: conceived, designed, and performed experiments, carried out data analyses, revised the original draft of the manuscript, supervised the study, and obtained funding for the study.

## Competing interests

The authors declare no competing interests.
