## [Peer Review File · Nature Communications]

T cell cholesterol efflux suppresses apoptosis and senescence and increases atherosclerosis in middle aged miceREVIEWER COMMENTS

Reviewer #1 (Remarks to the Author):

In this manuscript the authors make t cell specific ABCA1/G1 knockouts and study the link between cholesterol accumulation in several varieties of t cells and their dysfunction. Several points below should be considered by the authors to strengthen their conclusions.

1. Much of what is contained in Figs 1-3 have been covered in other labs published work (Fig 1 and 2 -redundant) and maybe should be in supplemental data.
2. the extreme phenotype of tissue specific A1/G1 deletion has been used extensively yet, the role each plays is still debated. before mechanistic work can move forward, it seems like a good idea to tease the importance of these two apart since this would be valuable information in the field of cholesterol efflux and intra-cellular flux.
3. Dysfunctional T cells in Fig 4 and 5 are characterized by surface stains but functional assays are essential. Fig 5 showing young and aged proliferation is subtle and should be followed up with a functional assay as well.
4. Fig 6 showing premature T cell aging as measure by surface stain and proliferation (CFSE) is interesting but a loss and/or gain of function with loading cholesterol or cyclodextrin would be more convincing.
5. Figs 7 and 8 are interesting since it appears that it goes against what one might expect (greater athero) however, the big question of how dysfunctional the T cells are remain and the question of auto-immunity or B cell alternation is still open.

Reviewer #2 (Remarks to the Author):

The report by Bazioti et al. entitled "T-cell Abca1 and Abcg1 cholesterol efflux pathways suppress T-cell apoptosis and senescence and increase atherosclerosis in middle-aged Ldlr^{-/-} mice" is interesting. The authors show that the combined absence of cholesterol transporters Abca1 and Abcg1 in T cells alters plasma cholesterol accumulation, T cell activation and apoptosis promoting an aged T cell phenotype affecting atherosclerosis progression in middle aged mice. Overall, the experiment performed are convincing. I have some specific questions suggestions to strengthen the manuscript.

Specific instances are outlined in the response to the authors.

1. Most of the experiments are performed in whole body Ldlr^{-/-} background (to induce atherosclerosis). It is unclear the influence of lack of Ldlr expression in T cells in the phenotype observed both in vitro and in vivo. Is the same phenotype observed in T cells in vivo, in a model of induction of atherosclerosis using Adenoviral-hepatic mediated expression of Pcsk9 in mice which results in a low-density lipoprotein receptor knockout phenotype by only affecting hepatic expression and thus the T cells express normal levels of LDR?
2. It is unclear, why plasma cholesterol levels and distribution in different lipoproteins is altered in mice lacking ABCA1 and ABCG1 when fed a WD. This might explain why there are not differences in plaque size in WT and ABCA1/G1 deficient mice fed a Western diet. The authors should perform FPLC analysis and assess TC, TAG and HDL in aged Ldlr^{-/-} and Ldlr-Abca1/g1 fed a chow diet.
3. The manuscript will benefit from a more detailed characterization of atherosclerotic lesions. For instance, since T-cell activation influences macrophage apoptosis. The authors should analyze number of apoptotic macrophages and in addition of plaque necrotic area quantification of fibrous cap thickness. This analysis is important for mice fed a CD for 12 months.
4. Is the efferocytotic capacity of macrophages, to remove apoptotic T-cells, influenced by

different T cell membrane cholesterol content?

5. The authors have performed a very comprehensive analysis of T-cell in circulation, spleen, etc... However, the impact of ABCA1/G1 deficiency in T-cells within the plaque which is the more relevant aspect of the manuscript is somehow poor. The authors rely on para-aortic LN analysis to evaluate T-cells in the aorta, which could be a confounding measurement. This could be improved using flow cytometry analysis in digested aortas and/or sc-RNA Seq. This analysis will provide a better understanding of how both transporters influence T-cell activation, proliferation and apoptosis in T-cells accumulated in atherosclerotic plaques.

Reviewer #3 (Remarks to the Author):

Bazioti and co-workers present a very interesting manuscript about the role of cholesterol efflux on T cell-mediated responses, and their consequences to the development of atherosclerosis. To this end, the investigators have produced a T-cell-specific Abca1/Abcg1 knockout strain that was used for the characterization of T-cell responses; the same strain was backcrossed into LdlrKO mice to evaluate the effects of Abca1/Abcg1 deficiency on T cells in atherogenesis. Of note, the investigators also attempted to parallel the consequences of efflux deficiency with phenotypic changes that may occur during aging.

The study presented a clear hypothesis, and the manuscript is quite well written. The study design looked to a great extent appropriated, particularly regarding the characterization of the consequences of the combine deficiency of Abca1/Abcg1 on intracellular and membrane cholesterol, as well as the repertoire of CD4+ and CD8+ T cells in the targeted mice. Nevertheless, although interesting, the attempt to explore both aging and atherogenesis made the overall story sometimes a bit confusing and difficult to follow. Thus, several questions should be taken into consideration by the authors for revising the manuscript:

- One of the initial findings shown was the fact that T-cell Abca1/Abcg1 deficiency led to intracellular and plasma membrane cholesterol accumulation (Fig1). Although this could be expected, it raises the question on how cholesterol (lipoprotein) uptake is handled on these cells. Regulated by the accumulation of esterified cholesterol, Srbp2 over-expression should downregulate LDL receptor, which could reduce or even up lipid accumulation. Considering cells originated from LdlrKO, is that the main reason for the observed phenotype? How relevant is efflux on T cells that can upregulate LDLR?

- The investigators performed extensive flow cytometry analysis of the repertoire of T cells in T-AbcdkoLdlr-/- and Ldlr-/- mice. However, the presentation of data is a bit intriguing by showing the “%” data as well as the “% corrected by total number of cells”. Why percentages need to be corrected? What is “total number of cells”? the cells isolated from the organs? Cells analyzed? Were whole organs analyzed? In addition to not been described in methods, this approach seems unnecessary and perhaps wrong, which could change slightly some conclusions. The same criticism applies to the subset analysis of T cells in the atherosclerosis experiment, which should be clarified.

- In the end of the last paragraph of the section “T-cell Abca1/Abcg1 Deficiency Increases T-cell Apoptosis Upon TCR Stimulation”, the authors stated: “Collectively, these data indicate that T-cell Abca1/Abcg1 deficiency enhances T-cell apoptosis downstream of TCR and CD25, mediated by Bcl2 in the intrinsic apoptosis pathway, especially in Tmemory/effector cells. This likely accounts for the ~50% decrease in peripheral T-cells in T-cell Abca1/Abcg1 deficiency”. Were other modalities of cell death evaluated? How can the authors be so sure this is the main pathway?

- The results under the section “Aging Increases Cholesterol Accumulation and Apoptosis in T-cells from Wild-Type Mice” are to some extent puzzling. Although the authors could build some nice parallels between the phenotype of aged C57BL6 mice and T-AbcdkoLdlr-/-, it seems very

speculative that only efflux defects could explain aging-driven phenotypic changes. How to reconcile that fact that in the current study, a comparison is made between a WT strain, which has HDL as the main circulating lipoprotein, against LdlrKO, which has VLDL and LDL? In this context, did the T cells from aged WT mice really had an accumulation of cholesterol intracellularly and in the plasma membrane? Fig.5a is not very convincing. How is the expression status of Abca1, Abcg1, Ldlr on T cells upon aging?

- A few times in the manuscript, the authors indicate "data not shown", which can generate some credibility doubts. Unless there would be editorial limits from the journal, and if relevant to the story, these data should be incorporated to the manuscript, for example on supplements.

- Fig.8 showed that T-AbcdkoLdlr/- presented reduced atherosclerotic burden compared to Ldlr/-. Unexpectedly, an increased percentage of Mac2 positive cells was observed in T-AbcdkoLdlr/- mice. mRNA profiling showed reduced Mcp1 expression in the same group, how to reconcile the 2 previous findings? Additionally, the increased percentage of macrophages in T-AbcdkoLdlr/- group seems counterintuitive considering the finding of smaller lesions; it would be interesting to evaluate M1 and M2 markers, e.g. on mRNA levels and check whether the latter polarization dominated the tissue.

- Fig.8 and Supp.Fig. 6 show quantification and representative pictures of lesions that were stained with H&E. Why was Oil Red O staining of lipids not used instead? It looks odd that a study investigating cholesterol-related mechanisms does not explore/show the lipid content on plaques. Moreover, representative H&E pictures suggest some differences on cellularity between genotypes, and perhaps also chow and diet protocols (Fig.8 and Supp.Fig.6) ; the study would benefit of further characterization of plaques, e.g. SMA staining.

- A minor point, in the discussion, some conclusions/suggestions from the current study are finalized with a citation, which looks confusing. For example, page 11 of the Word file, "We suggest that the decreased inflammation may be the consequence of uptake of apoptotic T-cells by myeloid cells in the aorta, which is an anti-inflammatory process 65". If this is a statement from the authors, it should not cite other work. This happens again in the subsequent paragraph, and such faults should be revised.

Response to Reviewers (*Responses in Italics*)

Reviewer #1: In this manuscript the authors make T cell specific ABCA1/G1 knockouts and study the link between cholesterol accumulation in several varieties of T cells and their dysfunction. Several points below should be considered by the authors to strengthen their conclusions.

We thank the reviewer for the suggestions to strengthen the conclusions of our manuscript.

1. Much of what is contained in Figs 1-3 have been covered in other labs published work (Fig 1 and 2 -redundant) and maybe should be in supplemental data.

We partially agree with the reviewer. We previously generated a mouse model with T-cell Abca1 and Abcg1 deficiency under control of the LckCre promoter that we used for studies on autoimmunity, as published¹¹, and distributed to Dr. Swirski for studies on T-cell maturation in the thymus¹². The CD4Cre promoter shows a higher level of recombination with loxP sites in mature T-cells than the LckCre promoter^{8,13-15}. Since we studied the role of T-cell cholesterol efflux pathways in atherogenesis, where mature T-cells play a major role, we generated CD4CreAbca1^{fl/fl}Abcg1^{fl/fl} mice for our studies. These mice have not been generated before, and therefore characterization in terms of Abca1/Abcg1 transporter expression, cholesterol, and cholesteryl ester (CE) accumulation in T-cells was required (Figure 1 of the previous version of the manuscript). As per the suggestion of the reviewer we have moved Figure 1 on the characterization of CD4CreAbca1^{fl/fl}Abcg1^{fl/fl} mice to Supplementary Figure 1.

We feel that Figure 2 and 3 show data that are instrumental to the understanding of the paper and therefore need to be in the main text, now shown as Figure 1 and 2. Of these, current Figure 1 shows that deficiency of Abca1/Abcg1 in T-cells increases membrane stiffening, which has not been studied directly before in the setting of Abca1 and Abcg1 deficiency. These data indicate a link between membrane cholesterol accumulation and membrane stiffening that may explain effects on downstream T-cell receptor (TCR) signaling. We therefore prefer to keep current Figure 1 within the main data.

As for current Figure 2, none of these data have been shown previously in the context of T-cell Abca1/Abcg1 deficiency, except for data in Fig 2d-e that show a decrease in splenic CD4⁺ and CD8⁺ T-cells in CD4CreAbca1^{fl/fl}Abcg1^{fl/fl}Ldlr^{-/-} compared to Abca1^{fl/fl}Abcg1^{fl/fl}Ldlr^{-/-} mice. These data share similarities with findings in the Swirski paper comparing LckCreAbca1^{fl/fl}Abcg1^{fl/fl} to Abca1^{fl/fl}Abcg1^{fl/fl} mice, even though in the Swirski paper the decrease in splenic CD4⁺ T-cells was not observed at both timepoints¹², perhaps consistent with a lower degree of recombination between LckCre and loxP sites in peripheral CD4⁺ T-cells as described above. We prefer to keep Fig 2d-e within the main data, in order to show effects of T-cell Abca1/Abcg1 deficiency on T-cells in peripheral tissues (spleen, lymph nodes (LNs), blood) together in the same figure.

We have added a separate table that states the updates to all figures, for the reviewers and editor only.

2. The extreme phenotype of tissue specific A1/G1 deletion has been used extensively yet, the role each plays is still debated. before mechanistic work can move forward, it seems like a good idea to tease the importance of these two apart since this would be valuable information in the field of cholesterol efflux and intra-cellular flux.

We agree with the reviewer that this is an interesting point. However, we wish to point out that the role of single Abcg1 deficiency in T-cells has already been studied extensively by the labs of Dr. Tontonoz and Dr. Hedrick¹⁻³. The Tontonoz lab has identified Abcg1 as the main cholesterol transporter in T-cells, and has revealed its crucial role in T-cell proliferation¹. These findings were replicated by the Hedrick lab². The Tontonoz and Hedrick labs have shown that Abca1 expression in T-cells is low^{1,2}, except for the setting of Abcg1 deficiency, which leads to a 6-fold upregulation of Abca1².

We have described the main findings of the Hedrick and Tontonoz papers in our introduction on page 3, as a critical part of the premise of our experiments (mostly part of the original submission): 'It has been suggested that *Abcg1* is the most highly expressed cholesterol transporter in T-cells¹. T-cell *Abcg1* deficiency increases differentiation of naïve T-cells into *T_{regulatory}* cells (*T_{regs}*), which suppresses atherosclerosis³. Even though it has been reported that *Abca1* expression in T-cells is low¹, *Abcg1*^{-/-} T-cells show a 6-fold increase in *Abca1* expression², suggesting that similar to macrophages¹⁶, the cholesterol transporters *ABCA1* and *ABCG1* have overlapping roles and show mutual compensation in T-cells.'

Following the suggestion of the reviewer, we have performed additional experiments in T-cell *Abca1* and T-cell *Abcg1* deficient mice assessing intracellular lipid droplets. We describe these on page 4 of the results section: 'We also observed BODIPY 493/503 staining reflecting lipid droplets, in *Abcg1*-deficient T-cells, in line with CE accumulation data as shown in², but not in *Abca1*-deficient T-cells or controls (Supplementary Fig. 1q, r).' Further, we highlight the previous studies on single *Abcg1* deficiency and our newly generated data on page 5: 'Collectively, previous studies have shown that *Abcg1* deficiency increases membrane cholesterol³, and cholera toxin B staining reflecting increased lipid rafts as well as CE in T-cells². We here show that T-cell *Abcg1* deficiency increases intracellular lipid droplets, consistent with CE accumulation, while *Abca1* deficiency shows no effect, in line with *Abcg1* being the predominant T-cell cholesterol transporter¹. However, *Abcg1* deficiency in T-cells increases *Abca1* expression by 6-fold², suggesting that *Abca1* may contribute to cholesterol accumulation in the setting of *Abcg1* deficiency. Consistently, we found that combined deficiency of *Abca1* and *Abcg1* in T-cells increased both free and esterified cholesterol, reflected by plasma membrane cholesterol accumulation and presence of lipid droplets, which was independent of *Ldlr* expression.'

In the previous version of the manuscript we had shown that T-cell single *Abca1* or *Abcg1* deficiency did not affect peripheral T-cell percentages/counts in blood (Supplementary Fig. 3a, b), while combined T-cell *Abca1/Abcg1* deficiency decreased blood CD4⁺ and CD8⁺ T-cells by ~50% in *Ldlr*^{-/-} (Fig. 2a-c) and *Ldlr*^{+/+} mice (Supplementary Fig. 3c). In addition, single T-cell *Abca1* or *Abcg1* deficiency did not affect T-cell activation in blood (Supplementary Fig. 4a, b), while combined T-cell *Abca1/Abcg1* deficiency increased *T_{memory/effector}* (*T_{mem/eff}*; CD4⁺CD44⁺CD62L⁻ and CD8⁺CD44⁺CD62L⁻) and *T_{central memory}* (*T_{central mem}*; CD8⁺CD44⁺CD62L⁺) cells as a percentage of CD4⁺ or CD8⁺ T-cells by >3-fold (Fig. 2h-j).

We found that T-cell *Abca1/Abcg1* deficiency showed a pronounced effect on T-cell apoptosis and we thus investigated whether single T-cell *Abca1* or T-cell *Abcg1* deficiency contributed to this phenotype. We describe these data on page 7: 'We found that T-cell *Abca1/Abcg1* deficiency also increased cleaved caspase 3/7 in *Ldlr*^{+/+} T-cells, as assessed by flow cytometry (Fig. 3c, d), while single *Abca1* or *Abcg1* deficiency showed no effect (Supplementary Fig. 6n, o).' We discuss these data on page 15: 'Single deficiency of *Abca1* or *Abcg1* did not affect peripheral T-cells in blood or T-cell apoptosis, likely because deficiency of one transporter results in upregulation of the other. Indeed, it has been shown previously that *Abcg1* deficiency upregulates *Abca1* expression in T-cells².'

While previous reports have been essential for dissecting key transporters in T-cells and their effect on T-cell behavior, they left a gap in our understanding of the physiological situation, where both *Abca1* and *Abcg1* are present and functional. Due to our unique colony of mice with combined *Abca1/Abcg1* deficiency in T-cells, we are adding significantly to this field. The synergy between cholesterol transporters in T-cells may be a critical physiological mechanism for regulating T-cells. This is further supported by a recent study⁸ that we discuss on page 16 in view of already published data and our own findings: 'Upon TCR signaling, the plasma membrane cholesterol content increases^{1,10}, presumably due to suppression of *Abca1* and *Abcg1* expression and increased cholesterol synthesis¹. Indeed, our studies have shown that T-cell *Abca1/Abcg1* deficiency increased plasma membrane cholesterol accumulation leading to plasma membrane stiffening and enhanced TCR signaling. Moreover, a recent study has shown that T-cell histone deacetylase 3 (*Hdac3*) deficiency decreases TCR signaling and CD4⁺ T-cell

proliferation, which was attributed to decreased membrane cholesterol content and increased *Abca1* and *Abcg1* mRNA expression⁸. This substantiates our data on T-cell *Abca1/Abcg1* deficiency increasing TCR signaling.'

3. Dysfunctional T cells in Fig 4 and 5 are characterized by surface stains but functional assays are essential.

We thank the reviewer for this suggestion. Additional experiments that we performed to address this offered new insights and helped us clarify downstream effects on atherogenesis (in response to point 3 of Reviewer 2 and point 6 of Reviewer 3). We describe the outcome of these experiments on page 8: 'Upon stimulation by α CD3/interleukin 2 (IL-2)), T-cell *Abca1/Abcg1* deficiency increased CD8⁺granzyme B⁺ T-cells as a percentage of total CD8⁺ T-cells (Fig. 4a, b), and CD4⁺ interferon gamma (IFN γ)⁺ and CD8⁺IFN γ ⁺ T-cells as percentages of CD4⁺ and CD8⁺ T-cells, respectively (Fig. 4c, d). T-cell *Abca1/Abcg1* deficiency also increased lysosomal-associated membrane protein 1 (LAMP-1) surface expression on CD8⁺ T-cells (Fig. 4e, f), indicating T-cell degranulation. These findings are suggestive of T-cell *Abca1/Abcg1* deficiency increasing the TCR response, and increasing T-cell functionality. However, in the same assay, T-cell *Abca1/Abcg1* deficiency decreased IFN γ secretion into the media (Fig. 4g), presumably due to increased T-cell apoptosis. Indeed, *Abca1/Abcg1* deficiency decreased T-cell counts by ~80% after stimulation by α CD3/IL-2 (Supplementary Fig. 6z), while the number of T-cells at the start of the assay was similar between genotypes. Therefore, even though T-cell *Abca1/Abcg1* deficiency increased the percentage of IFN γ ⁺ T-cells, it decreased IFN γ secretion presumably due to increased T-cell apoptosis. We subsequently investigated T-cell mediated effects on macrophage function.

T-cell IFN γ secretion enhances the capacity of macrophages to kill bacteria¹⁷. To examine whether this was affected by T-cell *Abca1/Abcg1* deficiency, we co-incubated wild-type bone marrow derived macrophages (BMDMs) with conditioned medium from α CD3/IL-2 stimulated *Ldlr*^{-/-} or T-*Abc*^{dko}*Ldlr*^{-/-} CD8⁺ T-cells prior to infection with *E.coli* bacteria. Conditioned medium from T-*Abc*^{dko}*Ldlr*^{-/-} CD8⁺ T-cells increased the number of *E.coli* bacteria colony forming units compared to medium from *Ldlr*^{-/-} CD8⁺ T-cells (Fig. 4h), reflecting reduced bacterial killing capacity by macrophages.

We then examined effects of *Abca1/Abcg1* deficiency on T-cell mediated cytotoxicity by co-incubating α CD3/IL-2 stimulated *Ldlr*^{-/-} or T-*Abc*^{dko}*Ldlr*^{-/-} T-cells with wild-type BMDMs. Lactate dehydrogenase (LDH), a cytosolic enzyme released into the medium upon damage of the plasma membrane, reflects cell death⁶. T-*Abc*^{dko}*Ldlr*^{-/-} T-cells induced less BMDM LDH release compared to *Ldlr*^{-/-} T-cells (Fig. 4i), indicating less macrophage killing and decreased T-cell mediated cytotoxicity.

Therefore, despite increasing IFN γ ⁺ T-cells and CD8⁺granzyme B⁺ T-cells, T-cell *Abca1/Abcg1* deficiency decreased the capacity of macrophages to kill bacteria as well as T-cell mediated macrophage killing, presumably due to increased T-cell apoptosis. These findings indicate that T-cell *Abca1/Abcg1* deficiency decreases T-cell functionality.'

In addition, as per the request of the reviewer, we performed additional experiments in T-cells from young and aged wild-type mice, as described on page 9-10: 'Further, upon stimulation by α CD3/IL-2, aging increased CD8⁺granzyme B⁺ cells as percentage of CD8⁺ T-cells (Supplementary Fig. 7f, g) and CD4⁺IFN γ ⁺ and CD8⁺IFN γ ⁺ cells as percentages of CD4⁺ and CD8⁺ T-cells, respectively (Supplementary Fig. 7h, i). Aging also increased LAMP-1 surface expression on CD4⁺ and CD8⁺ cells (Supplementary Fig. 7j, k), indicating increased T-cell degranulation. These findings suggest that aging increased the TCR response and enhanced T-cell functionality. Consistently, aging increased IFN γ secretion from T-cells (Supplementary Fig. 7l), in line with previous studies^{18,19}, but different from mice with T-cell *Abca1/Abcg1* deficiency. The latter was presumably due to the more pronounced increase of T-cell apoptosis upon T-cell *Abca1/Abcg1* deficiency than upon T-cell aging. Still, the immune system loses its

ability to mount an effective immune response to pathogens during aging²⁰, and our findings suggest that increased T-cell apoptosis may contribute.'

- Fig 5 showing young and aged proliferation is subtle and should be followed up with a functional assay as well.

We were not meaning to show a difference between young and aged wild-type mice in terms of T-cell proliferation, but rather meant to show no difference, even though aging slightly decreased CD4⁺ T-cell proliferation. We describe in the text on page 9 (already in the previous version): 'CD4⁺ T-cells from aged mice showed a slight decrease in T-cell proliferation in response to TCR stimulation, while CD8⁺ T-cell proliferation was not affected by aging (Fig. 5i-k), in line with previous findings showing that T_{naive} cells from aged mice still proliferate efficiently²¹.' This figure is mainly illustrating that aged T-cells still do proliferate efficiently. The main purpose of our studies was to show that aging increased T-cell cholesterol accumulation and lipid rafts (Fig 5a-e) as well as T-cell apoptosis (Fig 5 l-m).

- Fig 6 showing premature T cell aging as measure by surface stain and proliferation (CFSE) is interesting but a loss and/or gain of function with loading cholesterol or cyclodextrin would be more convincing.

It has been reported that methyl- β -cyclodextrin (M β CD) decreases CD8⁺ T-cell proliferation (due to membrane cholesterol depletion) and methyl- β -cyclodextrin-cholesterol (M β CD-cholesterol) increases CD8⁺ T-cell proliferation (due to membrane cholesterol loading)²². In keeping the times of incubation and the doses similar to those reported in this previous publication (1 mM M β CD for 5 mins for cholesterol depletion and 40 μ g/ml cholesterol (from M β CD-cholesterol complex) for 15 mins for cholesterol loading; the latter being twice the concentration from the publication), we observed that M β CD decreased CD8⁺ T-cell proliferation (~50% increase in T-cells that were not proliferating (in phase '0') compared to controls), while M β CD-cholesterol increased it (~50% decrease in T-cells that were not proliferating (in phase '0') compared to controls) in T-cells from middle-aged (12-13 months) Ldlr^{-/-} mice. This is consistent with plasma membrane cholesterol regulating CD8⁺ T-cell proliferation. We are showing the figure below for the reviewer. Data on CD8⁺ Ldlr^{-/-} T-cells are shown in panel b.

Effects of methyl- β -cyclodextrin or methyl- β -cyclodextrin-cholesterol on proliferation of T-cells from middle-aged *Ldlr*^{-/-} mice with T-cell *Abca1/Abcg1* deficiency and controls. Splenic T-cells isolated from 12-13 months old *Ldlr*^{-/-} and *T-Abc*^{dko}*Ldlr*^{-/-} mice were labeled with CFSE, incubated with 1mM methyl- β -cyclodextrin (M β CD) for 5 minutes or 40 μ g/mL cholesterol (from M β CD-cholesterol complex) for 15 minutes, and stimulated with α CD3/ α CD28 beads. CFSE dilution was measured at 72 h after stimulation by flow cytometry. The number of divisions were quantified for CD4⁺ (a, c) and CD8⁺ T-cells (b, d). n=3-6. For all panels, error bars represent SEM. n indicates biological replicates. p value was determined by unpaired two-tailed t-test. *p<0.05, **p<0.01.

However, as shown in panel a of this figure, in CD4⁺ T-cells, M β CD induced only a minor decrease in T-cell proliferation, and cholesterol-loading showed no effect, presumably due to the short time of the incubation of T-cells with M β CD or M β CD-cholesterol. In a previous publication employing a T-cell proliferation assay using thymidine, T-cell cholesterol loading using 20 μ g/ml cholesterol (from M β CD-cholesterol complex) for 2 hrs (half the dose but 8x the time of our experiment) did increase CD4⁺ T-cell proliferation in wild-type T-cells². However, cholesterol-loading only affected proliferation of wild-type T-cells, but not T-cells deficient in *Abcg1*, which was attributed to the latter already having accumulated high levels of cholesterol in their plasma membranes. Consistent with the observations in *Abcg1*^{-/-} T-cells, we observed no effects of M β CD-cholesterol on T-cell proliferation in CD4⁺ or CD8⁺ T-cells from mice with T-cell *Abca1/Abcg1* deficiency, as shown in panel c and d of the figure above.

Since treatment with M β CD was rather short, and since long treatment of M β CD leads to cytotoxicity, we used reconstituted high-density lipoprotein (rHDL) to induce T-cell cholesterol depletion. These studies are described on page 10: 'Moreover, treatment with reconstituted HDL (rHDL), which promotes cholesterol efflux in the absence of *Abca1/Abcg1* via passive efflux mechanisms, at least in macrophages²³, for the whole duration of the proliferation assay (72 hours), still further decreased T-cell proliferation in control and *Abca1/Abcg1* deficient T-cells (Supplementary Fig. 8d-h). These data indicate that even when proliferation is almost impaired in middle-aged T-cells with *Abca1/Abcg1* deficiency, cholesterol depletion still suppresses T-cell proliferation. The remaining T-cell proliferation is thus cholesterol-dependent.' We discuss these data on page 16: 'We found that even when TCR-induced T-cell proliferation was almost completely abolished in T-cells with *Abca1/Abcg1* deficiency, presumably due to T-cell senescence, cholesterol depletion by rHDL still further decreased it, substantiating the importance of T-cell membrane cholesterol for TCR signaling'.

These results thus clearly show that effects of T-cell *Abca1/Abcg1* deficiency on T-cell proliferation are cholesterol-dependent.

6. Figs 7 and 8 are interesting since it appears that it goes against what one might expect (greater athero) however, the big question of how dysfunctional the T cells are remain and the question of auto-immunity or B cell alternation is still open.

We have examined the functionality of T-cells with *Abca1/Abcg1* deficiency in response to point 3. We investigated the effects of T-cell *Abca1/Abcg1* deficiency on auto-immunity in previous studies, and found that at 40 weeks of age, *LckCreAbca1*^{fl/fl}*Abcg1*^{fl/fl} mice did not show auto-immune glomerulonephritis, while mice with dendritic cell *Abca1/Abcg1* deficiency showed an auto-immune glomerulonephritis phenotype, accompanied by increased germinal center (GC) B-cells and plasma cells¹¹. Since *LckCreAbca1*^{fl/fl}*Abcg1*^{fl/fl} mice did not show glomerulonephritis in the previous study, we did not further investigate this phenotype in *CD4CreAbca1*^{fl/fl}*Abcg1*^{fl/fl} mice.

In response to the reviewer's request, we measured GC B-cells and plasma cells in para-aortic LNs from *Ldlr*^{-/-} and *T-Abc*^{dko}*Ldlr*^{-/-} mice and found that T-cell *Abca1/Abcg1* deficiency did not affect GC B-cells, while decreasing plasma cells. Since this is not the major topic of our study, we have not added these data to the manuscript, but are showing them below for the reviewer only.

Effects of T-cell *Abca1/Abcg1* deficiency on germinal center B-cells and plasma cells in middle-aged *Ldlr*^{-/-} mice. *Ldlr*^{-/-} and *T-Abc*^{dko}*Ldlr*^{-/-} mice were fed a chow diet for 12-13 months. Para-aortic lymph nodes (LN) were collected, cells were stained with the indicated antibodies, and analyzed by flow cytometry. **(a-c)** Representative flow cytometry plots of B220⁺GL7⁺ germinal center (GC) B-cells **(a)**, GC B-cells as percentage of total para-aortic LN cells **(b)** and expressed as cells/LN after correction for total para-aortic LN cell number **(c)**. *n*=6. **(d-f)** Representative flow cytometry plots of CD19⁺CD138⁺ plasma cells **(d)**, plasma cells as percentage of total para-aortic LN cells **(e)** and expressed as cells/LN after correction for total para-aortic LN cell number **(f)**. *n*=6. For all panels, error bars represent SEM. *p* value was determined by unpaired two-tailed *t*-test. **p*<0.05.

Reviewer #2: The report by Bazioti et al. entitled “T-cell Abca1 and Abcg1 cholesterol efflux pathways suppress T-cell apoptosis and senescence and increase atherosclerosis in middle-aged Ldlr^{-/-} mice” is interesting. The authors show that the combined absence of cholesterol transporters Abca1 and Abcg1 in T cells alters plasma cholesterol accumulation, T cell activation and apoptosis promoting an aged T cell phenotype affecting atherosclerosis progression in middle aged mice. Overall, the experiment performed are convincing. I have some specific questions suggestions to strengthen the manuscript. Specific instances are outlined in the response to the authors.

We thank the reviewer for the positive comments and the suggestions to strengthen the manuscript.

1. Most of the experiments are performed in whole body Ldlr^{-/-} background (to induce atherosclerosis). It is unclear the influence of lack of Ldlr expression in T cells in the phenotype observed both in vitro and in vivo. Is the same phenotype observed in T cells in vivo, in a model of induction of atherosclerosis using Adenoviral-hepatic mediated expression of Pcsk9 in mice which results in a low-density lipoprotein receptor knockout phenotype by only affecting hepatic expression and thus the T cells express normal levels of LDRr?

The reviewer brings up an interesting point that we had also considered. In designing these studies, we had performed several experiments in mice that did not lack Ldlr. Mice with T-cell Abca1/Abcg1 deficiency expressing the Ldlr show exactly the same phenotype as mice deficient in Ldlr, in terms of membrane cholesterol and lipid droplet accumulation, T_{memory/effector} (T_{mem/eff}), T_{central memory} (T_{central mem}), and total T-cells in blood, and T-cell proliferation and T-cell apoptosis in vitro.

We describe these data on page 4 (new data): ‘T-cell Abca1/Abcg1 deficiency increased filipin staining at the plasma membrane, reflecting cholesterol accumulation, in Ldlr^{-/-} and Ldlr^{+/+} T-cells (Supplementary Fig. 1f-h).’ And also on page 4: ‘In line with cholesteryl ester (CE) accumulation, T-cell Abca1/Abcg1 deficiency increased Oil Red O staining, reflecting lipid droplets in para-aortic lymph nodes (LNs) (Supplementary Fig. 1l, m), and BODIPY 493/503 staining in Ldlr^{-/-} and Ldlr^{+/+} T-cells (Supplementary Fig. 1n-p).’ We subsequently conclude on page 5: ‘Consistently, we found that combined deficiency of Abca1 and Abcg1 in T-cells increased both free and esterified cholesterol, reflected by plasma membrane cholesterol accumulation and presence of lipid droplets, which was independent of Ldlr expression.’

Also in terms of T_{mem/eff}, T_{central mem}, and total T-cells, phenotypes in mice expressing the Ldlr and deficient in Ldlr were similar, as stated on page 6: ‘We thus assessed the effect of single and combined T-cell Abca1/Abcg1 deficiency on T-cell activation. Single T-cell Abca1 or Abcg1 deficiency did not affect T-cell activation in blood (Supplementary Fig. 4a, b). Combined T-cell Abca1/Abcg1 deficiency increased T_{memory/effector} (T_{mem/eff}; CD4⁺CD44⁺CD62L⁻ and CD8⁺CD44⁺CD62L⁻) and T_{central memory} (T_{central mem}; CD8⁺CD44⁺CD62L⁺) cells as a percentage of CD4⁺ or CD8⁺ T-cells by >3-fold in Ldlr^{-/-} (Fig. 2h-j) and Ldlr^{+/+} mice (Supplementary Fig. 4c, d). Further, T-cell Abca1/Abcg1 deficiency decreased CD4⁺ and CD8⁺ T_{naive} cells (CD44⁻CD62L⁺) cells by ~50% as a percentage of CD4⁺ or CD8⁺ T-cells in Ldlr^{-/-} (Fig. 2h-j) and Ldlr^{+/+} mice (Supplementary Fig. 4c, d).’

In terms of T-cell receptor (TCR) induced T-cell proliferation, the outcome in T-cells isolated from mice expressing the Ldlr and deficient in Ldlr were similar, as described on page 7: ‘We thus assessed whether T-cell Abca1/Abcg1 deficiency affected T-cell proliferation downstream of TCR signaling. Upon TCR stimulation, Abca1/Abcg1 deficiency increased T-cell proliferation in CD4⁺ and CD8⁺ Ldlr^{-/-} and Ldlr^{+/+} T-cells (Supplementary Fig. 6c-h).’

Also in terms of TCR induced T-cell apoptosis, the outcome in T-cells isolated from mice expressing the Ldlr and deficient in Ldlr were similar, as described on page 7: ‘Using αCD3 and interleukin 2 (IL-2) as stimuli, we examined whether T-cell Abca1/Abcg1 deficiency enhanced apoptosis downstream of TCR stimulation, by monitoring expression of cleaved caspase 3/7 in T-cells over time using the Incucyte

system. T-cell Abca1/Abcg1 deficiency increased cleaved caspase 3/7 by 3.7-fold in CD4⁺ T-cells and 1.8-fold in CD8⁺ T-cells, reflecting increased T-cell apoptosis (Fig. 3a, b and Supplementary Fig. 6m). These experiments were carried out in Ldlr^{-/-} T-cells. We found that T-cell Abca1/Abcg1 deficiency also increased cleaved caspase 3/7 in Ldlr^{+/+} T-cells, as assessed by flow cytometry (Fig. 3c, d), while single Abca1 or Abcg1 deficiency showed no effect (Supplementary Fig. 6n, o).'

In addition, effects of T-cell Abca1/Abcg1 deficiency on abolished T-cell proliferation in middle-aged mice were independent of Ldlr, as we describe on page 10: 'Strikingly, upon TCR stimulation, T-cell Abca1/Abcg1 deficiency almost completely abolished CD4⁺ and CD8⁺ T-cell proliferation, while T-cells from control mice still proliferated, in the absence (Fig. 6c-e) and presence of Ldlr expression (Fig. 6f-h).'

We also discuss these data on page 15: 'Although Ldlr mediated LDL uptake increases TCR stimulation in vitro⁴, illustrating the importance of membrane cholesterol content for TCR signaling, Ldlr deficiency has not been reported to decrease peripheral T-cell numbers in vivo. Consistently, we found that the increased T-cell apoptosis and decreased peripheral T-cells upon T-cell Abca1/Abcg1 deficiency were independent of Ldlr expression.'

The outcome of these experiments illustrates that the main outcomes of our study, i.e. that T-cell Abca1/Abcg1 deficiency increases T-cell cholesterol and CE accumulation, increases T-cell activation, decreases peripheral T-cells, increases TCR signaling and T-cell apoptosis, and abolishes T-cell proliferation in middle-aged mice were independent of Ldlr expression in T-cells. Since we obtained the same results in CD4CreAbca1^{fl/fl}Abcg1^{fl/fl} and CD4CreAbca1^{fl/fl}Abcg1^{fl/fl}Ldlr^{-/-} mice compared to their controls, we did not think that experiments using the PCSK9-AAV would yield additional insights as to whether effects of T-cell Abca1/Abcg1 deficiency on abovementioned parameters would be independent of the Ldlr, and thus did not study this further. We should also note that mice with whole-body Ldlr deficiency have been used by the Hedrick and Moore labs to study T cell dynamics^{3,5}, also for studies on T-cell Abcg1 deficiency and atherogenesis³.

We have enclosed a separate table that states the updates to all figures (above), for the reviewers and editor only.

2. It is unclear, why plasma cholesterol levels and distribution in different lipoproteins is altered in mice lacking ABCA1 and ABCG1 when fed a WD. This might explain why there are not differences in plaque size in WT and ABCA1/G1 deficient mice fed a Western diet. The authors should perform FPLC analysis and assess TC, TAG and HDL in aged Ldlr^{-/-} and Ldlr-Abca1/g1 fed a chow diet.

The reviewer is correct that the decrease in plasma very low-density lipoprotein/low-density lipoprotein (VLDL/LDL)-cholesterol may have contributed to atherosclerotic plaque size not being different between Ldlr^{-/-} mice with T-cell Abca1/Abcg1 deficiency and controls fed Western-type diet (WTD). We have performed fast performance liquid chromatography (FPLC), total cholesterol (TC), and triglyceride (TAG) analyses for T-Abc^{dko}Ldlr^{-/-} and Ldlr^{-/-} mice fed WTD or chow diet and describe the data on page 11-12, including the reason for different observation on WTD and chow diet: 'After 10 weeks of WTD, we assessed atherosclerotic lesion size at the level of the aortic root. T-cell Abca1/Abcg1 deficiency did not affect atherosclerotic lesion size (Fig. 7j and Supplementary Fig. 10a). This was accompanied by a decrease in plasma total cholesterol levels of ~15%, which reflects decreased VLDL and LDL cholesterol, while plasma triglycerides (TG) or VLDL-TG was not affected (Supplementary Table 1 and Supplementary Fig. 10b, c). The decrease in plasma VLDL/LDL-cholesterol was likely a consequence of increased total cholesterol accumulation in T-cells with Abca1/Abcg1 deficiency. Similarly, previous studies have shown that hematopoietic Abca1/Abcg1 deficiency decreased plasma VLDL/LDL-cholesterol levels in Ldlr^{-/-} mice fed WTD²⁴. However, on a chow diet, hematopoietic Abca1/Abcg1 deficiency, while still inducing cholesterol accumulation in hematopoietic cells, did not affect plasma VLDL/LDL-cholesterol levels^{23,25}, presumably because the clearance of VLDL/LDL particles by hepatic

Ldlr is not as much of a limiting factor as in *Ldlr*^{-/-} mice fed WTD. In an attempt to exclude the confounding factor of decreased VLDL/LDL cholesterol levels to atherogenesis, we fed mice a chow diet for 28 weeks, similar to a study we carried out previously²³. Chow diet-fed T-Abc^{dko}*Ldlr*^{-/-} and *Ldlr*^{-/-} mice had similar plasma total cholesterol and TG levels, as well as distribution of these lipids over VLDL, LDL, or high-density lipoprotein (HDL) (Supplementary Table 1 and Supplementary Fig. 10d, e) and developed atherosclerotic lesions similar in size compared to WTD-fed *Ldlr*^{-/-} mice (Supplementary Fig. 10f), but there was no difference in atherosclerotic lesion size between the genotypes (Fig. 7k).’

We also carried out plasma lipid measurements in the middle-aged T-Abc^{dko}*Ldlr*^{-/-} and *Ldlr*^{-/-} mice, as described on page 12 (new data): ‘Under conditions of similar plasma total cholesterol and TG levels, and similar distribution of these lipids on VLDL, LDL, and HDL (Supplementary Table 1 and Supplementary Fig. 12n, o), T-cell *Abca1/Abcg1* deficiency decreased atherosclerotic lesion size by ~35% (Fig. 8a, b),’

3. The manuscript will benefit from a more detailed characterization of atherosclerotic lesions. For instance, since T-cell activation influences macrophage apoptosis. The authors should analyze number of apoptotic macrophages and in addition of plaque necrotic area quantification of fibrous cap thickness. This analysis is important for mice fed a CD for 12 months.

We thank the reviewer for the insightful suggestion. We have performed these measurements and describe these on page 13 of the manuscript: ‘To obtain more insights into the consequences of these processes for atherosclerotic plaques of middle-aged T-Abc^{dko}*Ldlr*^{-/-} mice, we characterized their atherosclerotic lesions further. Similar to data in young *Ldlr*^{-/-} mice (Supplementary Fig. 11n, q, r), T-cell *Abca1/Abcg1* deficiency did not affect smooth muscle actin (α -SMA) staining or fibrous cap thickness in middle-aged *Ldlr*^{-/-} mice (Supplementary Fig. 12p-r), but decreased total collagen area in middle-aged *Ldlr*^{-/-} mice (Supplementary Fig. 12s, t). However, after correction for total lesion area, collagen content was not different between the genotypes (Supplementary Fig. 12u). Only few necrotic cores were present in atherosclerotic plaques and the necrotic core area was not different between the genotypes (Supplementary Fig. 12v-w). Although the density of macrophages in atherosclerotic lesions was relatively low (~6-10% of the atherosclerotic lesions), T-cell *Abca1/Abcg1* deficiency did increase macrophage content by ~60% (Fig. 9a-c). Previous studies have shown that in advanced atherosclerotic plaques, T-cells induce macrophage apoptosis, primarily mediated by granzyme B or perforin²⁶. While T-cell *Abca1/Abcg1* deficiency showed a tendency to decrease the total Terminal deoxynucleotidyl transferase dUTP nick end labeling (TUNEL)⁺Mac-2⁺ area in middle-aged mice, this decrease became statistically significant when corrected for total Mac-2⁺ area (Fig. 9d-f), reflecting a decrease in macrophage apoptosis. These data are consistent with T-cell *Abca1/Abcg1* deficiency inducing less macrophage killing upon TCR stimulation in the T-cell macrophage co-incubation experiment (Fig. 4i).’

At the end of page 13-14, we summarize these observations: ‘Collectively, T-cell *Abca1/Abcg1* deficiency decreased atherosclerotic lesion size in middle-aged *Ldlr*^{-/-} mice, presumably due to increased T-cell apoptosis and decreased inflammatory gene expression in the aorta. Moreover, T-cell *Abca1/Abcg1* deficiency increased macrophage lesion content in middle-aged *Ldlr*^{-/-} mice, likely due to decreased macrophage apoptosis.’

We discuss these findings on page 14: ‘In advanced lesions, CD8⁺ T-cells induce macrophage apoptosis and necrotic core formation mediated by granzyme B and perforin²⁶. T-cell *Abca1/Abcg1* deficiency did not affect blood myeloid cells or necrotic core formation, but increased macrophage content in middle-aged *Ldlr*^{-/-} mice, accompanied by a decrease in macrophage apoptosis. The latter would perhaps not have been expected since T-cell *Abca1/Abcg1* deficiency increased CD8⁺granzyme B⁺ T-cells upon α CD3/IL-2 stimulation. However, upon the same stimulus, *Abca1/Abcg1* deficiency increased T-cell apoptosis, and decreased macrophage death in a T-cell mediated cytotoxicity assay. Presumably, the effect of *Abca1/Abcg1* deficiency on T-cell apoptosis is predominant, and consequently, less granzyme B may have been secreted to mediate macrophage death. The decreased macrophage death in the T-

cell mediated cytotoxicity assay likely explains the decreased macrophage apoptosis and increased macrophage lesion content in plaques.'

In summary, even though the reviewer is correct that T-cell activation induces macrophage apoptosis, we think that the effect on T-cell apoptosis is predominant and explains why we observe less macrophage apoptosis and an increase in macrophage content in atherosclerotic lesions upon T-cell Abca1/Abcg1 deficiency in middle-aged Ldlr^{-/-} mice.

4. Is the efferocytotic capacity of macrophages, to remove apoptotic T-cells, influenced by different T cell membrane cholesterol content?

We thank the reviewer for this comment. Indeed, in the previous version of our manuscript, we suggested that the decrease in inflammation in the aorta was the consequence of an increase in efferocytosis, i.e. phagocytosis of apoptotic T-cells, which is an anti-inflammatory mechanism. We have performed the experiment the reviewer requested, and based our method on previous studies²⁷⁻²⁹. The data are shown below.

Although the data do show an increase in efferocytosis when macrophages were co-incubated with Abca1/Abcg1 deficient T-cells (panels a-b), and a decrease in inflammation (panel c), as we had hypothesized, we did not add these data to the manuscript, and omitted any statements suggesting that T-cell Abca1/Abcg1 deficiency increases efferocytosis. We had generated more data in response to other questions from the reviewers, suggesting that a different mechanism may account for the decrease in aortic inflammation in Ldlr^{-/-} mice with T-cell Abca1/Abcg1 deficiency. These data indicate that: 1) Abca1/Abcg1 deficient T-cells showed decreased secretion of highly pro-inflammatory interferon gamma (IFNγ) compared to their controls (Figure 4g); 2) T-cell Abca1/Abcg1 deficiency decreases mRNA expression of pro-inflammatory (M1) cytokines monocyte chemoattractant protein 1 (Mcp-1), Il-6, and Il-1β in the aorta and increases mRNA expression of the anti-inflammatory (M2) cytokine Ym-1 (Figure 9g); since IFNγ promotes M1 macrophage differentiation, data on decreased inflammation in the aorta are consistent with a decrease in IFNγ secretion by Abca1/Abcg1 deficient T-cells; and 3) Even though T-cell Abca1/Abcg1 deficiency increased T-cell apoptosis in the aorta (Figure 8i, 8k), T-cell Abca1/Abcg1

deficiency also decreased T-cell content of atherosclerotic plaques by ~50% indicating that a large majority of T-cells have already undergone apoptosis before entering the atherosclerotic plaque. We therefore think that the more plausible interpretation of our data is that T-cell *Abca1/Abcg1* deficiency decreases pro-inflammatory cytokine expression in the aorta by decreasing IFN γ secretion, ultimately resulting in decreased atherosclerotic lesion area in middle-aged *Ldlr*^{-/-} mice. Therefore, we omitted any suggestions towards an increase in efferocytosis.

5. The authors have performed a very comprehensive analysis of T-cell in circulation, spleen, etc... However, the impact of ABCA1/G1 deficiency in T-cells within the plaque which is the more relevant aspect of the manuscript is somehow poor. The authors rely on para-aortic LN analysis to evaluate T-cells in the aorta, which could be a confounding measurement. This could be improved using flow cytometry analysis in digested aortas and/or sc-RNA Seq. This analysis will provide a better understanding of how both transporters influence T-cell activation, proliferation and apoptosis in T-cells accumulated in atherosclerotic plaques.

*We agree with the reviewer and have performed the requested flow cytometry analyses on atherosclerotic plaques. We describe the outcome on page 13: 'However, T-cell Abca1/Abcg1 deficiency did not affect α CD3/ α CD28 induced proliferation of CD4⁺ T-cells isolated from aortas (Fig. 8g), perhaps due to these cells not showing a high level of proliferation compared to our findings in splenic T-cells (Fig. 6c-e). Nevertheless, similar to findings on splenic T-cells (Fig. 6a, b), T-cell Abca1/Abcg1 deficiency also increased cleaved caspase 3/7 in T-cells isolated from the aorta of middle-aged *Ldlr*^{-/-} mice upon stimulation with α CD3/ α CD28, which we only assessed in the CD4⁺ T-cell population (Fig. 8h, i), as we were facing technical challenges to isolate CD8⁺ T-cells from aortas. The increase in aortic CD4⁺ T-cell apoptosis was accompanied by a decrease in the percentage of aortic CD4⁺ T_{mem/eff} cells, (Fig. 8j) that also showed an increase in cleaved caspase 3/7 (Fig. 8k).'*

Reviewer #3: Bazioti and co-workers present a very interesting manuscript about the role of cholesterol efflux on T cell-mediated responses, and their consequences to the development of atherosclerosis. To this end, the investigators have produced a T-cell-specific *Abca1/Abcg1* knockout strain that was used for the characterization of T-cell responses; the same strain was backcrossed into *Ldlr*KO mice to evaluate the effects of *Abca1/Abcg1* deficiency on T cells in atherogenesis. Of note, the investigators also attempted to parallel the consequences of efflux deficiency with phenotypic changes that may occur during aging.

The study presented a clear hypothesis, and the manuscript is quite well written. The study design looked to a great extent appropriated, particularly regarding the characterization of the consequences of the combine deficiency of *Abca1/Abcg1* on intracellular and membrane cholesterol, as well as the repertoire of CD4⁺ and CD8⁺ T cells in the targeted mice. Nevertheless, although interesting, the attempt to explore both aging and atherogenesis made the overall story sometimes a bit confusing and difficult to follow. Thus, several questions should be taken into consideration by the authors for revising the manuscript:

We thank the reviewer for the positive comments on our manuscript and the suggestions to improve it.

1. One of the initial findings shown was the fact that T-cell *Abca1/Abcg1* deficiency led to intracellular and plasma membrane cholesterol accumulation (Fig1). Although this could be expected, it raises the question on how cholesterol (lipoprotein) uptake is handled on these cells. Regulated by the accumulation of esterified cholesterol, *Srebp2* over-expression should downregulate LDL receptor, which could reduce or even up lipid accumulation. Considering cells originated from *Ldlr*KO, is that the main reason for the observed phenotype? How relevant is efflux on T cells that can upregulate LDLR?

In response to the question from the reviewer, we evaluated mRNA expression of several genes involved in the cholesterol synthesis pathway. We describe these on page 4-5: 'Previous studies have shown that Abcg1 deficiency increases sterol regulatory element binding protein 1c (Srebp1c) mRNA expression in CD4⁺ T-cells, while decreasing Ldlr, 3-hydroxy-3-methyl-glutaryl-coenzyme A reductase (Hmgcr), HMG-CoA synthase (Hmgcs), and Srebp2 mRNA expression, reflecting a decrease in cholesterol synthesis, consistent with increased cholesterol accumulation in the ER^{2,30}. In line with these data, T-cell Abca1/Abcg1 deficiency moderately decreased Srebp2 mRNA expression in CD4⁺ T-cells and Ldlr and Hmgcr mRNA expression in CD8⁺ T-cells (Supplementary Fig. 1u, v).'

Based on these data, we rather observed that T-cell Abca1/Abcg1 deficiency induced a modest decrease in mRNA expression of genes affecting cholesterol synthesis, and definitely not an increase, as the reviewer suggested. This is in line with previous reports on T-cell Abcg1 deficiency as cited above. Of note, T-cell Abca1/Abcg1 deficiency increased membrane cholesterol and lipid droplet accumulation to a similar extent in wild-type and Ldlr^{-/-} T-cells, indicating that this phenotype was independent of Ldlr deficiency. See also reviewer 2, point 1, for a more elaborate response on the effects of T-cell Abca1/Abcg1 deficiency on membrane cholesterol and lipid droplet accumulation.

Please also find enclosed a separate table that states the updates to all figures, for the reviewers and editor only.

- 2. The investigators performed extensive flow cytometry analysis of the repertoire of T cells in T-AbcdkoLdlr^{-/-} and Ldlr^{-/-} mice. However, the presentation of data is a bit intriguing by showing the “%” data as well as the “% corrected by total number of cells”. Why percentages need to be corrected? What is “total number of cells”? the cells isolated from the organs? Cells analyzed? Were whole organs analyzed? In addition to not been described in methods, this approach seems unnecessary and perhaps wrong, which could change slightly some conclusions. The same criticism applies to the subset analysis of T cells in the atherosclerosis experiment, which should be clarified.*

We thank the reviewer for this comment. We should have stated our approach more clearly in the previous version of the manuscript. For the flow cytometry analyses, we have expressed the cells as percentages in order to observe the specific effect of T-cell Abca1/Abcg1 deficiency per cell type. However, we feel that the absolute number of cells per tissue is highly informative, since we observed a ~50% decrease in peripheral T-cells and therefore a quantification based on percentage only may be biased. The absolute number of cells that we reported are the numbers of cells per tissue. We have now indicated this more clearly on all graphs and in the text, and express the outcome as cells/thymus, cells/lymph node (LN), or cells/spleen. In an attempt to clarify this, we state both measurements in the manuscript, starting on page 5: 'single T-cell Abca1 or Abcg1 deficiency did not affect T-cells as a percentage of total blood leukocytes or, after correction for total blood leukocyte counts, as measures for T-cell concentration in blood (Supplementary Fig. 3a, b),'

Also on page 5: 'T-Abc^{dko}Ldlr^{-/-} mice showed no changes in thymic CD4⁺CD8⁻ double negative (DN) cells, thymic CD4⁺CD8⁺ double positive (DP) cells, or thymic CD4⁺ or CD8⁺ single positive (SP) cell populations compared to littermate controls, when shown as percentage of thymocytes, or after correction for total thymic cell numbers (Supplementary Fig. 3e-g),'

On page 6: 'Collectively, as percentages of total CD4⁺ or CD8⁺ T-cells, T-cell Abca1/Abcg1 deficiency increased T_{memory/effector} (T_{mem/eff}) and T_{central memory} (T_{central mem}) cells in blood, spleen, and para-aortic LNs. However, when corrected for total number of blood leukocytes or total number of cells per LN or spleen, the increases in T_{mem/eff} and T_{central mem} cells were no longer significant between the genotypes (Fig. 2k, l and Supplementary Fig. 4i-k).'

On page 11: 'After correction for total para-aortic LN cell number, T-cell *Abca1/Abcg1* deficiency decreased $CD25^+Foxp3^+$ $T_{regulatory}$ cells (T_{regs}) by ~75%, while not affecting $CD25^+Foxp3^+$ T_{regs} (Fig. 7c), consistent with the decrease in $CD25^+Foxp3^+$ T_{regs} that we observed in vitro (Fig. 3e, f), presumably because there were less total para-aortic LN T-cells (Fig. 2g) due to increased T-cell apoptosis. T-cell *Abca1/Abcg1* deficiency increased $T_{follicular\ helper}$ (T_{FH}) cells as a percentage of $CD4^+$ $T_{mem/eff}$ cells in para-aortic LNs (Fig. 7d, e), but did not affect T_{FH} cells after correction for total para-aortic LN cell number (Fig. 7f), presumably also due to the decrease in total para-aortic LN T-cells (Fig. 2g).'

On page 11: 'We also investigated whether these changes in T-cells affected myeloid cells in blood. When expressed as a percentage of total leukocytes, T-cell *Abca1/Abcg1* deficiency increased blood monocytes, $Ly6C^o$ and $Ly6C^{hi}$ monocyte subsets, as well as neutrophils by ~50% (Supplementary Fig. 9e). We explain these increases by T-cell *Abca1/Abcg1* deficiency decreasing T-cells by 50% (Fig. 2a, b) and therefore other leukocyte populations, including myeloid cells showing an increase as percentage of total leukocytes. Indeed, the absolute number of leukocytes was decreased by T-cell *Abca1/Abcg1* deficiency (Supplementary Fig. 9f), presumably due to the decrease in blood T-cells (Fig. 2a-c), and therefore, when corrected for total blood leukocyte number, T-cell *Abca1/Abcg1* deficiency no longer affected levels of myeloid cells in blood (Supplementary Fig. 9g).

We then investigated the effects of these changes on atherosclerosis. To induce atherogenesis, female T-*Abc^{dko}Ldlr^{-/-}* mice and *Ldlr^{-/-}* littermate controls were fed a Western-type diet (WTD). Findings on T-cell activation and T-cell subsets as well as on percentage of blood myeloid cells were similar to mice fed chow diet (Supplementary Fig. 9h-r). However, the increase in total monocytes, even though only ~15%, remained significant after correction for total blood leukocyte numbers in *Ldlr^{-/-}* mice with T-cell *Abca1/Abcg1* deficiency compared to controls (Supplementary Fig. 9s).'

On page 12: 'In line with observations in blood and secondary lymphoid organs, T-*Abc^{dko}Ldlr^{-/-}* mice showed a ~50% decrease in plaque $CD3^+$ T-cells on both diets compared to *Ldlr^{-/-}* controls, both when shown as total $CD3^+$ T-cells per section or as percentage $CD3^+$ T-cells of total cells (nuclei) in lesions (Fig. 7l-o and Supplementary Fig. 11a, b).'

On page 12: 'Similar to observations in young T-*Abc^{dko}Ldlr^{-/-}* mice (Fig. 7a-c), T-cell *Abca1/Abcg1* deficiency increased $CD25^+Foxp3^+$ T_{regs} but not $CD25^+Foxp3^+$ T_{regs} as a percentage of $CD4^+$ T-cells (Supplementary Fig. 12c, d). T-cell *Abca1/Abcg1* deficiency did not affect the number of $CD25^+Foxp3^+$ T_{regs} in whole para-aortic LNs, but decreased $CD25^+Foxp3^+$ T_{regs} by ~70% (Supplementary Fig. 12e), similar to young mice (Fig. 7c), and presumably due to an increase in T-cell apoptosis. Unlike in young mice (Supplementary Fig. 4g), T-cell *Abca1/Abcg1* deficiency did not affect $CD4^+$ $T_{mem/eff}$ cells as a percentage of $CD4^+$ T-cells (Supplementary Fig. 12f). Moreover, after correction for the total para-aortic LN cell number, T-cell *Abca1/Abcg1* deficiency decreased this population by ~50% (Supplementary Fig. 12g). $CD8^+T_{bet}^+$ cells were increased both as a percentage and after correction for total para-aortic LN cell number in para-aortic LNs from middle-aged T-*Abc^{dko}Ldlr^{-/-}* mice compared to controls, while $CD4^+T_{bet}^+$ cells were not affected (Supplementary Fig. 12h-j). Similar to $CD25^+Foxp3^+$ T_{regs} and the observations in young mice (Fig. 7d-f), T-cell *Abca1/Abcg1* deficiency increased T_{FH} cells as a percentage of LN $CD4^+$ $T_{mem/eff}$ cells, but not after correction for total para-aortic LN cell number (Supplementary Fig. 12k-m).'

Please also find enclosed a table (above) stating major changes to figures in the manuscript and also indicating which data we have now added as percentages.

3. In the end of the last paragraph of the section "T-cell *Abca1/Abcg1* Deficiency Increases T-cell Apoptosis Upon TCR Stimulation", the authors stated: "Collectively, these data indicate that T-cell *Abca1/Abcg1* deficiency enhances T-cell apoptosis downstream of TCR and CD25, mediated by Bcl2 in the intrinsic apoptosis pathway, especially in Tmemory/effector cells. This likely accounts for the ~50% decrease in peripheral T-cells in T-cell *Abca1/Abcg1* deficiency". Were

other modalities of cell death evaluated? How can the authors be so sure this is the main pathway?

The reviewer is correct that we cannot draw this conclusion that strongly based on the data we generated. In response to this point, we have performed a more in-depth analysis of T-cells in the thymus, and evaluated other modes of T-cell death such as pyroptosis. Data on the thymus are described on page 5 (new data): 'T-Abc^{dko}Ldlr^{-/-} mice showed no changes in thymic CD4⁺CD8⁻ double negative (DN) cells, thymic CD4⁺CD8⁺ DP cells, or thymic CD4⁺ or CD8⁺ SP cell populations compared to littermate controls, when shown as percentage of thymocytes, or after correction for total thymic cell numbers (Supplementary Fig. 3e-g), even though T-cell receptor β (TCR β)⁺CD24⁻ and TCR β ⁺CD69⁻ cells, indicative of excessive negative selection³¹, were decreased compared to Ldlr^{-/-} controls as a percentage of thymocytes (Supplementary Fig. 3h-m).' We discuss these findings on page 15: 'A previous study has attributed the decrease in peripheral T-cells in mice with T-cell Abca1/Abcg1 deficiency or T-cell deficiency of the transcription factor the liver X receptor (LXR) α and β that induces Abca1 and Abcg1 expression, to increased thymic CD4⁺ and CD8⁺ T-cell apoptosis, while not excluding that extrathymic effects on T-cells may also have contributed¹². However, in these studies, the Abca1 and Abcg1 floxed genes and also the Lxra/Lxr β floxed genes were expressed under control of the LckCre promoter¹².' Also on page 15-16: 'Perhaps the LckCre promoter resulted in a more complete deficiency of Abca1 and Abcg1 in single positive thymic CD4⁺ and CD8⁺ T-cells than the CD4Cre promoter that we and others used¹⁰, explaining the more pronounced effects on thymic T-cells.'

We thus do not think that the thymus is a main contributor to the decrease in peripheral T-cells, although we cannot entirely exclude this. We also evaluated pyroptosis as an alternative mode of cell death and describe the data on page 7: 'We then evaluated whether other modes of cell death may have contributed to the decrease in T-cells in mice with T-cell Abca1/Abcg1 deficiency. T-cell pyroptosis, a highly pro-inflammatory form of lytic programmed cell death is executed by gasdermin D cleavage⁶, and decreases T-cell numbers, at least in the setting of chronic HIV-1 infection³². Upon α CD3/IL-2 stimulation, we detected a very low level of cleaved gasdermin D p30, which was not different between the genotypes (Supplementary Fig. 6p, q), indicating that T-cell pyroptosis was not affected.'

We have attenuated our statement on page 8: 'Collectively, these data indicate that T-cell Abca1/Abcg1 deficiency enhances T-cell apoptosis, independent of Ldlr expression, and downstream of TCR and CD25, mediated by B-cell lymphoma 2 (Bcl2) in the intrinsic apoptosis pathway, especially in T_{mem/eff} cells. This likely contributed to the ~50% decrease in peripheral T-cells in T-cell Abca1/Abcg1 deficiency.'

4. The results under the section "Aging Increases Cholesterol Accumulation and Apoptosis in T-cells from Wild-Type Mice" are to some extent puzzling. Although the authors could build some nice parallels between the phenotype of aged C57BL6 mice and T-AbcdkoLdlr^{-/-}, it seems very speculative that only efflux defects could explain aging-driven phenotypic changes. How to reconcile that fact that in the current study, a comparison is made between a WT strain, which has HDL as the main circulating lipoprotein, against LdlrKO, which has VLDL and LDL? In this context, did the T cells from aged WT mice really had an accumulation of cholesterol intracellularly and in the plasma membrane? Fig.5a is not very convincing. How is the expression status of Abca1, Abcg1, Ldlr on T cells upon aging?

We wish to point out that as also examined in response to point 1 from this reviewer, and point 1 from reviewer 2, effects of T-cell Abca1/Abcg1 deficiency on peripheral T-cell numbers and apoptosis were independent of Ldlr expression. Therefore, we do not think that elevated very low-density lipoprotein (VLDL) and low-density lipoprotein (LDL) contributed to the T-cell phenotype in mice with T-cell Abca1/Abcg1 deficiency that we used in our studies. Although not directly asked by the reviewer, we did evaluate the lipoprotein profile of aged mice compared to young mice and describe the data on page 9: 'Aging did not affect plasma cholesterol, or its distribution over lipoproteins, but, in line with previous

observations³³, decreased plasma triglycerides (TG) by ~42% ($P < 0.001$), reflected by a decrease in the very low-density lipoprotein (VLDL)-TG fraction (Supplementary Table 1 and Supplementary Fig. 7d, e).'

We agree with the reviewer that it is likely not only efflux effects that contributes to the changes in T-cells during aging. We rather meant that cholesterol accumulation in the plasma membrane contributed to T-cell aging in wild-type mice. We performed additional experiments to address the points brought up by the reviewer and describe these on page 9: 'Similar to findings in aged humans^{34,35}, blood CD4⁺ and CD8⁺ T-cells from aged mice show an increase in filipin staining compared to young mice, reflecting an increase in plasma membrane cholesterol accumulation (Fig. 5a, b). Aging also increased cholera toxin B staining on CD8⁺ T-cells, suggestive of more lipid rafts (Fig. 5c, d). Further, aging increased total T-cell cholesterol content as assessed by Gas Chromatography – Mass Spectrometry (GC-MS) ~1.5-fold in T-cells from aged mice (Fig. 5e), while there were no signs of lipid droplets based on BODIPY 493/503 staining (Supplementary Fig. 7a), indicating that the increase in cellular cholesterol reflects an increase in free cholesterol. Aging decreases *Abca1* and *Abcg1* mRNA expression in mouse splenic and peritoneal macrophages³⁶; however, we found no effect of aging on mRNA expression of *Abca1* and *Abcg1* in T-cells, while mRNA expression of other genes affecting cholesterol metabolism such as 3-hydroxy-3-methyl-glutaryl-coenzyme A reductase (*Hmgcr*) and HMG-CoA synthase (*Hmgcs*) was minimally decreased and *Ldlr* mRNA expression showed a moderate decrease, especially in CD8⁺ T-cells (Supplementary Fig. 7b, c). *Ldlr* expression does not affect T-cell cholesterol accumulation (Supplementary Fig. 1f-h, n-p). Therefore, effects of aging on T-cell cholesterol accumulation cannot be explained by changes in expression of these genes. Perhaps repeated TCR stimulation increases membrane cholesterol accumulation during aging.'

5. A few times in the manuscript, the authors indicate "data not shown", which can generate some credibility doubts. Unless there would be editorial limits from the journal, and if relevant to the story, these data should be incorporated to the manuscript, for example on supplements.

We thank the reviewer for this comment. We have added all these data to the manuscript. In the table that summarizes the major changes to the manuscript (above), a column is shown that summarizes all data that were previously referred to as 'not shown' but now are shown in the manuscript.

6. Fig.8 showed that T-AbcdkoLdlr^{-/-} presented reduced atherosclerotic burden compared to Ldlr^{-/-}. Unexpectedly, an increased percentage of Mac2 positive cells was observed in T-AbcdkoLdlr^{-/-} mice. mRNA profiling showed reduced *Mcp1* expression in the same group, how to reconcile the 2 previous findings? Additionally, the increased percentage of macrophages in T-AbcdkoLdlr^{-/-} group seems counterintuitive considering the finding of smaller lesions; it would be interesting to evaluate M1 and M2 markers, e.g. on mRNA levels and check whether the latter polarization dominated the tissue.

We have clarified these observations and evaluated the mRNA expression of M1 and M2 markers in aortic CD3⁺ cells as described on page 13-14: 'Although the density of macrophages in atherosclerotic lesions was relatively low (~6-10% of the atherosclerotic lesions), T-cell *Abca1/Abcg1* deficiency did increase macrophage content by ~60% (Fig. 9a-c). Previous studies have shown that in advanced atherosclerotic plaques, T-cells induce macrophage apoptosis, primarily mediated by granzyme B or perforin²⁶. While T-cell *Abca1/Abcg1* deficiency showed a tendency to decrease the total Terminal deoxynucleotidyl transferase dUTP nick end labeling (TUNEL)⁺Mac-2⁺ area in middle-aged mice, this decrease became statistically significant when corrected for total Mac-2⁺ area (Fig. 9d-f), reflecting a decrease in macrophage apoptosis. These data are consistent with T-cell *Abca1/Abcg1* deficiency inducing less macrophage killing upon TCR stimulation in the T-cell macrophage co-incubation experiment (Fig. 4i). We then evaluated mRNA expression of inflammatory cytokines in the aortic T-cell negative fraction, consisting of endothelial cells, myeloid cells, and smooth muscle cells (SMCs), in middle-aged Ldlr^{-/-} or T-Abc^{dko}Ldlr^{-/-} mice. T-cell *Abca1/Abcg1* deficiency decreased mRNA expression of the M1 macrophage markers monocyte chemoattractant protein 1 (*Mcp1*), interleukin 1 beta (*Il-1β*), and

tumor necrosis factor alpha (*Tnfa*), and increased mRNA expression of the M2 macrophage marker *Ym1*, while not affecting mRNA expression of the myeloid cell marker *Cd11b*, or the cytokines *Il-6*, *Il-10*, or *Il-23a*, or intracellular adhesion molecule-1 (*Icam-1*), or the M2 macrophage marker *Fizz1* (Fig. 9g). These changes in expression are consistent with a decrease in aortic inflammation, most likely due to increased T-cell apoptosis and decreased interferon gamma (*IFN γ*) secretion. Collectively, T-cell *Abca1/Abcg1* deficiency decreased atherosclerotic lesion size in middle-aged *Ldlr^{-/-}* mice, presumably due to increased T-cell apoptosis and decreased inflammatory gene expression in the aorta. Moreover, T-cell *Abca1/Abcg1* deficiency increased macrophage lesion content in middle-aged *Ldlr^{-/-}* mice, likely due to decreased macrophage apoptosis.'

While *Mcp-1* indeed promotes monocyte infiltration, as suggested by the reviewer, we do not think that our findings on T-cell *Abca1/Abcg1* deficiency decreasing aortic inflammation and increasing macrophage content are in disagreement; rather, these findings are accounted for by two separate mechanisms involving 1) decreased macrophage killing by *Abca1/Abcg1* deficient T-cells and 2) T-cell *Abca1/Abcg1* deficiency decreasing inflammatory gene expression in the aorta. Moreover, an increase in lesion macrophage content indicates that these lesions are less advanced.

7. Fig.8 and Supp.Fig. 6 show quantification and representative pictures of lesions that were stained with H&E. Why was Oil Red O staining of lipids not used instead? It looks odd that a study investigating cholesterol-related mechanisms does not explore/show the lipid content on plaques. Moreover, representative H&E pictures suggest some differences on cellularity between genotypes, and perhaps also chow and diet protocols (Fig.8 and Supp.Fig.6) ; the study would benefit of further characterization of plaques, e.g. SMA staining.

As per the reviewer's request, we have performed a more thorough characterization of atherosclerotic lesions for *Ldlr^{-/-}* mice fed WTD or chow diet, including Oil Red O staining for mice fed chow diet, as described on page 12: 'Further, characterization of atherosclerotic lesions revealed no changes in fibrous cap thickness, collagen content, smooth muscle actin (α -SMA) or galectin-3 (*Lgals3* or *Mac-2*) staining, reflecting predominantly macrophages, after either WTD or chow diet (Supplementary Fig. 11c-u), suggesting no effects on plaque stability. In a separate cohort of mice, we evaluated lipid accumulation in atherosclerotic lesions in mice fed chow diet. After 38 weeks of chow diet (~9 months), T-cell *Abca1/Abcg1* deficiency did not affect atherosclerotic lesion area, similar to observations at 28 weeks of chow diet (Fig. 7k), but increased Oil Red O area as a percentage of lesion area, reflecting increased lipid accumulation (Supplementary Fig. 11v-y), presumably in both T-cells and macrophages.' See also our response to point 3 of reviewer 2 for effects of T-cell *Abca1/Abcg1* deficiency on lesion composition in middle aged *Ldlr^{-/-}* mice.

8. A minor point, in the discussion, some conclusions/suggestions from the current study are finalized with a citation, which looks confusing. For example, page 11 of the Word file, "We suggest that the decreased inflammation may be the consequence of uptake of apoptotic T-cells by myeloid cells in the aorta, which is an anti-inflammatory process 65". If this is a statement from the authors, it should not cite other work. This happens again in the subsequent paragraph, and such faults should be revised.

We thank the reviewer for this comment. We have re-written the corresponding sentences according to the reviewer's suggestion, for example on page 15: 'Our *in vitro* studies suggested that the decrease in *CD25⁺Foxp3⁺ T_{regs}* was entirely due to T-cell apoptosis. T-cell apoptosis is an anti-inflammatory process 37.' Also on page 15: 'Single deficiency of *Abca1* or *Abcg1* did not affect peripheral T-cells in blood or T-cell apoptosis, likely because deficiency of one transporter results in upregulation of the other. Indeed, it has been shown previously that *Abcg1* deficiency upregulates *Abca1* expression in T-cells 2.'

References

1. Bensinger, S. J., Bradley, M. N., Joseph, S. B., Zelcer, N., Janssen, E. M., Hausner, M. A., Shih, R., Parks, J. S., Edwards, P. A., Jamieson, B. D. & Tontonoz, P. LXR Signaling Couples Sterol Metabolism to Proliferation in the Acquired Immune Response. *Cell* **134**, 97–111 (2008).
2. Armstrong, A. J., Gebre, A. K., Parks, J. S. & Hedrick, C. C. ATP-Binding Cassette Transporter G1 Negatively Regulates Thymocyte and Peripheral Lymphocyte Proliferation. *The Journal of Immunology* **184**, 173–183 (2010).
3. Cheng, H. Y., Gaddis, D. E., Wu, R., McSkimming, C., Haynes, L. D., Taylor, A. M., McNamara, C. A., Sorci-Thomas, M. & Hedrick, C. C. Loss of ABCG1 influences regulatory T cell differentiation and atherosclerosis. *Journal of Clinical Investigation* **126**, 3236–3246 (2016).
4. Yuan, J., Cai, T., Zheng, X., Ren, Y., Qi, J., Lu, X., Chen, H., Lin, H., Chen, Z., Liu, M., He, S., Chen, Q., Feng, S., Wu, Y., Zhang, Z., Ding, Y. & Yang, W. Potentiating CD8+ T cell antitumor activity by inhibiting PCSK9 to promote LDLR-mediated TCR recycling and signaling. *Protein and Cell* **12**, (2021).
5. Sharma, M., Schlegel, M. P., Afonso, M. S., Brown, E. J., Rahman, K., Weinstock, A., Sansbury, B. E., Corr, E. M., van Solingen, C., Koelwyn, G. J., Shanley, L. C., Beckett, L., Peled, D., Lafaille, J. J., Spite, M., Loke, P., Fisher, E. A. & Moore, K. J. Regulatory T cells license macrophage pro-resolving functions during atherosclerosis regression. *Circulation Research* **127**, (2020).
6. Shi, J., Zhao, Y., Wang, K., Shi, X., Wang, Y., Huang, H., Zhuang, Y., Cai, T., Wang, F. & Shao, F. Cleavage of GSDMD by inflammatory caspases determines pyroptotic cell death. *Nature* **526**, (2015).
7. Gaddis, D. E., Padgett, L. E., Wu, R., Nguyen, A., McSkimming, C., Dinh, H. Q., Araujo, D. J., Taylor, A. M., McNamara, C. A. & Hedrick, C. C. Atherosclerosis Impairs Naive CD4 T-Cell Responses via Disruption of Glycolysis. *Arteriosclerosis, Thrombosis, and Vascular Biology* (2021) doi:10.1161/ATVBAHA.120.314189.
8. Wilfahrt, D., Philips, R. L., Lama, J., Kizerwetter, M., Shapiro, M. J., McCue, S. A., Kennedy, M. M., Rajcula, M. J., Zeng, H. & Shapiro, V. S. Histone deacetylase 3 represses cholesterol efflux during CD4+ T-cell activation. *eLife* **10**, (2021).
9. Gaddis, D. E., Padgett, L. E., Wu, R., McSkimming, C., Romines, V., Taylor, A. M., McNamara, C. A., Kronenberg, M., Crotty, S., Thomas, M. J., Sorci-Thomas, M. G. & Hedrick, C. C. Apolipoprotein AI prevents regulatory to follicular helper T cell switching during atherosclerosis. *Nature Communications* **9**, (2018).
10. Michaels, A. J., Campbell, C., Bou-Puerto, R. & Rudensky, A. Y. Nuclear receptor LXR β controls fitness and functionality of activated T cells. *The Journal of experimental medicine* **218**, (2021).
11. Westerterp, M., Gautier, E. L., Ganda, A., Molusky, M. M., Wang, W., Fotakis, P., Wang, N., Randolph, G. J., D'Agati, V. D., Yvan-Charvet, L. & Tall, A. R. Cholesterol Accumulation in Dendritic Cells Links the Inflammasome to Acquired Immunity. *Cell metabolism* **65**, 3176–3185 (2017).
12. Chan, C. T., Fenn, A. M., Harder, N. K., Mindur, J. E., McAlpine, C. S., Patel, J., Valet, C., Rattik, S., Iwamoto, Y., He, S., Anzai, A., Kahles, F., Poller, W. C., Janssen, H., Wong, L. P., Fernandez-

- Hernando, C., Koolbergen, D. R., ... Swirski, F. K. Liver X receptors are required for thymic resilience and T cell output. *Journal of Experimental Medicine* **217**, e20200318 (2020).
13. Shimizu, C., Kawamoto, H., Yamashita, M., Kimura, M., Kondou, E., Kaneko, Y., Okada, S., Tokuhisa, T., Yokoyama, M., Taniguchi, M., Katsura, Y. & Nakayama, T. Progression of T cell lineage restriction in the earliest subpopulation of murine adult thymus visualized by the expression of Ick proximal promoter activity. *International Immunology* **13**, 105–117 (2001).
 14. Lee, P. P., Fitzpatrick, D. R., Beard, C., Jessup, H. K., Lehar, S., Makar, K. W., Pérez-Melgosa, M., Sweetser, M. T., Schlissel, M. S., Nguyen, S., Cherry, S. R., Tsai, J. H., Tucker, S. M., Weaver, W. M., Kelso, A., Jaenisch, R. & Wilson, C. B. A critical role for Dnmt1 and DNA methylation in T cell development, function, and survival. *Immunity* **15**, 763–774 (2001).
 15. Wolfer, A., Bakker, T., Wilson, A., Nicolas, M., Ioannidis, V., Littman, D. R., Wilson, C. B., Held, W., MacDonald, H. R. & Radtke, F. Inactivation of Notch1 in immature thymocytes does not perturb CD4 or CD8 T cell development. *Nature Immunology* **2**, (2001).
 16. Yvan-Charvet, L., Ranalletta, M., Wang, N., Han, S., Terasaka, N., Li, R., Welch, C. & Tall, A. R. Combined deficiency of ABCA1 and ABCG1 promotes foam cell accumulation and accelerates atherosclerosis in mice. *Journal of Clinical Investigation* **117**, 3900–3908 (2007).
 17. Greenlee-Wacker, M. C. & Nauseef, W. M. IFN- γ targets macrophage-mediated immune responses toward *Staphylococcus aureus*. *Journal of Leukocyte Biology* **101**, (2017).
 18. Engwerda, C. R., Fox, B. S. & Handwerger, B. S. Cytokine production by T lymphocytes from young and aged mice. *Journal of immunology (Baltimore, Md. : 1950)* **156**, (1996).
 19. Lages, C. S., Lewkowich, I., Sproles, A., Wills-Karp, M. & Chougnet, C. Partial restoration of T-cell function in aged mice by in vitro blockade of the PD-1/PD-L1 pathway. *Ageing Cell* **9**, (2010).
 20. Nikolich-Zugich, J. Ageing and life-long maintenance of T-cell subsets in the face of latent persistent infections. *Nature Reviews Immunology* vol. 8 (2008).
 21. Quinn, K. M., Fox, A., Harland, K. L., Russ, B. E., Li, J., Nguyen, T. H. O., Loh, L., Olshanksy, M., Naeem, H., Tsyganov, K., Wiede, F., Webster, R., Blyth, C., Sng, X. Y. X., Tiganis, T., Powell, D., Doherty, P. C., ... La Gruta, N. L. Age-Related Decline in Primary CD8+ T Cell Responses Is Associated with the Development of Senescence in Virtual Memory CD8+ T Cells. *Cell Reports* **23**, 3512–3524 (2018).
 22. Yang, W., Bai, Y., Xiong, Y., Zhang, J., Chen, S., Zheng, X., Meng, X., Li, L., Wang, J., Xu, C., Yan, C., Wang, L., Chang, C. C. Y., Chang, T. Y., Zhang, T., Zhou, P., Song, B. L., ... Xu, C. Potentiating the antitumour response of CD8+ T cells by modulating cholesterol metabolism. *Nature* **531**, 651–5 (2016).
 23. Westerterp, M., Murphy, A. J., Wang, M., Pagler, T. A., Vengrenyuk, Y., Kappus, M. S., Gorman, D. J., Nagareddy, P. R., Zhu, X., Abramowicz, S., Parks, J. S., Welch, C. L., Fisher, E. A., Wang, N., Yvan-Charvet, L. & Tall, A. R. Deficiency of ABCA1 and ABCG1 in Macrophages Increases Inflammation and Accelerates Atherosclerosis in Mice. *Circulation research* **112**, 1456–1465 (2013).
 24. Out, R., Hoekstra, M., Habets, K., Meurs, I., de Waard, V., Hildebrand, R. B., Wang, Y., Chimini, G., Kuiper, J., van Berkel, T. J. C. & van Eck, M. Combined deletion of macrophage ABCA1 and

ABCG1 leads to massive lipid accumulation in tissue macrophages and distinct atherosclerosis at relatively low plasma cholesterol levels. *Arteriosclerosis, Thrombosis, and Vascular Biology* **28**, 258–264 (2008).

25. Yvan-Charvet, L., Pagler, T., Gautier, E. L., Avagyan, S., Siry, R. L., Han, S., Welch, C. L., Wang, N., Randolph, G. J., Snoeck, H. W. & Tall, A. R. ATP-binding cassette transporters and HDL suppress hematopoietic stem cell proliferation. *Science (New York, N.Y.)* **328**, 1689–93 (2010).
26. Kyaw, T., Winship, A., Tay, C., Kanellakis, P., Hosseini, H., Cao, A., Li, P., Tipping, P., Bobik, A. & Toh, B. H. Cytotoxic and proinflammatory CD8⁺ T lymphocytes promote development of vulnerable atherosclerotic plaques in ApoE-deficient mice. *Circulation* **127**, (2013).
27. Yurdagul, A., Kong, N., Gerlach, B. D., Wang, X., Ampomah, P., Kuriakose, G., Tao, W., Shi, J. & Tabas, I. ODC (Ornithine Decarboxylase)-Dependent Putrescine Synthesis Maintains MerTK (MER Tyrosine-Protein Kinase) Expression to Drive Resolution. *Arteriosclerosis, Thrombosis, and Vascular Biology* (2021) doi:10.1161/ATVBAHA.120.315622.
28. Proto, J. D., Doran, A. C., Gusarova, G., Yurdagul, A., Sozen, E., Subramanian, M., Islam, M. N., Rymond, C. C., Du, J., Hook, J., Kuriakose, G., Bhattacharya, J. & Tabas, I. Regulatory T Cells Promote Macrophage Efferocytosis during Inflammation Resolution. *Immunity* **49**, (2018).
29. Doran, A. C., Ozcan, L., Cai, B., Zheng, Z., Fredman, G., Rymond, C. C., Dorweiler, B., Sluimer, J. C., Hsieh, J., Kuriakose, G., Tall, A. R. & Tabas, I. CAMKII γ suppresses an efferocytosis pathway in macrophages and promotes atherosclerotic plaque necrosis. *Journal of Clinical Investigation* **127**, (2017).
30. Radhakrishnan, A., Goldstein, J. L., McDonald, J. G. & Brown, M. S. Switch-like Control of SREBP-2 Transport Triggered by Small Changes in ER Cholesterol: A Delicate Balance. *Cell Metabolism* **8**, (2008).
31. Xu, X., Zhang, S., Li, P., Lu, J., Xuan, Q. & Ge, Q. Maturation and emigration of single-positive thymocytes. *Clinical and Developmental Immunology* vol. 2013 (2013).
32. Zhang, C., Song, J. W., Huang, H. H., Fan, X., Huang, L., Deng, J. N., Tu, B., Wang, K., Li, J., Zhou, M. J., Yang, C. X., Zhao, Q. W., Yang, T., Wang, L. F., Zhang, J. Y., Xu, R. N., Jiao, Y. M., ... Wang, F. S. NLRP3 inflammasome induces CD4⁺ T cell loss in chronically HIV-1-infected patients. *Journal of Clinical Investigation* **131**, (2021).
33. Houtkooper, R. H., Argmann, C., Houten, S. M., Canfo, C., Jenninga, E. H., Andreux, P., Peñelope A., Thomas, C., Doenlen, R., Schoonjans, K. & Auwerx, J. The metabolic footprint of aging in mice. *Scientific Reports* **1**, (2011).
34. Larbi, A., Dupuis, G., Khalil, A., Douziech, N., Fortin, C. & Fülöp, T. Differential role of lipid rafts in the functions of CD4⁺ and CD8⁺ human T lymphocytes with aging. *Cellular Signalling* **18**, 1017–1030 (2006).
35. Larbi, A., Fortin, C., Dupuis, G., Berrougui, H., Khalil, A. & Fulop, T. Immunomodulatory role of high-density lipoproteins: impact on immunosenescence. *Age* **36**, (2014).
36. Sene, A., Khan, A. A., Cox, D., Nakamura, R. E. I., Santeford, A., Kim, B. M., Sidhu, R., Onken, M. D., Harbour, J. W., Hagbi-Levi, S., Chowers, I., Edwards, P. A., Baldan, A., Parks, J. S., Ory,

D. S. & Apte, R. S. Impaired cholesterol efflux in senescent macrophages promotes age-related macular degeneration. *Cell Metabolism* **17**, (2013).

37. Szondy, Z., Sarang, Z., Kiss, B., Garabuczi, É. & Köröskényi, K. Anti-inflammatory mechanisms triggered by apoptotic cells during their clearance. *Frontiers in Immunology* **909**, (2017).

REVIEWERS' COMMENTS

Reviewer #1 (Remarks to the Author):

The authors have addressed all of my concerns

Reviewer #2 (Remarks to the Author):

The authors have satisfactorily addressed all my comments.

Reviewer #3 (Remarks to the Author):

The manuscript has improved substantially. I have no further questions.

Response to Reviewers

Reviewer #1 (Remarks to the Author):

The authors have addressed all of my concerns

Reviewer #2 (Remarks to the Author):

The authors have satisfactorily addressed all my comments.

Reviewer #3 (Remarks to the Author):

The manuscript has improved substantially. I have no further questions.

We thank all reviewers for their comments on the resubmission that improved our manuscript considerably.